# Chloroquine resistance evolution in *Plasmodium falciparum* is mediated by the putative amino acid transporter AAT1

Malaria parasites break down host haemoglobin into peptides and amino acids in the digestive vacuole for export to the parasite cytoplasm for growth: interrupting this process is central to the mode of action of several antimalarial drugs. Mutations in the chloroquine (CQ) resistance transporter, *pfcrt*, located in the digestive vacuole membrane, confer CQ resistance in *Plasmodium falciparum*, and typically also affect parasite fitness. However, the role of other parasite loci in the evolution of CQ resistance is unclear. Here we use a combination of population genomics, genetic crosses and gene editing to demonstrate that a second vacuolar transporter plays a key role in both resistance and compensatory evolution. Longitudinal genomic analyses of the Gambian parasites revealed temporal signatures of selection on a putative amino acid transporter (*pfaat1*) variant S258L, which increased from 0% to 97% in frequency between 1984 and 2014 in parallel with the *pfcrt1* K76T variant. Parasite genetic crosses then identified a chromosome 6 quantitative trait locus containing *pfaat1* that is selected by CQ treatment. Gene editing demonstrated that *pfaat1* S258L potentiates CQ resistance but at a cost of reduced fitness, while *pfaat1* F313S, a common southeast Asian polymorphism, reduces CQ resistance while restoring fitness. Our analyses reveal hidden complexity in CQ resistance evolution, suggesting that *pfaat1* may underlie regional differences in the dynamics of resistance evolution, and modulate parasite resistance or fitness by manipulating the balance between both amino acid and drug transport.

Drug resistance in microbial pathogens complicates control efforts. Therefore, understanding the genetic architecture and the complexity of resistance evolution is critical for resistance monitoring and the development of improved treatment strategies. In the case of malaria parasites, deployment of five classes of antimalarial drugs over the past half century have resulted in well-characterized hard and soft selective sweeps associated with drug resistance, with both worldwide dissemination and local origins of resistance driving drug resistance alleles across the range of *Plasmodium falciparum*[1–3]. Chloroquine (CQ) monotherapy had a central role in an ambitious plan to eradicate malaria in the last century. Resistance to CQ was first observed in 1957 in southeast Asia (SEA), and subsequently arrived and spread across Africa from the late 1970s, contributing to the end of this ambitious global eradication effort[4].

Resistance to CQ has been studied intensively. The CQ resistance transporter gene (*pfcrt*, chromosome (chr.) 7) was originally identified using a *P. falciparum* genetic cross conducted between a CQ-resistant SEA parasite and a CQ-sensitive South American parasite generated in a chimpanzee host[5,6]. Twenty years of intensive research revealed the mechanistic role of the chloroquine resistance transporter (*pf*CRT)

✉ e-mail: Ashley.Vaughan@seattlechildrens.org; ferdig.1@nd.edu; tanderso@txbiomed.org

**Fig. 1 | Rapid allele frequency change and strong signals of selection around *pfaat1* in Gambia. a**, Temporal allele frequency change at SNPs coding for *pfaat1* S258L and *pfcrt* K76T between 1984 and 2014. The map and expanded West African region show the location of Gambia. **b**, Significance of haplotype differentiation across temporal populations of *P. falciparum* parasites determined using hapFLK. *P* values were corrected for multiple testing using the BH method. Significance thresholds at −log₁₀(false discovery rate (FDR)-corrected *P* value) of 5 are indicated with red dotted horizontal lines. Regions within the top 1% tail of FDR-corrected *P* values are marked with gene symbols. The strongest signals genome-wide seen are around *pfcrt*, *pfaat1* and *pfdhfr* (which is involved in pyrimethamine resistance). **c**, IBD, quantified with the isoRelate (iR) statistic, for temporal populations sampled from Gambia. *P* values were corrected for multiple testing using the BH method. Significance thresholds at −log₁₀(FDR-corrected *P* value) of 5 are indicated with red dotted horizontal lines. Regions within the top 1% tail of FDR-corrected *P* values are marked with gene symbols. Consistently high peaks of IBD around *pfcrt* and *pfaat1* are seen for parasite populations in all years of sampling. The 1990 sample (*n* = 13) is not shown in **c**.

haemoglobin digestion in the digestive vacuole, preventing conversion of haem, a toxic by-product of haemoglobin digestion, into inert haemozoin crystals. Parasites carrying CQ resistance mutations at *pf*CRT transport CQ out of the food vacuole, away from the site of drug action[7,8]. The *pfcrt* K76T single nucleotide polymorphism (SNP) is widely used as a molecular marker for CQ resistance[10], while additional variants within *pfcrt* modulate levels of resistance to CQ[11] and other quinoline drugs[12], and determine associated fitness costs[13]. While mutations in a second transporter located in the food vacuole membrane, the multidrug resistance transporter (*pfmdr1*), have been shown to modulate CQ resistance in some genetic backgrounds[14], the role of other genes in CQ resistance evolution remains unclear. In this Article, we sought to understand the contribution of additional parasite loci to CQ resistance evolution using a combination of population genomics, experimental genetic crosses and gene editing.

## Results

### Strong signatures of selection on *pfaat1*

Longitudinal population genomic data can provide compelling evidence of the evolution of drug resistance loci[15]. We conducted a longitudinal whole genome sequence analysis of 600 *P. falciparum* genomes collected between 1984 and 2014 in Gambia to examine signatures of selection under drug pressure (Supplementary Table 1). Following filtration using genotype missingness (<10%) and minor allele frequency (>2%), we retained 16,385 biallelic SNP loci from 321 isolates (1984 (134), 1990 (13), 2001 (34), 2008 (75) and 2014 (65)). The *pfcrt* K76T mutation associated with CQ resistance increased from 0% in 1984 to 88% in 2014. Notably, there was also rapid allele frequency change on chr. 6: the strongest differentiation is seen at an S258L mutation in a putative amino acid transporter, *pfaat1* (PF3D7_0629500, chr. 6), which increased during the same time period from 0% to 97% (Fig. 1a). Assuming a generation time (mosquito to mosquito) of 6 months for malaria parasites, these changes were driven by selection coefficients of 0.18 for *pfaat1* S258L, and 0.11 for *pfcrt* K76T (Extended Data Fig. 1). Both *pfaat1* S258L and *pfcrt* K76T mutations were absent in 1984 samples, but present in 1990, suggesting that they arose and spread in a short time window. Both *pfaat1* and *pfcrt* showed similar temporal haplotype structures in Gambia (Extended Data Fig. 2). These were characterized by almost complete replacement of well-differentiated haplotypes at both loci between 1984 and 2014. During this period, we also observed major temporal changes in another known drug resistance locus (*pfdhfr*) (chr. 4)[16] (Fig. 1b). That these rapid changes in allele frequency occur at *pfcrt*, *pfaat1* and *pfdhfr*, but not elsewhere in the genome (Fig. 1b), provides unambiguous evidence for strong directional selection.

Further evidence of strong selection on *pfaat1* and *pfcrt* came from the analysis of identity-by-descent (IBD) in Gambian parasite genomes. We saw the strongest signals of IBD in the genome around both *pfaat1* and *pfcrt* (Fig. 1c). These signals are dramatic, because there is minimal IBD elsewhere in the genome, with the exception of a strong signal centring on *pfdhfr* after 2008. Interestingly, the strong IBD is observed in all four temporal samples examined including 1984, before the spread of either *pfaat1* S258L or *pfcrt* K76T. However, only a single synonymous variant at *pfaat1* (I552I) and none of the CQ-resistant associated mutant variants in *pfcrt* were present at that time. CQ was the first-line treatment across Africa from the 1950s. These results are consistent with the possibility of CQ-driven selective sweeps conferring low-level CQ resistance before 1984, perhaps targeting promotor regions of resistance-associated genes. *pfaat1* has also been selected in other global locations: this is evident from prior population genomic analyses from Africa[17], SEA[18] and South America (SM)[19]. Plots summarizing IBD in these regions are provided in Extended Data Fig. 3.

Patterns of linkage disequilibrium (LD) provide further evidence for functional linkage between *pfcrt* and *pfaat1*. The strongest genome-wide signal of inter-chromosomal LD was found between

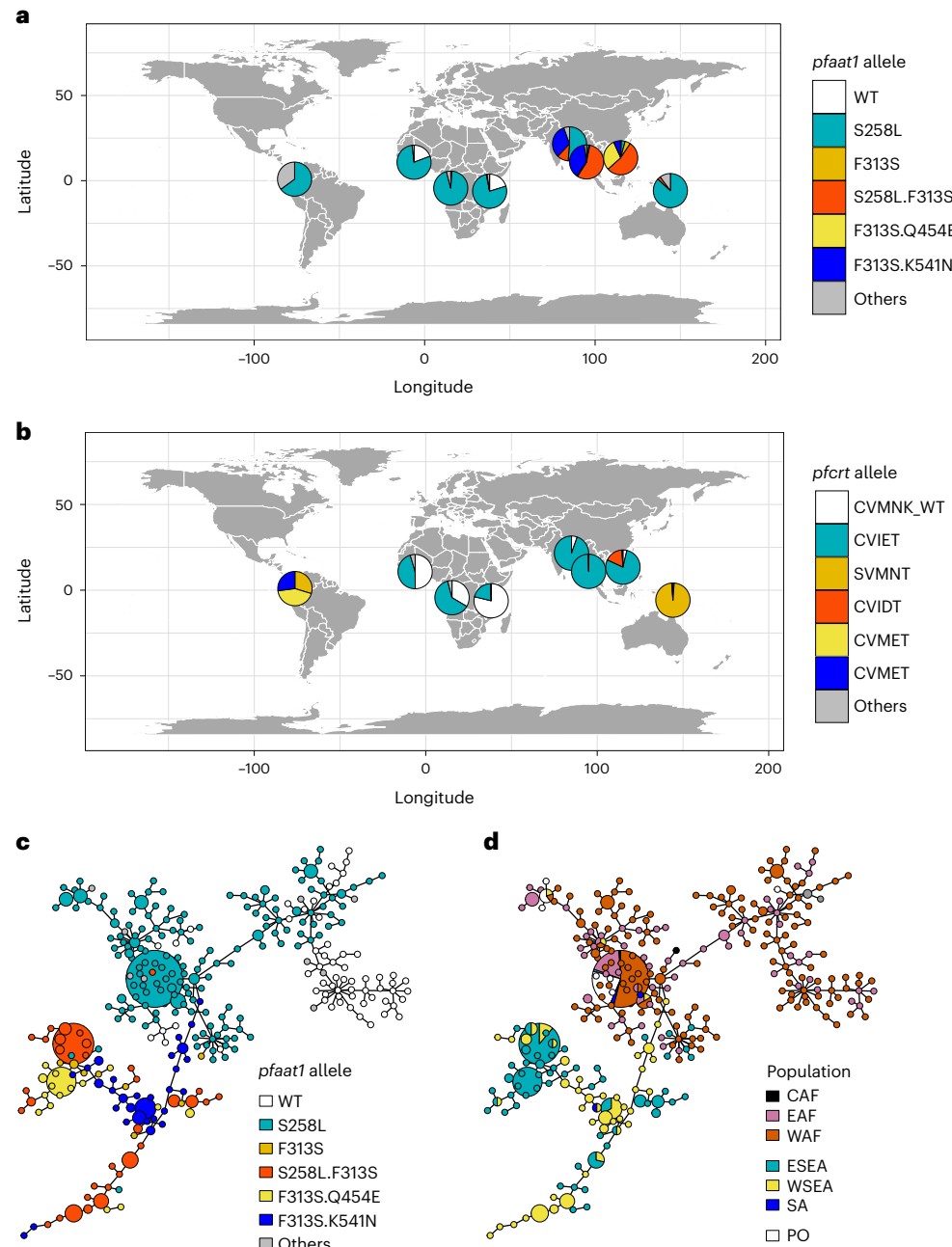

**Fig. 2 | Distinctive trajectory of *pfaat1* evolution in SEA. a**, Global distribution of *pfaat1* alleles. **b**, Comparable maps showing percentages of *pfcrt* haplotypes for amino acids 72–76. The coloured segments show the major *pfcrt* haplotypes varying at the K76T mutation. We used dataset from MalariaGEN release 6 for *pfaat1* and *pfcrt* allele frequency analysis. Data used for the figure are contained in Supplementary Table 2. Only samples with monoclonal infections ($N$ = 4,051) were included (1,233 from west Africa (WAF), 415 from east Africa (EAF), 170 from central Africa (CAF), 994 from east southeast Asia (ESEA), 998 from west southeast Asia (WSEA), 37 from south Asia (SA), 37 from south America (SM) and 167 from the Pacific Ocean region (PO)). **c,d**, MSNs of haplotypes coloured by *pfaat1* allele (**c**) and geographical location (**d**), respectively. Networks were constructed from 50 kb genome regions centred by *pfaat1* (25 kb up- and downstream). This spans the genome regions showing LD around *pfaat1* (Supplementary Fig. 2). A total of 581 genomes with the highest sequence coverage were used to generate the network. The networks were generated on the basis of 1,847 SNPs (at least one mutant in the full dataset−MalariaGEN release 6). Circle size indicates number of samples represented (smallest, 1; largest, 87). Haplotypes from the same region (Asia or Africa) were clustered together, indicating independent origin of *pfaat1* alleles.

these two loci both in our Gambian data (Supplementary Fig. 1) and in samples from across Africa[20]. LD between *pfaat1* and *pfcrt* was strongest in 2001, and then decayed in 2008 and 2014 (Supplementary Figs. 1 and 2), consistent with maintenance of LD during intensive CQ usage, and subsequent LD decay after CQ monotherapy was replaced by sulfadoxine-pyrimethamine + CQ combinations in 2004, and then with artemisinin combinations in 2008 (ref. 16).

Correlations in allele frequencies are expected between *pfcrt* and *pfaat1* if these loci are interacting or are co-selected. Frequencies of the CVIET haplotype for amino acids 72−76 in *pf*CRT are significantly correlated with allele frequencies of *pfaat1* S258L in West Africa (WAF) ($R^2$ = 0.65, $P$ = 0.0017) and across all African populations ($R^2$ = 0.44, $P$ = 0.0021) (Extended Data Fig. 4). This analysis further strengthens the argument for co-evolution or epistasis between these two genes.

## Divergent selection on *pfaat1* in SEA

We examined the haplotype structure of *pfaat1* from *P. falciparum* genomes (MalariaGEN release 6 (ref. [21])) (Fig. 2 and Supplementary Table 2). The *pfaat1* S258L SNP is at high frequency in SEA (58%) but is found on divergent flanking haplotypes suggesting an independent origin from the *pfaat1* S258L in Gambia and elsewhere in Africa (Fig. 2c,d and Extended Data Fig. 5). Hendon et al.[18] reached the same conclusion for the chr. 6 region using an IBD analysis of parasites from global locations. Convergent evolution of *pfaat1* S258L provides further evidence for selection, and contrasts with *pfcrt* and *pfdhfr*, where resistance alleles that spread in Africa had an Asian origin[1,2]. The evolution of *pfaat1* is more complex in SEA than elsewhere in the world. There are three additional common derived amino acid changes in SEA. *pfaat1* F313S has spread close to fixation in SEA (total 96%, $F_{ST}$ = 0.91 compared with African samples) paired with *pfaat1* S258L (55%), Q454E (15%) or K541N (22%). The pairing of F313S with three different mutations, suggests that F313S arose first. We speculate that these geographically localized *pfaat1* haplotypes have had an important role in CQ resistance evolution in SEA and could also reflect geographic differences in the historical use of other quinoline drugs (mefloquine, quinine, piperaquine and lumefantrine) in this region[22].

## Parasite genetic crosses using humanized mice identify a QTL containing *pfaat1*

*P. falciparum* genetic crosses can be achieved with human-liver chimaeric mice, reviving and enhancing this powerful tool for malaria genetics[23,24], after use of great apes for research was banned. We used two independent biological replicates of a cross between the CQ-sensitive African parasite, 3D7, and a recently isolated CQ-resistant parasite from the Thailand–Myanmar border, NHP4026 (Supplementary Table 3). We then compared genome-wide allele frequencies in CQ-treated and control-treated progeny pools to identify quantitative trait loci (QTL) (Supplementary Table 4). This bulk segregant analysis (BSA)[25] of progeny parasites robustly identified the chr. 7 locus containing *pfcrt* as expected, validating our approach (Fig. 3a and Supplementary Figs. 3 and 4). We were also intrigued to see a significant QTL on chr. 6 in each of the replicate crosses (Fig. 3, Supplementary Figs. 3 and 4 and Extended Data Fig. 6). We prioritized genes within the 95% confidence interval of each QTL (Supplementary Table 5) by inspecting the SNPs and indels that differentiated the two parents (Supplementary Table 6). The chr. 6 QTL spanned from 1,013 kb to 1,283 kb (270 kb) and contained 60 genes. Of these, 54 are expressed in blood stages, and 27 have non-synonymous mutations that differentiate 3D7 from NHP4026. *pfaat1* was located at the peak of the chr. 6 QTL (Fig. 3c). NHP4026 carried two derived non-synonymous mutations in *pfaat1* (S258L and F313S) compared with 3D7, which carries the ancestral allele. We thus hypothesized that one or both of these *pfaat1* SNPs may be driving the chr. 6 QTL.

We isolated individual clones from the bulk 3D7 × NHP4026 $F_1$ progeny to recover clones with all combinations of parental alleles at the chr. 6 and chr. 7 QTL loci. We cloned parasites both from a bulk progeny culture that was CQ selected (96 h at 250 nM CQ) and from a control culture. This generated 155 clonal progeny: 100 from the CQ-selected culture, 62 of which were genetically unique, and 55 from the untreated control culture, of which 47 were unique (Fig. 4a). We compared allele frequencies between these two progeny populations (Fig. 4b), revealing significant differences at both chr. 6 and chr. 7 QTL

regions, paralleling the BSA results. We observed a dramatic depletion of the NHP4026 CQ-resistant allele at the chr. 7 QTL in control-treated cultures, consistent with strong selection against CQ resistant *pfcrt* alleles in the absence of CQ selection. Conversely, all progeny isolated after CQ treatment harboured the NHP4026 CQ-resistant *pfcrt* allele. The inheritance of the *pfcrt* locus (chr. 7) and the *pfaat1* locus (chr. 6) was tightly linked in the isolated clones (Fig. 4c). To further examine whether the cross data were consistent with epistasis or co-selection, we examined a larger sample of recombinant clones isolated from five independent iterations of this genetic cross in the absence of CQ selection. This revealed significant under-representation of clones with genotype *pfcrt* 76T and *pfaat1* 258S/313F (WT) (Supplementary Table 7, $\chi^2$ = 12.295, $P$ = 0.0005). These results are consistent with the strong LD between these loci observed in nature (Extended Data Fig. 4 and Supplementary Fig. 1)[20] and suggest a functional relationship between the two loci. A role for *pfaat1* S258L/F313S in compensating for the reduced fitness of parasites bearing *pfcrt* K76T is one likely explanation for the observed results.

We next measured in vitro CQ half-maximal inhibitory concentration ($IC_{50}$) values for 18 parasites (a set of 16 progeny and both parents), carrying all combinations of the chr. 6 and chr. 7 QTL alleles (Supplementary Fig. 5 and Supplementary Table 8). The NHP4026 parent was the most CQ-resistant parasite tested. All progeny that inherited NHP4026 *pfcrt* showed a CQ-resistant phenotype while all progeny that inherited 3D7 *pfcrt* were CQ sensitive, consistent with previous reports. The effect of *pfcrt* alleles on parasite CQ resistance was significant on the basis of a two-way analysis of variance test ($P$ = 7.52 × 10^−11). We did not see an effect of the *pfaat1* genotypes on $IC_{50}$ values in clones carrying *pfcrt* 76T ($P$ = 0.06) or *pfcrt* 76K ($P$ = 0.19). This analysis has limited power because only two progeny parasites were recovered with *pfaat1* 258S/313F (WT) in combination with *pfcrt* 76T (Fig. 4a and Supplementary Fig. 5), but is consistent with the *pfaat1* QTL being driven by parasite fitness in our genetic crosses. We therefore focused on gene manipulation of isogenic parasites for functional analysis.

## Functional validation of the role of pfaat1 in CQ resistance

We utilized CRISPR–Cas9 modification of the NHP4026 CQ-resistant parent to investigate the effects of mutations in *pfaat1* on CQ $IC_{50}$ drug response and parasite fitness (Fig. 5). NHP4026 *pfaat1* carries the two most common SEA non-synonymous changes (S258L and F313S) (Fig. 2), relative to the sensitive 3D7 parent. We edited these positions back to the ancestral state both singly and in combination and confirmed the modifications in three to five clones isolated from independent edits for each allelic change (Fig. 5a). We then determined CQ $IC_{50}$ values and measured fitness using pairwise competition experiments for parental NHP4026^258L/313S, the single mutations NHP4026^258L/313F, NHP4026^258S/313S and the ancestral allele NHP4026^258S/313F. This revealed a highly significant impact of the S258L mutation, which increased CQ $IC_{50}$ values 1.5-fold, and a more moderate but significant impact of F313S and the double mutation (S258L/F313S), relative to the ancestral (258S/313F) allele (Fig. 5b). The observation that 258L shows reduced $IC_{50}$ values in combination with the F313S mutation reveals an epistatic interaction between these amino acid variants (Fig. 5b).

We also examined the effect of the S258L and F313S substitutions on responses to other quinoline drugs. The results revealed significant effects of *pfaat1* substitutions on quinine, amodiaquine and lumefantrine $IC_{50}$ responses, and no effect on the mefloquine

**Fig. 3 | Genetic crosses and BSA reveal two QTL after CQ selection. a**, Allele frequency plots across the genome before and after CQ treatment. Lines with the same colour indicate results from technical replicates. **b**, QTLs identified using the *G′* approach. Lines with the same colour indicate results from technical replicates. **a** and **b** include results from BSA with 48 h CQ treatment with samples collected at day 4. For the complete BSA from different collection timepoints and drug treatment duration under different CQ concentrations, see Supplementary

Figs. 3 and 4. **c**, Fine mapping of the chr. 6 QTL. The 95% confidence intervals (CIs) were calculated from the 250 nM CQ treated samples, including data from different collection time points (day 4 for 48 h CQ treatment and day 5 for 96 h CQ treatment), pools (pool 1 and pool 2), and drug treatment duration (48 h and 96 h). Light cyan shadow shows boundaries of the merged CIs of all the QTLs. Each line indicates one QTL; black dashed line indicates threshold for QTL detection (*G′* = 20). The vertical red dashed line indicates *pfaat1* location.

IC$_{50}$ (Extended Data Fig. 7). Notably, these IC$_{50}$ value shifts were well below the threshold associated with clinical resistance. Consequently, although mutations in *pfaat1* can subtly impact susceptibly to a range

of compounds, these results are consistent with CQ treatment being the primary selective force that drove the *pfaat1* S258L and F313S mutations along with those in *pfcrt*.

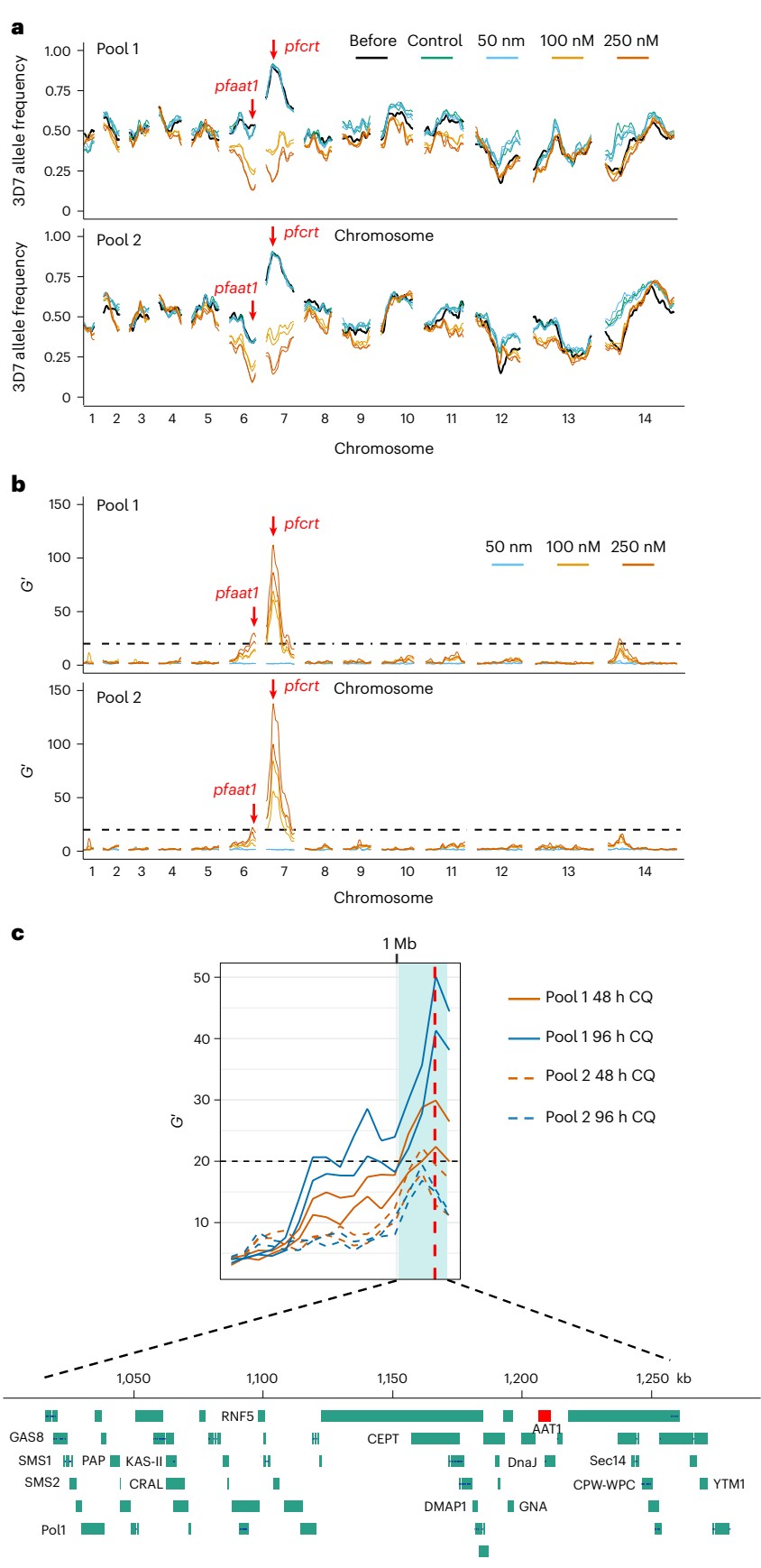

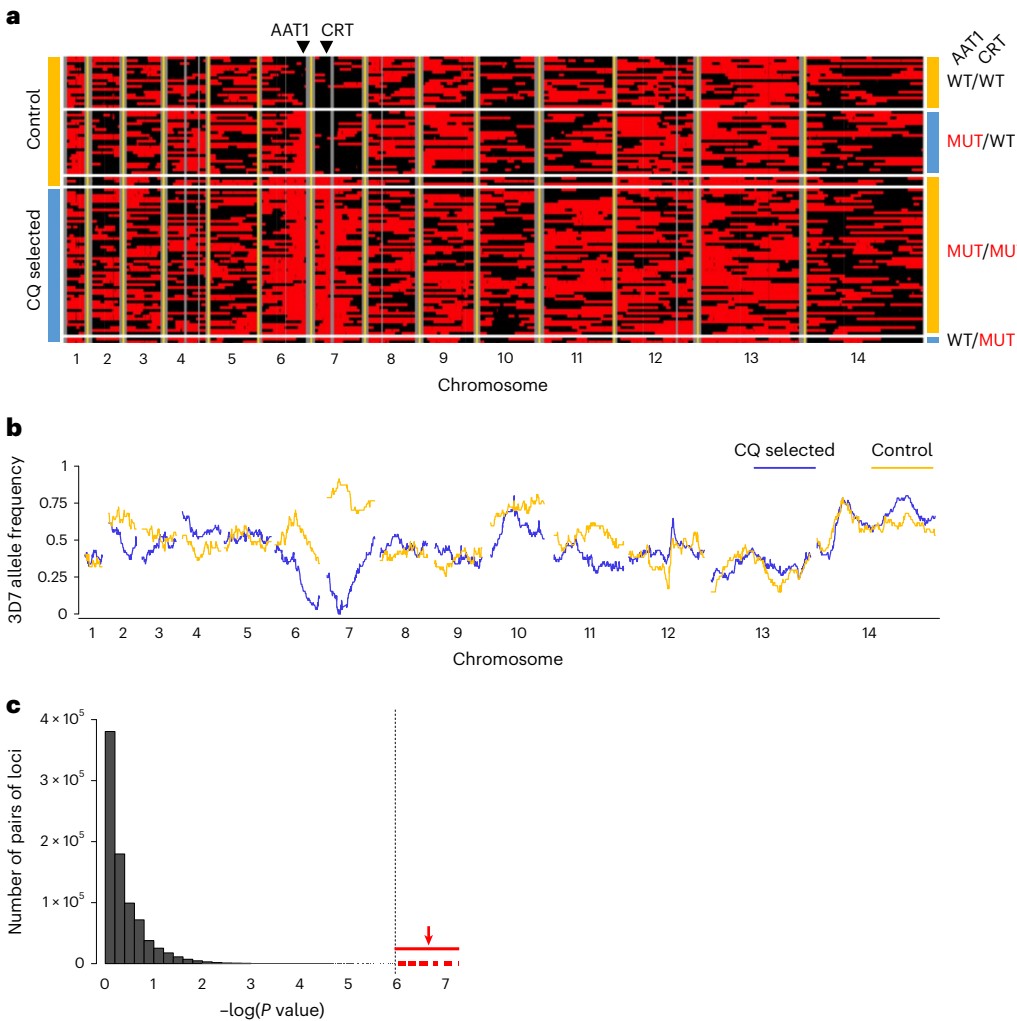

**Fig. 4 | Analysis of cloned progeny reveals linkage and epistatic interactions between *pfcrt* and *pfaat1*. a**, Allelic inheritance of 109 unique recombinant progeny. Black and red blocks indicate alleles from 3D7 and NHP4026, separately. Vertical grey lines show non-core regions where no SNPs were genotyped. Left: clones isolated from recombinant progeny pools with or without CQ treatment are labelled. Right: *pfaat1* and *pfcrt* alleles are labelled. WT indicates *pfaat1* and *pfcrt* alleles from 3D7 and MUT indicates alleles from NHP4026. The location of *pfaat1* and *pfcrt* is marked using black triangles on the top of the panel.

**b**, Genome-wide 3D7 allele frequency plot of unique progeny cloned from pools after 96 h of CQ (250 nM) treatment (blue) or from control pools (gold). **c**, Linkage between loci on different chromosomes measured by Fisher's exact test. The dotted vertical line marks the Bonferroni-corrected significance threshold (one-tailed), while points shown in red are comparisons between SNPs flanking *pfaat1* and *pfcrt*. Supplementary Table 7 shows non-random associations between genotypes in parasite clones recovered from untreated cultures.

Mutations conferring drug resistance often carry fitness costs in the absence of drug treatment. We thus examined parasite fitness by conducting pairwise competition experiments with the parental NHP4026 parasite against the same mutant *pfaat1* parasites created above. This revealed significant differences in fitness (Fig. 5c). The 258L/313F allele that showed a selective sweep in Gambia was the least fit of all genotypes, the ancestral allele (258S/313F) carried by the 3D7 parent was the most fit, while the 258S/313S mutation showed a similar fitness to the NHP4026 parent (258L/313S). These results also revealed strong epistatic interactions in fitness. While the 258L/313F allele that conferred high CQ IC$_{50}$ values (Fig. 5b) carried a heavy fitness penalty (Fig. 5c), fitness was partially restored by the 313S mutation in the 258L/313S allele that predominates in SEA. Together these results show that the *pfaat1* S258L substitution underpins a 1.5-fold increase in CQ resistance that probably drove its selective spread in Gambia. However, S258L carries a high fitness cost that in SEA parasites was probably mitigated by the substitution, F313S. Overall, these results demonstrate a large effect of *pfaat1* mutations on fitness of parasites carrying *pfcrt* K76T resistance alleles.

The editing experiments reveal that clones carrying the ancestral *pfaat1* allele in combination with *pfcrt* K76T show the highest fitness. By contrast, the close association of *pfaat1* S258L/F313S with *pfcrt* K76T in progeny from the genetic crosses revealed the opposite relationship. We speculate that these opposing results may reflect differing selection pressures in blood stage parasites in the case of CRISPR experiments, or in the mosquito and liver stages of the life cycle in the case of genetic crosses. The gene editing studies were conducted with a single SEA parasite genotype (NHP4026). While African *pfcrt* CQR alleles originated in SEA and share a common ancestor and identity at amino acids 72–76, most SEA parasites (including NHP4026) carry one or two additional mutations in *pfcrt* (N326S and I356T) associated with higher CQ IC$_{50}$ values and reduced fitness[13,26]. The predominant *pfcrt* haplotype in Gambia differs from NHP4026 at one amino acid, carrying the ancestral 326S, while NHP4026 carries the 326N mutation[13]. It will be important to examine the effect of *pfaat1* mutations on African genetic backgrounds in future work.

To further understand how *pfaat1* S258L impacts parasite phenotype, we used a yeast heterologous expression system. WT *pfaat1* is

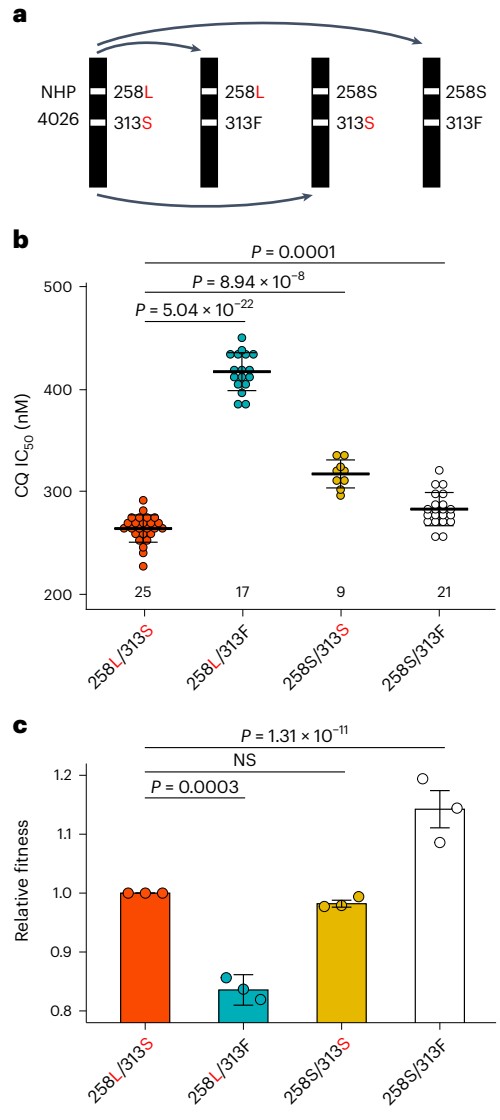

**Fig. 5 | Allelic replacement impacts drug response and parasite fitness.**
**a**, CRISPR–Cas9 gene editing. Starting with the NHP4026 parent, we generated all combinations of the SNP-states at *pfaat1*. **b**, Drug response. Each dot indicates one replicate $IC_{50}$ measurement: we used two to four independent CRISPR edited clones for each haplotype examined. The number of biological replicates is shown above the *x* axis. We conducted pairwise *t*-tests (two-tailed) to compare $IC_{50}$ values between parasite lines, without adjustment for multiple comparisons. Haplotypes are shown on the *x* axis with derived amino acids shown in red. Bars show means ± s.e.m.), while significant differences between haplotypes are marked. **c**, Fitness. The bars show mean relative fitness (±1 s.e.m.) measured in replicated competition experiments, and dots represent fitness from individual measurements. We conducted three independent competition experiments for each edited parasite group in the absence of CQ. *F*-statistic was used to compare fitness between parasite lines. Results from assays for each edited group were combined using meta-analyses with random effects. For allele frequency changes for each competition experiment, see Extended Data Fig. 10. NS, not significant.

expressed in the yeast plasma membrane[27], where it increases quinine and CQ uptake conferring sensitivity to quinoline drugs, resulting in reduced growth. CQ uptake was previously shown to be competitively inhibited by the aromatic amino acid tryptophan, suggesting a role for *pfaat1* in drug and amino acid transport[27]. We therefore expressed *pfaat1* S258L in yeast, which restored yeast growth in the presence of high levels of CQ (Extended Data Fig. 8). Interestingly, expression of another amino acid variant (T162E), responsible for CQ resistance in

rodent malaria parasites (*Plasmodium chabaudi*)[28], also prevents accumulation of quinoline drugs within yeast cells and restores cell growth in the presence of 1 mM CQ[27]. Together, these new and published results suggest that yeast expression of *pfaat1* mutations impact resistance and fitness by altering the rates of amino acid and quinoline transport.

We evaluated three-dimensional structural models based on the 3D7 PfAAT1 amino acid sequence using AlphaFold[29] and I-TASSER[30] (Extended Data Fig. 9). While *pf*CRT has 10 membrane-spanning helices[31], *pf*AAT1 has 11; this was corroborated using the sequence-based membrane topology prediction tool TOPCONS[32]. The common *pf*AAT1 mutations S258L, F313S and Q454E are situated in membrane-spanning domains, while K541L is in a loop linking domains 9 and 10. The location of these high-frequency non-synonymous changes in membrane-spanning domains has strong parallels with *pf*CRT evolution[31] and is consistent with a functional role for these amino acids in transporter function.

## Discussion

Identification of *pfcrt* as the major determinant of CQ resistance was a breakthrough that transformed the malaria drug resistance research landscape, but the contribution of additional genetic factors in the evolution and maintenance of CQ resistance remained unclear[26,33]. By combining longitudinal population genomic analysis spanning the emergence of CQ resistance in Gambia, analysis of bulk populations and progeny from controlled genetic crosses, and functional validation using both *P. falciparum* and yeast, we find compelling evidence that a second locus, *pfaat1*, has had an important role in CQ resistance evolution. This powerful combination of approaches allowed us to examine critical *pfaat1* variants that contribute to the architecture of CQ resistance and interactions between *pfcrt* and *pfaat1*.

Our results provide compelling evidence that consolidates disparate observations from several systems suggesting a role for *pfaat1* in drug resistance evolution. In the rodent malaria parasite *P. chabaudi*, a mutation (T162E) in the orthologous gene (*pcaat1*) was found to be a determinant of low-level CQ resistance in laboratory-evolved resistance[28]. In *P. falciparum* genome-wide association studies, the S258L mutation of *pfaat1* was associated with CQ resistance in field isolates collected along the China–Myanmar border[34], while *pfcrt* K76T and *pfaat1* S258L show the strongest LD between physically unlinked chromosomes genome-wide[20]. In addition, mutations in *pfaat1* have been linked to the in vitro evolution of resistance in *P. falciparum* to three different drug scaffolds[35]. Previous work identified strong signatures of recent selection in parasites in Africa at regions surrounding *pfcrt*, *pfaat1* and other drug resistance loci[16,17,36]; similar signatures of selection are seen in Asia and SM[18,19], while *pfaat1* was highlighted in a list of *P. falciparum* genes showing extreme geographical differentiation[21].

The different *pfaat1* haplotypes in Africa and Asia may be partly responsible for the contrasting evolution of CQ resistance in these two continents. CQ-resistant parasites carrying both *pfcrt* K76T and *pfaat1* S258L spread across Africa, but after removal of CQ as the first-line drug, the prevalence of CQ-resistant parasites declined in many countries[37–39]. This is consistent with the low fitness of parasites carrying *pfcrt* K76T and *pfaat1* S258L in the absence of drug pressure, and intense competition within malaria parasite infection in Africa[40].

In contrast, *pfcrt* K76T has remained at or near fixation in many SEA countries[21,41] (Fig. 2). On the Thailand–Myanmar border, CQ resistance has remained at fixation since 1995, when CQ was removed as first-line treatment of *P. falciparum* malaria[41]. Our *pfaat1* mutagenesis results demonstrate that parasites bearing *pfaat1* 258L/313S show reduced $IC_{50}$ values but elevated fitness relative to *pfaat1* 258L/313F. We speculate that restoration of fitness by F313S may help to explain retention of CQ-resistant *pfcrt* K76T alleles in SEA. The alternative hypothesis—that high frequencies of F313S mutations are driven by widespread use of other quinoline partner drugs in SEA[42]—is not supported, because we see only minor impacts of this substitution on

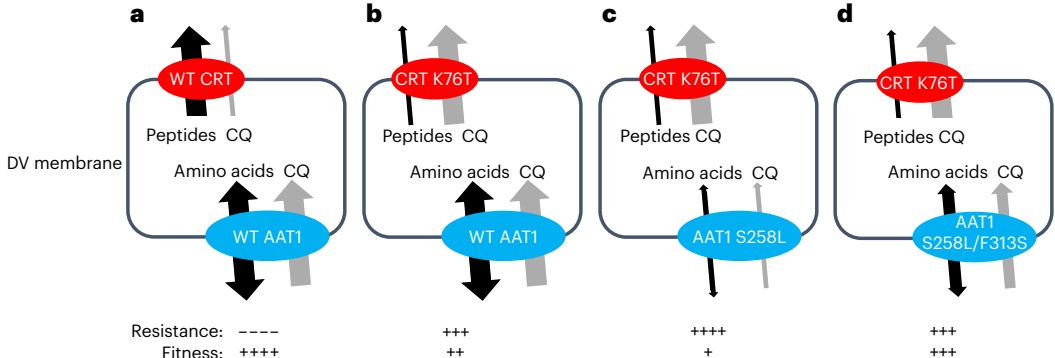

**Fig. 6 | Model for involvement of *pfaat1* haplotypes in CQ resistance and fitness.** *pf*CRT (red) and *pf*AAT1 (blue) are both situated in the digestive vacuole (DV) membrane. **a**, WT *pf*CRT and *pf*AAT1 transport peptides and aromatic amino acids, respectively, as well as CQ. **b**, *pf*CRT K76T exports CQ from the DV away from its site of action, leading to elevated resistance but transports peptides inefficiently leading to a loss of fitness. **c**, *pf*AAT1 S258L reduces entry of CQ into the DV, leading to elevated resistance, but amino acid flux is affected, leading to a loss of fitness. **d**, The *pf*AAT1 S258L/F313S double mutation increases CQ influx in comparison with the S258L alone but the amino acid transport function is restored, leading to reduced IC₅₀ values and increased fitness in the absence of drug treatment.

response to lumefantrine, quinine, mefloquine and amodiaquine (Extended Data Fig. 7).

Mutations in *pfcrt* confer CQ resistance by enabling efflux of CQ across the digestive vacuole membrane, away from its site of action[8]. *pf*AAT1 is also located in the digestive vacuole membrane[35], where it probably acts as a bidirectional transporter of aromatic amino acids[9,43]. Given the structural similarity of quinoline drugs and aromatic amino acids, *pfaat1* mutations may modulate the ability of *pf*AAT1 to transport CQ and/or amino acids[27,43]. The *pfaat1* S258L mutation could potentiate resistance by either increasing efflux of CQ out of the digestive vacuole or reducing the rate of entry into the vacuole. Given that this *pfaat1* mutation blocks entry of quinoline drugs into yeast cells when heterologously expressed in the yeast cell membrane[27], we hypothesize that the *pfaat1* S258L mutation reduces CQ uptake into the food vacuole (Fig. 6). Our mutagenesis analyses show that the S258L allele has a high fitness cost, perhaps due to a decreased capacity for amino acid transport from the vacuole. Interestingly, comparison of the *pfaat1* S258L/F313S haplotype segregating in our genetic cross with the WT *pfaat1* allele generated using gene editing revealed only marginal increases in IC₅₀ values and limited reductions in fitness. This is consistent with the F313S mutation restoring the natural *pfaat1* function of transporting amino acids, thereby reducing osmotic stress and starvation, while also partially reducing levels of CQ resistance (Fig. 6). That this haplotype has reached high frequency in SEA may contribute to the maintenance of *pfcrt* K76T alleles long after the removal of CQ as a first line drug. This model (Fig. 6) provides a working hypotheses that can be tested in future work examining the role of *pf*AAT1 and *pf*CRT.

Our results reveal hidden complexity in CQ resistance evolution: drug treatment has driven global selective sweeps acting on mutations in an additional transporter (*pf*AAT1) located in the *P. falciparum* digestive vacuole membrane, which fine tune the balance between nutrient and drug transport, revealing evidence for epistasis and compensation, and impacting both drug resistance and fitness.

## Methods

### Ethics approval and consent to participate
The study was performed in accordance with the Guide for the Care and Use of Laboratory Animals of the US National Institutes of Health (NIH). The Seattle Children's Research Institute (SCRI) has an Assurance from the Public Health Service through the Office of Laboratory Animal Welfare for work approved by its Institutional Animal Care and Use Committee. All of the work carried out in this study was specifically reviewed and approved by the SCRI Institutional Animal Care and Use Committee.

### Project design
The project design is summarized in Supplementary Fig. 6. In brief, we use (1) population genomic analyses, (2) genetic crosses and quantitative genetics analysis followed by (3) functional analyses to investigate the role of additional loci in CQ resistance.

### Gambia population analysis
***P. falciparum* genome sequences.** *P. falciparum*-infected blood samples collected from central (Farafenni) and coastal (Serrekunda) Gambia in 1984 and 2001, were processed for whole blood DNA and *P. falciparum* genomes and deep sequenced at the Wellcome Trust Sanger Institute. Data from isolates collected from coastal Gambia in 2008 and 2014 had been published previously[44,45] (Supplementary Table 1). Before sequencing, *P. falciparum* genomes were amplified from whole blood DNA of each sample from 1984 and 2001 using selective whole genome amplification (WGA) and then sequenced (paired-end reads) on the Illumina HiSeq platform[46]. Reads were mapped to the *P. falciparum* 3D7 reference genome using bwa mem (http://bio-bwa. sourceforge.net/). Mapping files (Binary Alignment Map) were sorted and deduplicated by Picard tools v2.0.1 (http://broadinstitute.github. io/picard/), and SNP and indel were called with GATK HaplotypeCaller (https://software.broadinstitute.org/gatk/) following best practices (https://www.malariagen.net/data/pf3K-5). Variant call format (VCF) files were generated by chromosome, merged using bcftools (https:// samtools.github.io/bcftools/bcftools.html) and filtered using vcftools (https://vcftools.sourceforge.net/). After filtration, only biallelic SNP variants with a VQSLOD score of ≥2, a map quality >30 and supported ≥5 reads per allelic variant were retained. SNPs with minor allele frequency <2% were removed from our analysis. We also removed samples with >10% genotypes missing. In the final dataset, there were in total 16,385 biallelic SNP loci and 321 isolates (1984 (134), 1990 (13), 2001 (34), 2008 (75) and 2014 (65)). The complexity of infection (monogenomic or polygenomic) was estimated as the inbreeding coefficient Fws from the merged VCF file using R package Biomix. The short-read sequence data analysed are listed in Supplementary Table 1.

**Allele frequencies and pairwise differentiation.** For each sample with a complexity of infection greater than 1, the allele with most reads was retained for mixed-allele genotypes to create a virtual haploid genome variation dataset. Allele frequencies were calculated in

plink, and pairwise differences between temporal populations and genetic clusters were estimated by Fst using Weir and Cockerham's method applied in the hierfstat package in R. The likelihood ratio test for allele frequency difference pFST was further calculated using vcflib. For a combined pFST *P* value, the fisher method was performed in R metaseq package. The summary *P* values were corrected for multiple testing using Benjamini–Hochberg (BH) method. To examine haplotype sharing at *pfaat1* (Pf3D7_06_v3:1,213,102-1,217,313) and *pfcrt* (Pf3D7_07_v3:403222-406317) between isolates from the different years of sampling in Gambia, we extracted the IBD matrix using isoRelate R package[18] for all pairs of isolates for gene regions spanning an additional 25 kb on each flank. We generated relatedness networks using the R package igraph following the scripts in the isoRelate R package[18]. Isolates are connected if they show >90% IBD.

**Genome scans for selection.** We considered samples collected in the same year as a single population irrespective of the location of collection. We used the hapFLK approach to detect signatures of positive selection through haplotype differentiation following hierarchical clustering of Gambian temporal population groups compared with an outgroup from Tanzania, as previously described[47]. *P* values were computed for each SNP-specific value using the Python script provided with the hapFLK program, and values were corrected for multiple testing using the BH method. Secondly, we used pairwise relatedness based on identity by descent to derive an iR statistic for each SNP as implemented by the IsoRelate[18] package in R. Regions with overlapping iR and hapFLK $-\log_{10}P$ values >5 were considered as regions of interest.

**Population analysis on *pfaat1* and *pfcrt* evolution**
**Datasets.** We included two datasets in this study: (1) genotypes of 7,000 worldwide *P. falciparum* samples from MalariaGEN Pf community project (version 6.0) (ref. [21]). This dataset includes samples from south America (SM), west Africa (WAF), Central Africa (CAF), East Africa (EAF), South Asia (SA), the western part of southeast Asia (WSEA), the eastern part of southeast Asia (ESEA) and the Pacific Oceania (PO) region. (2) We also included 194 Thailand samples with whole genome sequencing data available from Cerqueira et al.[15], and merged them into the WSEA population. Duplicate sequences were removed according to the sample's original ID (Hypercode). Only samples with single parasite infections (within-host diversity $F_{WS} > 0.90$) and >50% of SNP loci genotyped were included for further analysis. A total of 4,051 samples remained after filtration (Supplementary Table 2). Non-biallelic SNPs and heterozygous variant calls were further removed from the dataset. We then extracted genotype data at *pfaat1* and *pfcrt* gene regions and calculated the allele frequencies (Fig. 2a).

***pfaat1* haplotypes and evolutionary relationships.** To minimize the effect of recombination, we extracted 1,847 SNPs distributed within 25 kb upstream and 25 kb downstream of the *pfaat1* gene. Only samples with all 1,847 SNPs genotyped (581/4,051) were used for evolutionary analysis. To visualize the population structure, we calculated the pairwise genetic distance between samples and generated a minimum spanning network (MSN; Fig. 2b and Extended Data Fig. 5), using R package poppr. We compared genome sequences (PlasmoDB, version 46) between *P. falciparum* and *Plasmodium reichenowi* and extracted genotypes at 1,803/1,847 common loci. We then built an unweighted pair group method with arithmetic mean (UPGMA) tree rooted by *P. reichenowi* using the 581 haplotypes and 1,803 SNPs (Extended Data Fig. 5), using the R packages ape and phangorn under default parameters. MSN network and unweighted pair group method with arithmetic mean tree were plotted with ggplot2.

**Genetic cross and BSA**
**Genetic cross preparation.** We generated genetic crosses between parasite 3D7 and NHP4026 (ref. [48]), using FRG NOD huHep mice

with human chimaeric livers and *Anopheles stephensi* mosquitoes as described previously[23–25,49,50]. 3D7 is a parasite of African origin[51] that has been maintained in the lab for decades and is CQ sensitive, while NHP4026 was cloned from a patient visiting the Shoklo Malaria Research Unit clinic on the Thailand–Myanmar border (2007) and is CQ resistant (Supplementary Table 3). We generated three recombinant pools using independent cages of infected mosquitoes: these are independent pools of recombinants[48]. The estimated number of recombinant genotypes in each pool was ~2,800 (ref. [48]). We used two pools (pool 1 and pool 2) maintained in AlbuMAX-based culture medium for this study.

**Drug treatment and sample collection.** For each recombinant pool, the parasite culture was expanded under standard culture conditions[25]. Briefly, cultures were maintained in complete medium at 5% haematocrit in O⁺ red blood cells (RBCs) (Biochemed Services) at 37 °C, pH of 7.0–7.5, 5% $CO_2$, 5% $O_2$ and 90% $N_2$. Medium changes were performed every 48 h and cultures were expanded to keep the parasitaemia at ~1%. Once expanded, each recombinant pool was divided into 16 0.5 ml aliquots while diluting to 1% parasitaemia. The aliquots were maintained in 48-well plates and treated with CQ (Supplementary Fig. 7). In total, we had 32 cultures: 2 pools × 4 CQ concentrations (0 (control), 50, 100 or 250 nM) × 2 drug duration time (48 h or 96 h) × 2 technical replicates. We define the day when drug was applied as day 0. After 2 days (48 h) of drug treatment, the infected RBCs were washed with phosphate-buffered saline solution twice to remove residual drug. For the plate assigned for 48 h CQ treatment (48-well plate 1), cultures were maintained in complete medium; and samples were collected at days 0, 4 and 7. For the plate assigned for 96 h CQ treatment (48-well plate 2), fresh CQ was added back to the culture medium and treated for another 48 h; and after a total of 96 h CQ treatment, drug was removed and samples were collected at days 0, 5 and 10. CQ was dissolved in $H_2O$ and diluted in incomplete medium (Gibco, Life Technologies). Culture medium was changed every 48 h. Parasitaemia was monitored using 20% Giemsa-stained slides, and cultures were diluted to 1% parasitaemia if the parasitaemia was higher than 1%. Approximately 15 μl packed RBCs was collected per sample.

**Library preparation and sequencing.** We prepared Illumina libraries and sequenced both parents and the 96 segregant pools collected. We extracted genomic DNA using the Qiagen DNA mini kit and quantified DNA with Quant-iT PicoGreen Assay (Invitrogen). For samples with <50 ng DNA obtained, we performed WGA[52]. WGA products were cleaned with KAPA Pure Beads (Roche Molecular Systems) at a 1:1 ratio. We prepared sequencing libraries using 50–100 ng DNA or WGA product using KAPA HyperPlus Kit following the instructions with three cycles of PCR. All libraries were sequenced at 150 bp pair-end using Illumina Novaseq S4 or Hiseq X sequencers, to obtain >100× genome coverage per sample.

**Mapping and genotyping.** We mapped the sequencing reads against the 3D7 reference genome (PlasmoDB version 46) using BWA mem (http://bio-bwa.sourceforge.net/), and deduplicated and trans-formatted the alignment files using picard tools v2.0.1 (http://broadinstitute.github.io/picard/). We recalibrated the base quality score based on a set of verified known variants[53] using BaseRecalibrator, and called variants through HaplotypeCaller. Both functions were from Genome Analysis Toolkit GATK v3.7 (https://software.broadinstitute.org/gatk/). Only variants located in the core genome regions (defined in ref. [53]) were called and used for further analysis.

**Genotype of parents.** We merged calls from the two parents using GenotypeGVCFs in GATK, and applied standard filtration to the raw variant dataset as described in ref. [54]. We recalibrated the variant quality scores and removed loci with variant quality score <1.

The final variants in VCF format were annotated using snpEff v4.3 (https://pcingola.github.io/SnpEff/) with 3D7 (PlasmoDB, release 46) as the reference. After filtration and annotation, we selected SNP loci that are distinct in the two parents and used those SNPs for further BSA.

**BSA.** We used statistical methods described in refs. 25,48,50 for BSA. The variant calls from segregant progeny pools were merged together. Additionally, SNP loci with coverage <30× were removed. We counted reads with genotypes of each parent and calculated allele frequencies. Allele frequencies of 3D7 were plotted across the genome, and outliers were removed following Hampel's rule[55] with a window size of 100 loci. We performed the BSA using the R package QTLseqr[56]. Extreme QTLs were defined as regions with $G' > 20$ (ref. 57). Once a QTL was detected, we calculated an approximate 95% confidence interval using Li's method[58] to localize causative genes.

### Progeny cloning and phenotyping
**Progeny cloning.** Individual progeny were cloned via limiting dilution at 0.3 cells per well from bulk cultures on day 10 after 96 h of control/250 nM CQ treatment. Individual wells with parasites were determined by qPCR (as previously described[49]) and expanded to larger cultures under standard culture conditions to obtain enough material for both cryopreservation and genome sequencing.

**Sequencing and genotyping.** Cloned progeny were sequenced and genotyped as described in the 'Genetic cross and BSA' section, with these modifications: (1) the cloned progeny were sequenced at 25× genome coverage; (2) SNP calls were removed if the coverage was more than three reads per sample.

**Cloned progeny analysis.** Unique recombinant progeny were identified from all cloned progeny using a previously described pipeline[49]. Non-clonal progeny were identified on the basis of the number and distribution of heterozygous SNP calls. Selfed progeny were identified as having greater than 90% sequence similarity to either parent. Unique recombinant progeny that were sampled multiple times were identified as clusters of individual clonal progeny with greater than 90% sequence similarity. We plotted frequencies of 3D7 alleles across the genome in progeny populations with and without CQ treatment. Heatmaps were generated to visualize inheritance patterns in individual unique recombinant progeny (Fig. 4a). We selected 16 unique recombinant progeny with different allele combination at chromosome 6 and chromosome 7 QTL regions for further CQ IC$_{50}$ vvalues measurement (Supplementary Fig. 5).

**Genome-wide linkage analysis on pfaat1 in cloned progeny.** Fisher's exact test was used to test for linkage between all inter-chromosomal pairs of loci across the set of 109 unique recombinant progeny. The distribution of the −log of the resulting *P* values were plotted in Fig. 4c, and the significance cut-off was calculated on the basis of a Bonferroni correction for the number of loci.

**IC$_{50}$ measurement for cloned progeny.** Cryopreserved stocks of 3D7, NHP4026, 3D7×NHP4026 progeny were thawed and grown in complete medium under standard culture conditions as described above. Cultures were kept below 3% parasitaemia with medium changes every 48 h. Parents and progeny IC$_{50}$ values were assessed via a standard 72 h SYBR Green 1 fluorescence assay[59]. Cultures were assessed daily for parasitaemia and stage. Cultures that were at least 70% ring were loaded into CQ dose–response assays of a series of two-fold drug dilutions across ten wells at 0.15% parasitaemia. Drug stocks (1 mg ml$^{-1}$) for CQ were prepared in H$_2$O as single-use aliquots and stored at −20 °C until use. Drug dilutions were prepared in incomplete medium. Biological replicates were conducted with at least two cycles of culturing between load dates. IC$_{50}$ values were calculated in GraphPad Prism 8 using a four-parameter curve from two technical replicates loaded per plate.

### CRISPR–Cas9 editing at *pfaat1* and parasite phenotyping
**CRISPR–Cas9 editing.** We designed plasmids for CRISPR–Cas9 editing as previously described[60]. The guide RNA (GAAATTAAATACATAAAAGA) was designed to target *pfaat1* in NHP4026. Edits (258L/313F, 258S/313S and 258S/313F, Fig. 5a) were introduced to NHP4026 through homology arm sequence with target and shield mutations. Binding-site control mutants were not generated, as *P. falciparum* lacks error-prone non-homologous end joining[61]. The parasites were transfected at ring stages with 100 μg plasmid DNA, and successful transfectants were selected by treatment with 24 nM WR99210 (gift from Jacobus Pharmaceuticals) for 6 days. The parasites were recovered after ~3 weeks. To determine whether recovered parasites contained the expected mutations, we amplified the target region (forward primer, AGTAC GGTACTTTTTATATGTACAGCT; reverse primer, TGCATTTGGTTGTT GAGAGAAGG) and confirmed the mutation with Sanger sequencing. We cloned parasites from successful transfection experiments: independent edited parasites (from different transfection experiments) were recovered for each *pfaat1* genotype. Edited parasites were genome sequenced to identify off-target edits elsewhere in the genome. We were not able to find any SNP or indel changes between the original NHP4026 and any CRISPR-edited parasites other than the target and shield mutations.

**IC$_{50}$ measurement for CRISPR–Cas9-edited parasites.** Parasite IC$_{50}$ values for CQ, amodiaquine, lumefantrine, mefloquine and quinine were measured for two to four clones per CRISPR–Cas9-modified line and for NHP4026 across multiple load dates as described above for cloned progeny, except that each plate included two NHP4026 technical replicates as controls. This replication of genotype within each load date allowed for detection of batch effects due to load date.

**Batch correction for IC$_{50}$ data.** Analysis of variance was used to account for batch effects and to test for differences in IC$_{50}$ values between all genotype groups and for each contrast between each CRISPR–Cas9-modified line and NHP4026 for each drug tested[62]. A linear model with load date (batch) and genotype as explanatory variables was utilized to generate batch-corrected IC$_{50}$ values for visualization of the impact of CRISPRCas9 modifications (Fig. 5b and Extended Data Fig. 7).

**Measurement of parasite fitness using competitive growth assays.** Parasites were synchronized to late-stage schizonts using a density gradient[63]. The top layer of late-stage schizonts was removed and washed twice with Roswell Park Memorial Institute (RPMI) medium. Synchronized cultures were suspended in 5 ml of complete medium at 5% haematocrit and allowed to re-invade overnight with gentle shaking. Parasitaemia and parasite stage were quantified using flow cytometry. Briefly, 80 μl of culture and an RBC control were stained with SYBR Green I and SYTO 61 and measured on a Guava easyCyte HT (Luminex). A total of 50,000 events were recorded to determine relative parasitaemia and stage. When 80% of parasites were in the ring stage, the head-to-head competition experiments were set up[64]. Competition assays were set up between CRISPR–Cas9-edited parasites and NHP4026 in a 1:1 ratio at a parasitaemia of 1% in a 96-well plate (200 μl per well) and maintained for 30 days. Each of the assays contained three biological replicates (three independent clones from different CRISPR–Cas9 editing experiments) and two technical replicates (two wells of culture). Every 2 days, the parasitaemia was assessed by microscopy using Giemsa-stained slides, samples were taken and stored at −80 °C and the cultures were diluted to 1% parasitaemia with fresh RBCs and medium. The proportion of parasites in each competition (Extended Data Fig. 10) was measured using a rhAmp SNP Assay (Integrated DNA Technologies) with primers targeting the CRISPR–Cas9-edited region in *pfaat1*.

**Selection coefficient.** We measured selection coefficient ($s$) by fitting a linear model between the natural log of the allele ratio (freq (allele-edited parasite)/freq (NHP4026)) and time (measured in 48 h parasite asexual cycles). The slope of the linear model provides a measure of the driving $s$ of each mutation[65]. To compare relative fitness of parasites carrying different *pfaat1* alleles, we normalized the fitness of NHP4026 to 1 and used slope + 1 to quantify the fitness of CRISPR–Cas9-edited parasites (Fig. 5c).

### Overexpression of *pf*AAT1 in yeast

To generate *pf*AAT1 expressing yeast, plasmid carrying the *pfaat1* coding sequence was transformed into yeast *Saccharomyces cerevisiae* (BY4743) as previously described[27]. The doubling time (h) was measured for strains carry empty vector, WT pfAAT1 or S258L mutant *pf*AAT1. We measured doubling time under two culture conditions: control or with 1 mM CQ. Three independent experiments were performed for each assay.

### *pf*AAT1 protein structure analysis

Three-dimensional homology models for *pf*AAT1 were predicted using AlphaFold[29,66] and I-TASSER[30,67,68] and analysed with PyMol software (v2.3.0; Schrödinger, LLC). At the primary sequence level, we used TOPCONS[32] to predict transmembrane helix topology for comparison. We plotted a cartoon version of the protein transmembrane topology based on the computationally predicted structures and membrane topology (Extended Data Fig. 9). Models were truncated to exclude amino-terminal residues 1–166, probably positioned outside of the membrane, because AlphaFold assigns low confidence to this N-terminal stretch. Furthermore, mutations of interest map only to transmembrane helices according to both 3D models and TOPCONS. I-TASSER generated models with topology similar to AlphaFold with the highest variations in AlphaFold low-confidence regions 1–166 and 475–516. The top five I-TASSER models superimpose on the Alpha-Fold model with a root mean square deviation range of 2.4–2.8 Å over 303–327 of 440 aligned residues using the PDBeFold Server (http://www.ebi.ac.uk/msd-srv/ssm). The four common SNPs (S258L, F313S, Q454E and K541L) overlay closely between the homology models. We evaluated the effect of different mutations on protein stability using the mutagenesis function in PyMol.

### Reporting summary

Further information on research design is available in the Nature Portfolio Reporting Summary linked to this article.

## Data availability

All raw sequencing data have been submitted to the National Center for Biotechnology Information Sequence Read Archive (SRA, https://www.ncbi.nlm.nih.gov/sra) or European Nucleotide Archive (ENA) with accession numbers available in Supplementary Tables 1 and 2. All other data are available in the main text or supplementary materials. Source data are provided with this paper.

## Code availability

The code used in analysis and data analysed are available at GitHub through the following links: https://github.com/emilyli0325/CQ.AAT1.git (X.L.), https://github.com/MPB-mrcg?tab=repositories (A.A.-N.) and https://github.com/kbuttons/CQ.AAT1.progeny.git (K.A.B.-S.).

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

## Acknowledgements

We thank past and present staff and visiting researchers at the MRC Unit in Gambia involved in studies from which samples were archived, including those under the earlier directorships of B. Greenwood and T. Corrah. Work at Texas Biomedical Research Institute was conducted in facilities constructed with support from Research Facilities Improvement Program grant C06 RR013556. Shoklo Malaria Research Unit is part of the Mahidol Oxford University Research Unit supported by the Wellcome Trust of Great Britain. We thank the staff of the Wellcome Trust Sanger Institute Sample Logistics, Sequencing and Informatics facilities for their contributions. The Structural Biology Core at The University of Texas Health Science Center at San Antonio is a part of Institutional Research Cores supported by the Office of the Vice President for Research, Greehey Children's Cancer Research Institute and the Mays Cancer Center Drug Discovery and Structural Biology Shared Resource (NIH grant P30 CA054174). Funding: NIH grant P01 12072248 (M.T.F.); NIH grant R37 AI048071 (T.J.C.A.); MRC Career Development Fellowship MC_EX_MR/K02440X/1 (A.A.-N.); MRC malaria program grant MC_EX_MR/J002364/1 (U.D.A.); European Research Council grant AdG-2011-294428 (D.C. and A.A.-N.); MRC grant MR/S009760/1 (D.C.); Wellcome Trust (206194 and 204911) and the Bill & Melinda Gates Foundation (OPP1204628 and INV-001927) (D.K.); Biotechnology and Biological Sciences Research Council grant BB/M022161/1 (S.V.A.); and Wellcome Trust (grant number 220221) (F.H.N.). For the purpose of open access, the author has applied a CC BY public copyright license to any Author Accepted Manuscript version arising from this submission.

## Author contributions

A.A.-N., K.A.B.-S., X.L., S.K. and K.V.B. contributed equally to this work. Conceptualization: M.T.F., T.J.C.A., I.H.C., A.M.V., A.A.-N. and D.J.C. Methodology: A.M.V., S.K., S.V.A. and A.B.T. Investigation: A.A.-N., U.D., D.K., D.J.C., R.A., R.D.P., K.A.B.-S., K.V.B., M.M.-W., M.T.H., L.A.C., H.D., J.K.L., A.R., E.D., S.K., D.A.S., S.M.T., M.F., A.B.T. and T.J.C.A. Analysis: X.L., K.A.B.-S. and A.A.-N. Visualization: X.L., K.A.B.-S. and A.A.-N. Funding acquisition: M.T.F., T.J.C.A., D.J.C., A.A.-N., D.K., S.V.A. and F.H.N. Project administration: M.F., D.K., A.A.-N., D.J.C. and S.V.A. Supervision: M.F., A.M.V., T.J.C.A., I.H.C., S.H.I.K., S.V.A., D.K. and D.J.C. Writing—original draft: K.V.B., T.J.C.A., K.A.B.-S. and X.L. Writing—review and editing: M.T.F., A.M.V., I.H.C., S.K., A.A.-N., D.J.C., D.K., U.D., R.D.P., S.H.I.K., F.H.N., A.B.T. and T.J.C.A.

## Competing interests

The authors declare no competing interests.

## Additional information

**Extended data** is available for this paper at https://doi.org/10.1038/s41564-023-01377-z.

**Correspondence and requests for materials** should be addressed to Ashley M. Vaughan, Michael T. Ferdig or Timothy J. C. Anderson.

Alfred Amambua-Ngwa[1,14], Katrina A. Button-Simons [2,14], Xue Li [3,14], Sudhir Kumar [4,14], Katelyn Vendrely Brenneman [2,14], Marco Ferrari[3], Lisa A. Checkley[2], Meseret T. Haile [4], Douglas A. Shoue[2], Marina McDew-White[3], Sarah M. Tindall[5], Ann Reyes[3], Elizabeth Delgado [3], Haley Dalhoff[2], James K. Larbalestier [2], Roberto Amato[6], Richard D. Pearson [6], Alexander B. Taylor [7], François H. Nosten[8,9], Umberto D'Alessandro[1], Dominic Kwiatkowski [6], Ian H. Cheeseman [10], Stefan H. I. Kappe [4,11,12], Simon V. Avery [5], David J. Conway [13], Ashley M. Vaughan [4,11] ✉, Michael T. Ferdig [2] ✉ & Timothy J. C. Anderson [3] ✉

[1]MRC Unit The Gambia at London School of Hygiene and Tropical Medicine, Banjul, The Gambia. [2]Eck Institute for Global Health, Department of Biological Sciences, University of Notre Dame, Notre Dame, IN, USA. [3]Disease Intervention and Prevention Program, Texas Biomedical Research Institute, San Antonio, TX, USA. [4]Center for Global Infectious Disease Research, Seattle Children's Research Institute, Seattle, WA, USA. [5]School of Life Sciences, University of Nottingham, Nottingham, UK. [6]Wellcome Sanger Institute, Hinxton, UK. [7]Department of Biochemistry & Structural Biology, University of Texas Health Science Center at San Antonio, Antonio, TX, USA. [8]Shoklo Malaria Research Unit, Mahidol-Oxford Tropical Medicine Research Unit, Mahidol University, Mae Sot, Thailand. [9]Centre for Tropical Medicine and Global Health, Nuffield Department of Medicine, University of Oxford, Oxford, UK. [10]Host Pathogen Interactions Program, Texas Biomedical Research Institute, San Antonio, TX, USA. [11]Department of Pediatrics, University of Washington, Seattle, WA, USA. [12]Department of Global Health, University of Washington, Seattle, WA, USA. [13]Department of Infection Biology, London School of Hygiene and Tropical Medicine, London, UK. [14]These authors contributed equally: Alfred Amambua-Ngwa, Katrina A. Button-Simons, Xue Li, Sudhir Kumar, Katelyn Vendrely Brenneman. ✉e-mail: Ashley.Vaughan@seattlechildrens.org; ferdig.1@nd.edu; tanderso@txbiomed.org

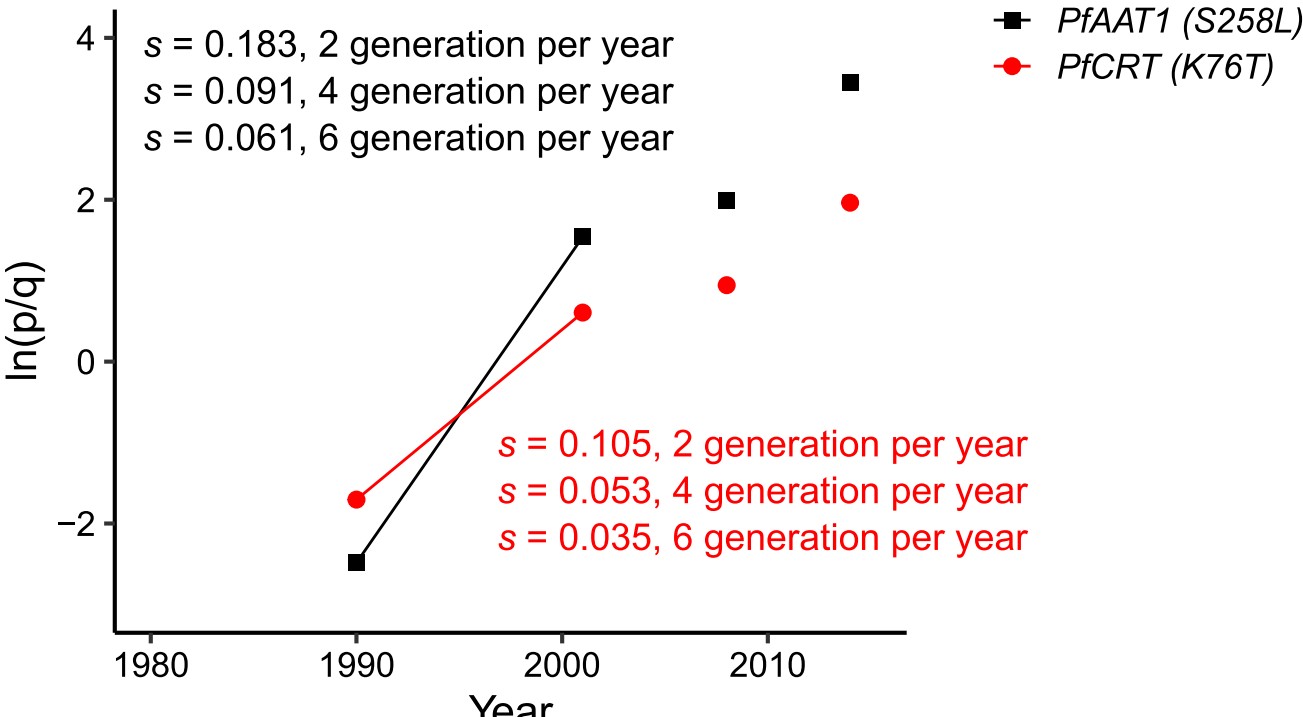

**Extended Data Fig. 1 | Estimation of selection coefficient (s) for *pfaat1 (S258L)* and *pfcrt (K76T)* alleles.** *p* is the frequency of mutant alleles (*pfaat1* S258L or *pfcrt* K76T) as indicated in Fig. 1a, and q (=1-p) is the inferred frequency of wild-type (3D7) alleles. The x-axis indicates parasite generations (labeled with sample collection year). We estimated selection coefficients (*s*) based on allele frequency from year 1990 and 2001, as CQ monotherapy was stopped in Gambia in 2004. *s* indicates the changes in relative growth per parasite generation (that is the duration of the complete lifecycle in both mosquito and human host). The calculation was based on estimates of 2, 4, or 6 generations per year.

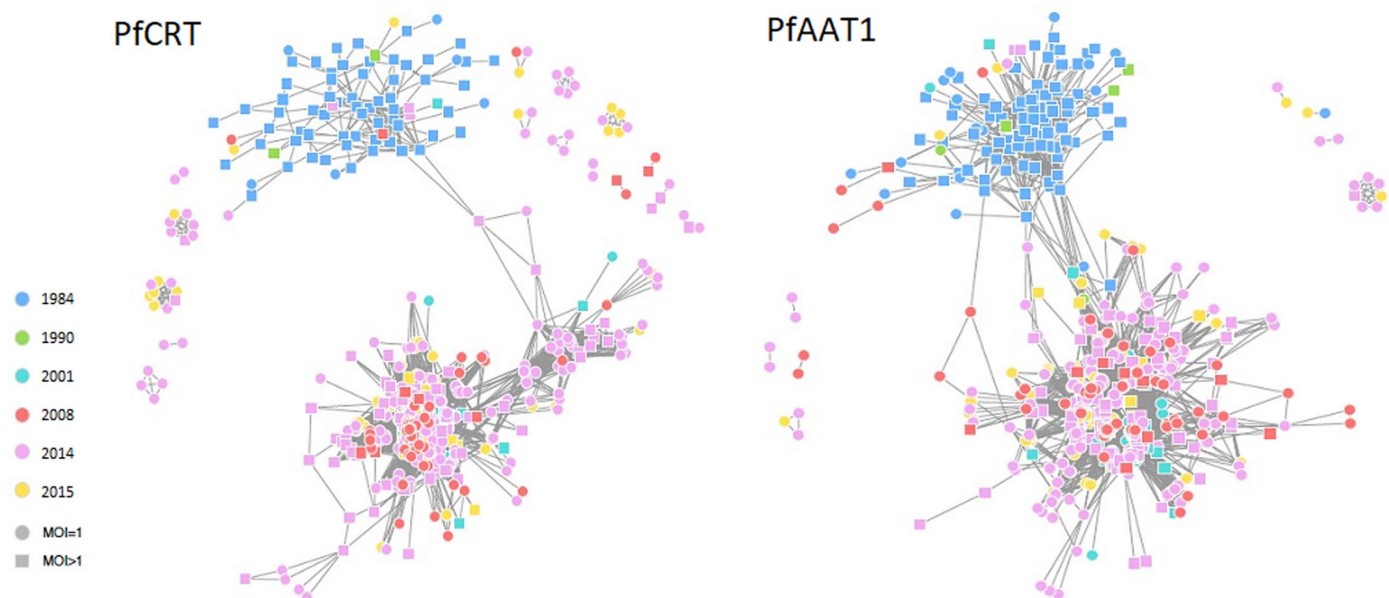

**Extended Data Fig. 2 | Haplotype structure at the *pfcrt* (left panel) and *pfaat1* (right panel) regions.** Haplotype relationships were based on Identity-by-Descent of genome segments encompassing 25 kb on either side of each gene (see methods). Haplotypes joined by lines indicate >90% IBD. Each point depicts an isolate with point colors representing the years from which they were sampled. Square points represent complex infections and circles represent monoclonals. MOI, multiplicity of infection.

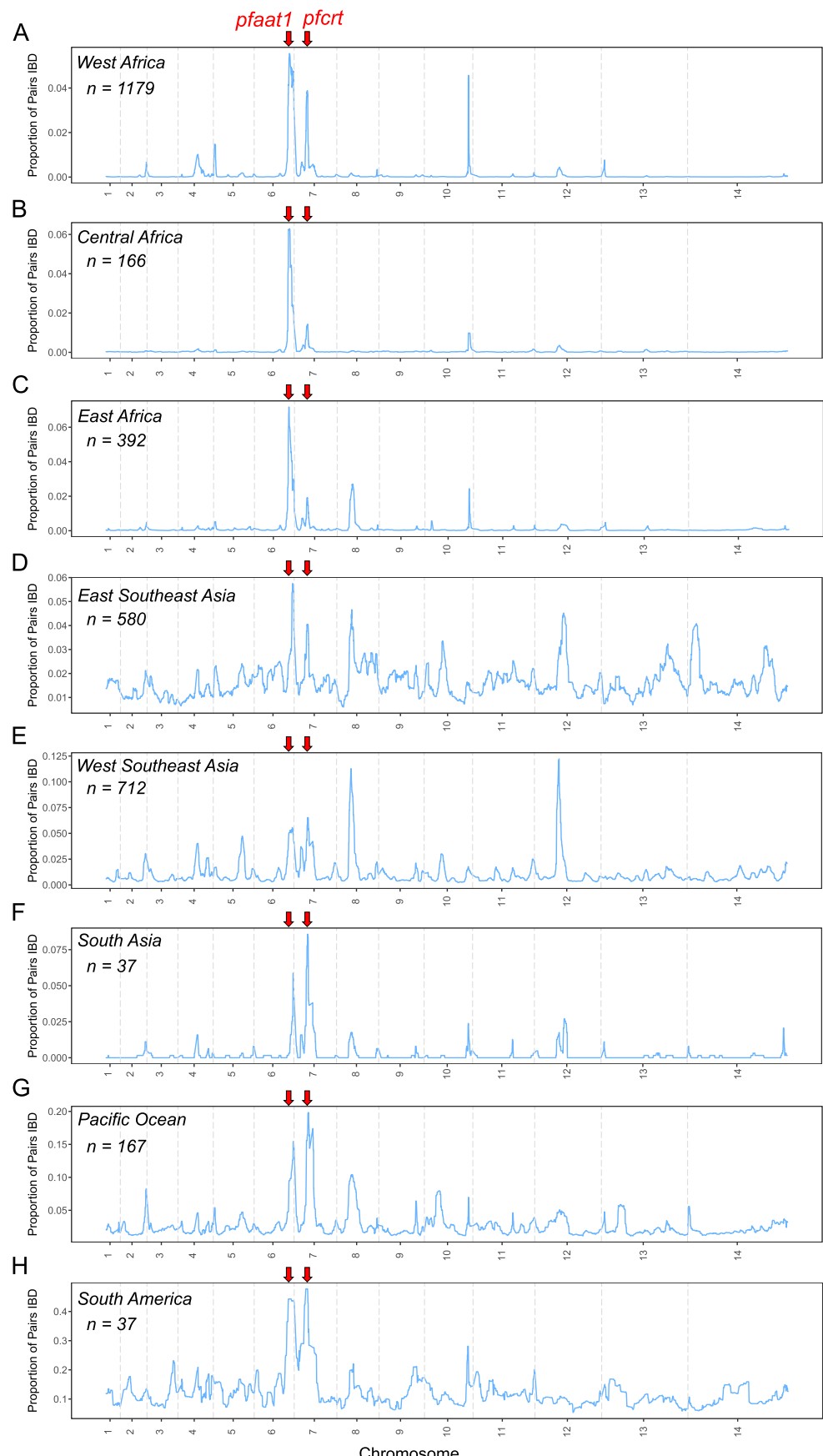

**Extended Data Fig. 3 | See next page for caption.**

**Extended Data Fig. 3 | The proportion of pairs identical by descent (IBD) within populations from global locations.** Panels A-H show the proportion of pairs IBD plotted across the genome for parasites from different geographical regions (marked in the top left of each panel). For samples where >90% of the genomes are IBD, only one representative sample with the highest genotype rate was selected and used for IBD analysis. Sample numbers are shown in each panel. Chromosome boundaries are indicated with grey dashed vertical lines.

The location of *pfaat1* and *pfcrt* are indicated with red arrows on top of each panel. The chr 10 peak (West Africa, A) contains *pfmspdbl2* associated with decreased sensitivity to halofantrine, mefloquine and lumefantrine[69]. The chr 8 peak (East Africa and Asia (C-H)) contains dihydropteroate synthase (sulfadoxine resistance)[70] and the chr 12 peak (D-F) contains GTP cyclohydrolase I, a compensatory locus for antifolate drugs[71]. See also analyses by Amambua-Ngwa et al.[17], Hendon et al.[18], and Carrasquilla et al.[19].

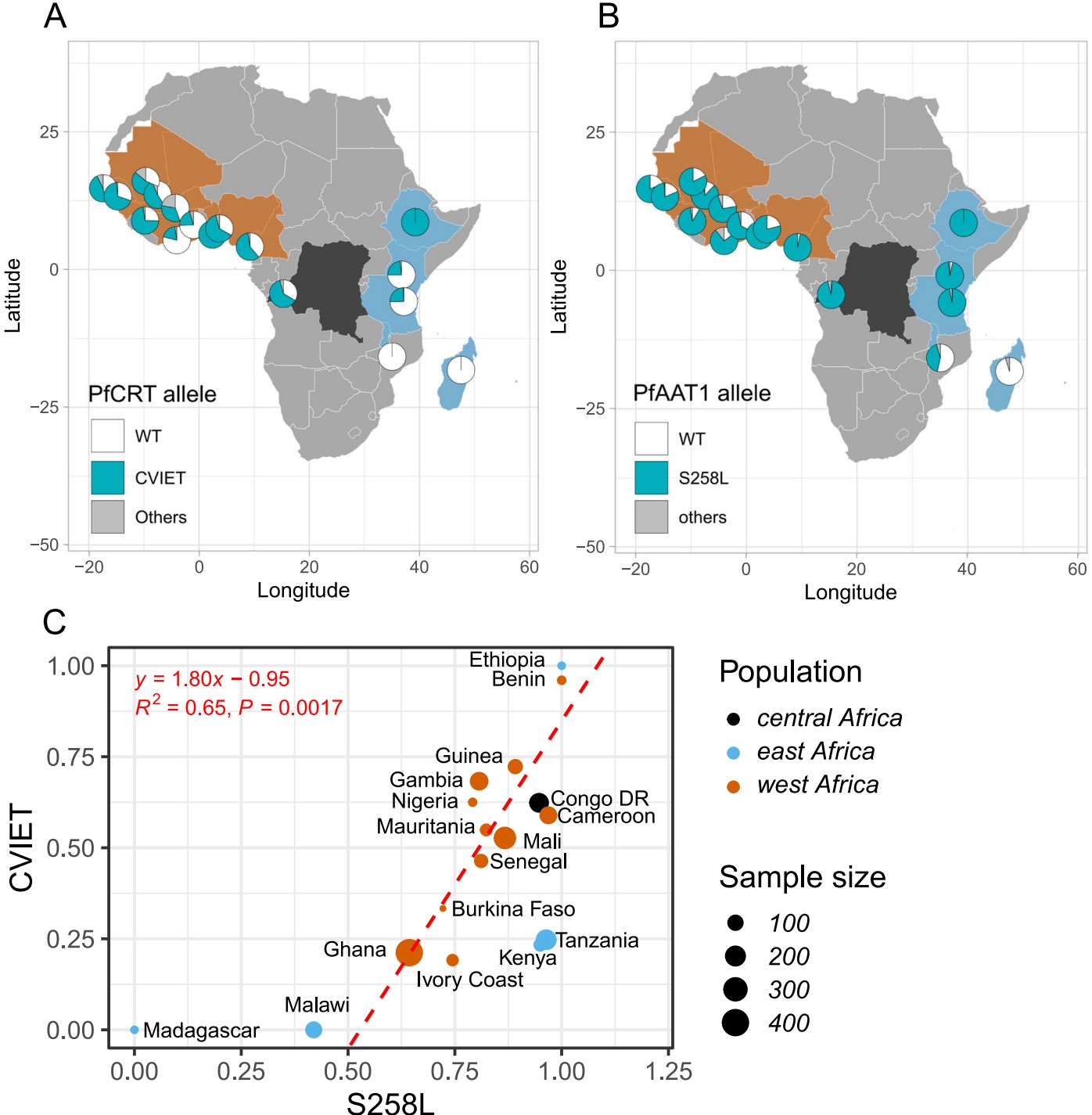

**Extended Data Fig. 4 | *pfcrt* and *pfaat1* allele frequency distributions and correlations in African countries.** A. *pfcrt* allele distribution in African countries. B. *pfaat1* allele distribution in African countries. C. Correlations in allele frequencies between *pfcrt* (CVIET) and *pfaat1* (S258L). Frequencies of the CVIET haplotype for amino acids 72-76 in *pfcrt* are significantly correlated with allele frequencies of *pfaat1* S258L in West Africa ($R^2 = 0.65$, $p = 0.0017$, red dashed line) or across all African populations ($R^2 = 0.44$, $p = 0.0021$). Point size indicates sample numbers, while color indicates sampling locations. We used t-test to establish if the Pearson's r statistic differs significantly from zero.

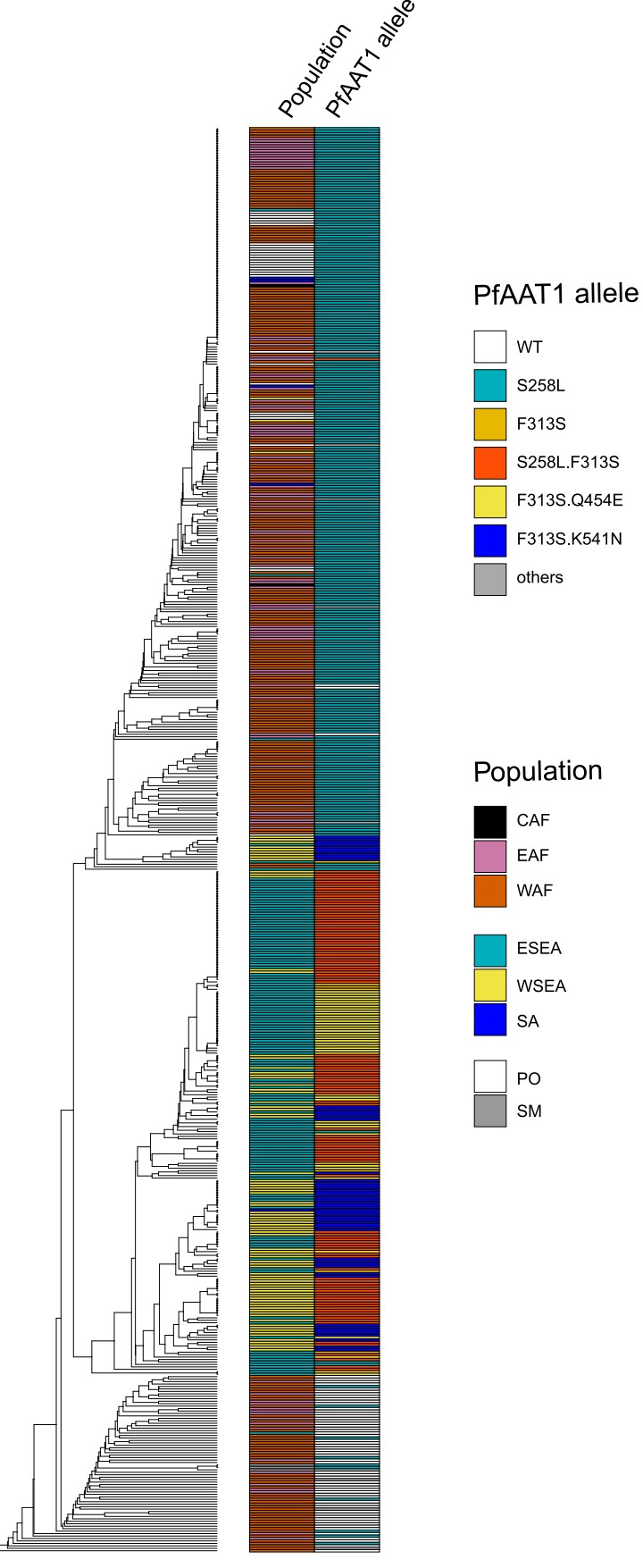

**Extended Data Fig. 5 | UPGMA tree showing the relationship of 581 haplotypes based on SNPs inside the 50 kb region surrounding *pfaat1*.** The tree was rooted with *Plasmodium reichenowi* (not shown in the tree). WAF: west Africa, EAF: east Africa, CAF: central Africa, SM: south America, ESEA: east Southeast (SE) Asia, SA: south Asia, WSEA: west SE Asia, PO: Pacific Ocean region.

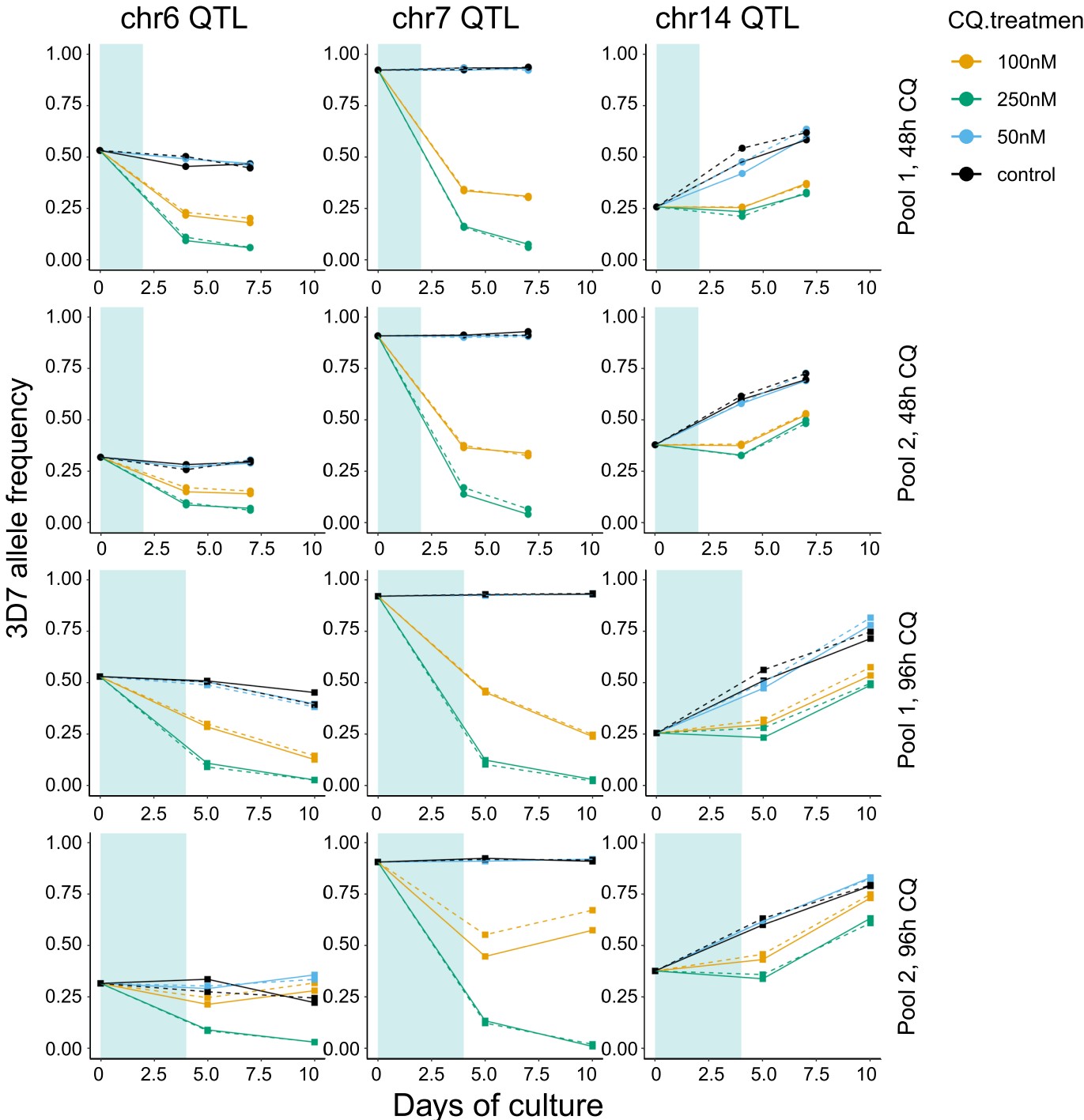

**Extended Data Fig. 6 | 3D7 allele frequency for QTLs at chr.6, chr.7 and chr.14.** CQ treatments were applied to the pools on day 0 at 0 (control), 50, 100, or 250 nM. CQ was removed on day 2 (48 hour treatment) or day 4 (96 hour treatment), as shaded with light blue. For 48 hour CQ treatments, samples were collected at day 0, 4, and 7; while for 96 hour CQ treatment, samples were collected at day 0, 5, and 10. Solid or dashed lines are results from different technical replicates. The 3D7 allele frequencies in CQ treated pools decrease at both chr.6 and chr.7 QTL regions. At the chr.14 QTL region, allele frequencies show no to little change following drug treatment, suggesting this QTL is unrelated to drug treatment.

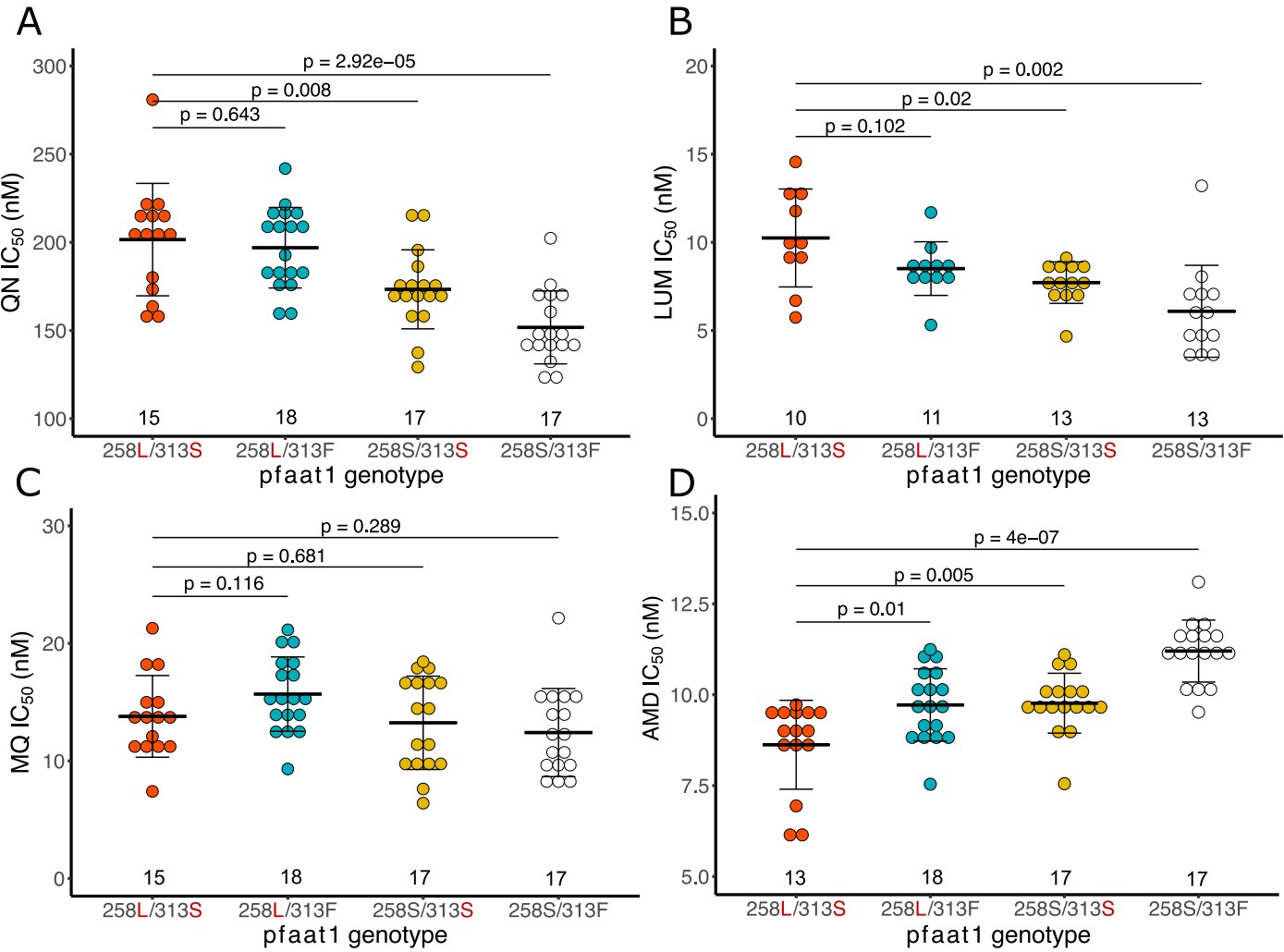

**Extended Data Fig. 7 | Impact of CRISPR/Cas9 substitutions on IC$_{50}$ of (A) quinine, (B) lumefantrine, (C) mefloquine and (D) amodiaquine.** CRISPR/Cas9 gene editing resulted in small differences in IC$_{50}$ for quinine (QN), lumefantrine (LUM) and Amodiaquine (AMD), but no significant changes for mefloquine (MQ). However, all IC$_{50}$s were below levels of clinical significance for these drugs (clinical thresholds: QN = 600 nM[72]; MQ = 30 nM[72], AMD = 60 nM[72]), or at the lower end of the *in vitro* range (0-150 nM) in the case of LUM[73]. The number of biological replicates is shown above the x-axis. *P* values indicate significance levels and are based on two-way ANOVA analysis. Data are presented as mean values +/− SEM.

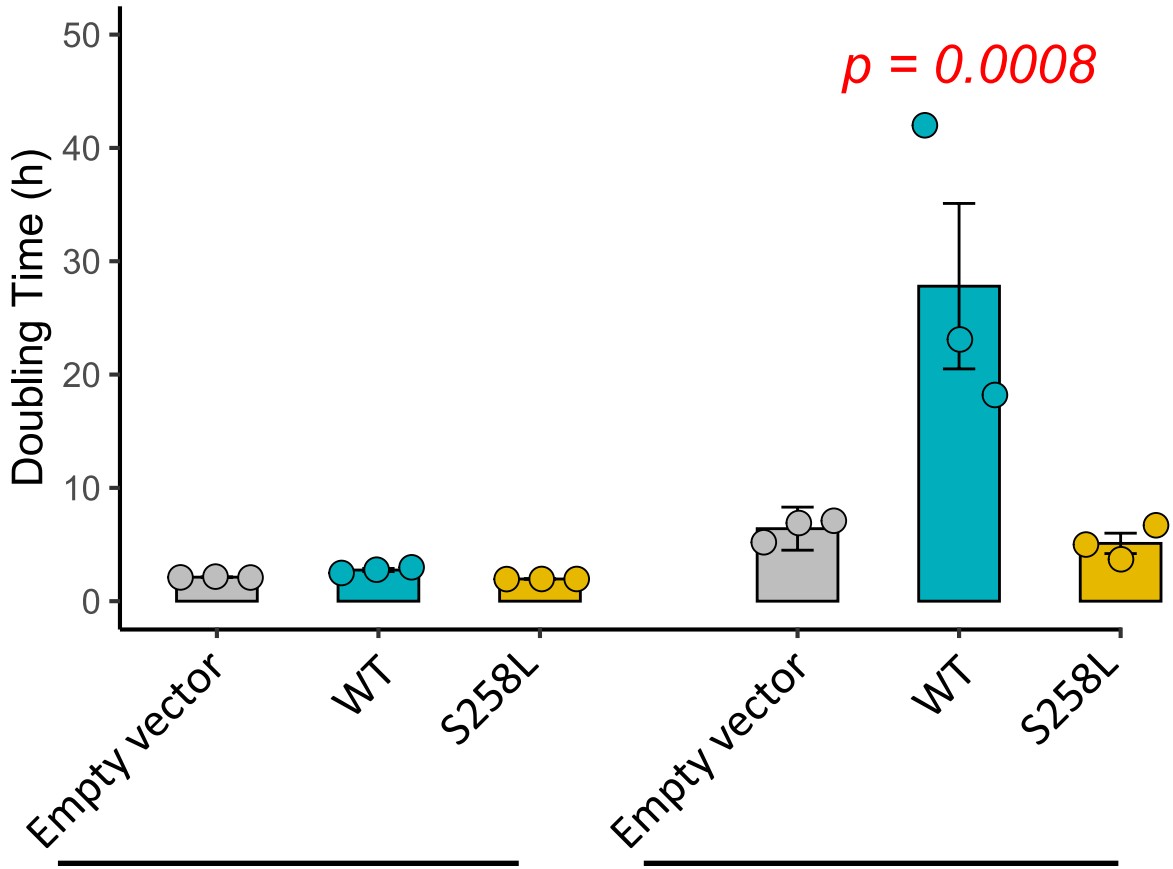

**Extended Data Fig. 8 | Introduction of S258L results in chloroquine resistance in yeast.** Yeast doubling time was calculated from the linear portion of exponential growth. Data was shown as means from 3 independent experiments ± SEM, and significance and was calculated according to multiple comparisons (with Turkey corrections) of two-way ANOVA. Growth of yeast cells expressing wild type *pfaat1* (WT) is severely impacted by CQ treatment (1 mM CQ through the experiments) but is recovered in yeast expressing *pfaat1* S258L. Published results demonstrate that AAT1 is expressed in the yeast cell membrane[27], while pfAAT1 localizes to the digestive vacuolar membrane[43], and may also be present in the plasma membrane[35].

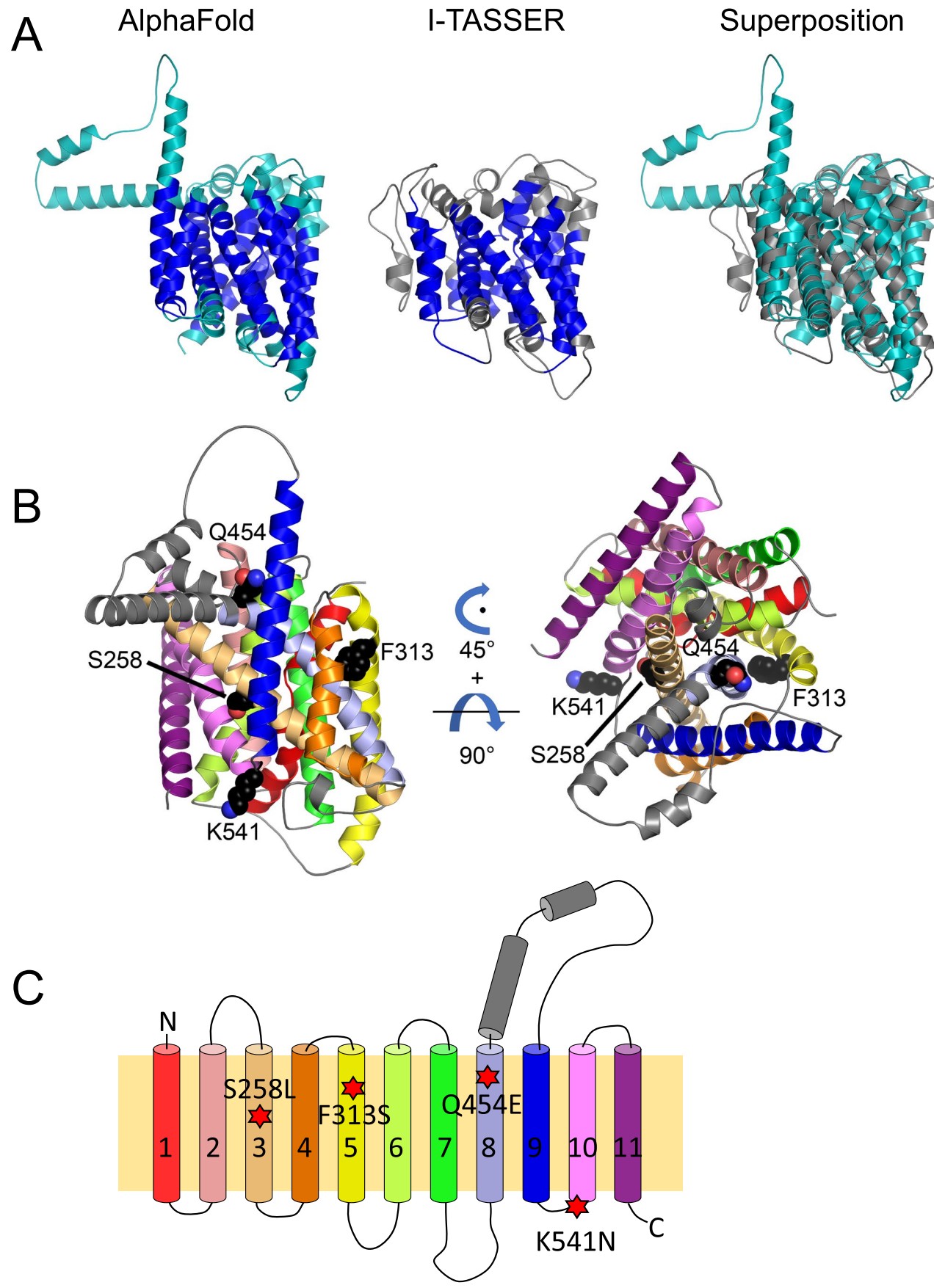

**Extended Data Fig. 9 | See next page for caption.**

**Extended Data Fig. 9 | Topology structure of pfAAT1 protein.** (A) AlphaFold model of PfAAT1 (left), representative I-TASSER model of PfAAT1 (center), structural superposition of the AlphaFold model (teal) and I-TASSER model (gray, right). TOPCONS transmembrane (TM) helix topology predictions are mapped onto the models in dark blue (left, center). AlphaFold and I-TASSER models align with a RMSD of 2.5 Å over 327 of 440 residues. Amino-terminal residues 1-166 were excluded from all models due to low confidence in structure prediction. (B) Detailed view of the mutations on the predicted PfAAT1 3-D structure using the AlphaFold model. The right view is related to the left by a 45° rotation about the axis looking down at the figure followed by a 90° rotation about the horizontal axis. The four SNPs shown as space-filling models are all arranged within a plane at one side of the model, perpendicular to the membrane. S258L (helix 3) and F313S (helix 5) are located opposite each other with helix 8 in between. Given the epistatic interactions between the PfAAT1 S258L and F313S SNPs evident from our functional analyses, the F313S substitution of the bulky, hydrophobic phenylalanine with the smaller, polar serine may compensate for a disruption in the transmembrane region that includes helices 3, 5, and 8 potentially allowing for partial restoration of predicted amino acid transport activity. Q454E is located on helix 8 near the TM surface and K541N is located in a loop connecting helix 9 and 10. (C) Topology of PfAAT1 inferred using 3D structure. There are eleven transmembrane (TM) helices. Three of the mutations are located at the TM helices, while K541N is located at a loop connecting helix 9 and 10. The color scheme matches the schematic in Panel B. The blue triangle indicates amino-terminal residues 1-166 that were excluded from structure prediction.

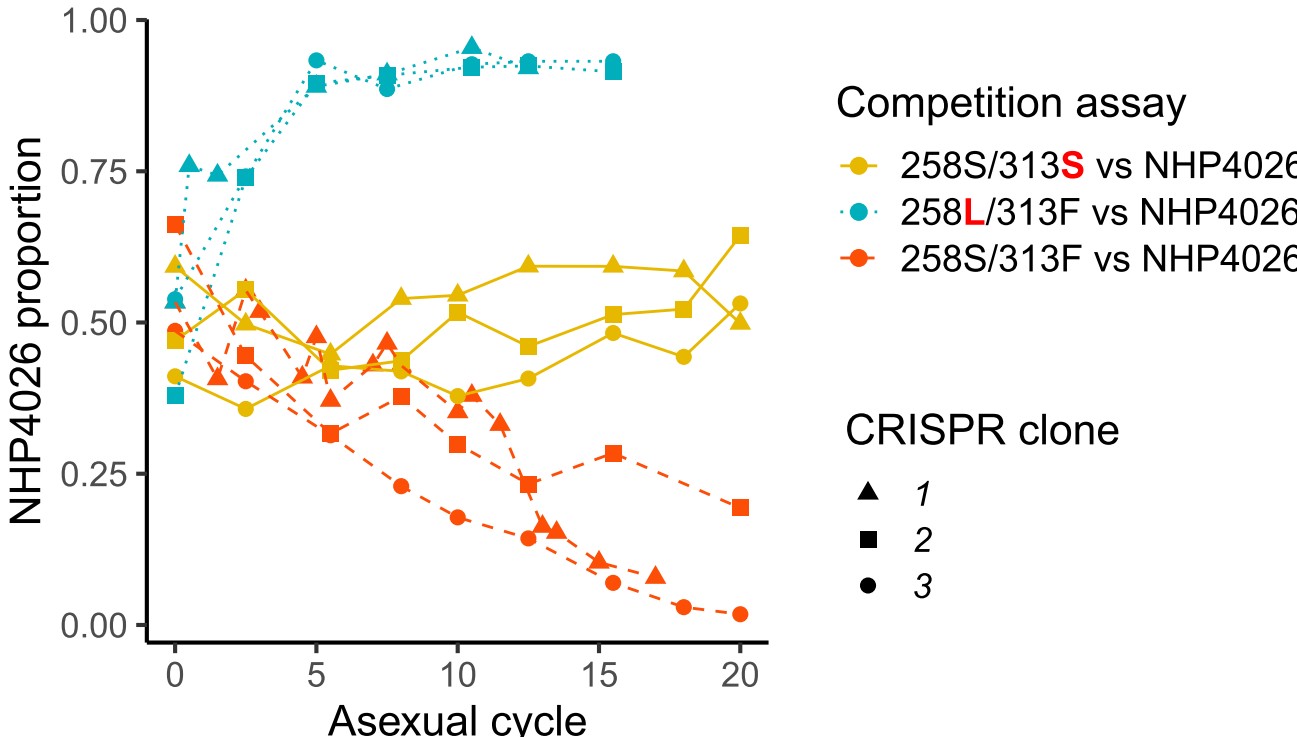

**Extended Data Fig. 10 | Allele frequency changes in head-to-head competition experiments between different CRISPR/Cas9 edited parasites and NHP4026.** We used three independent CRISPR/Cas9 edited clones for each genotype, and two technical replicates for each competition experiment (average values plotted).

# Reporting Summary

## Statistics

For all statistical analyses, confirm that the following items are present in the figure legend, table legend, main text, or Methods section.

| n/a | Confirmed | |
|---|---|---|
| ☐ | ☒ | The exact sample size ($n$) for each experimental group/condition, given as a discrete number and unit of measurement |
| ☐ | ☒ | A statement on whether measurements were taken from distinct samples or whether the same sample was measured repeatedly |
| ☐ | ☒ | The statistical test(s) used AND whether they are one- or two-sided<br>*Only common tests should be described solely by name; describe more complex techniques in the Methods section.* |
| ☐ | ☒ | A description of all covariates tested |
| ☐ | ☒ | A description of any assumptions or corrections, such as tests of normality and adjustment for multiple comparisons |
| ☐ | ☒ | A full description of the statistical parameters including central tendency (e.g. means) or other basic estimates (e.g. regression coefficient) AND variation (e.g. standard deviation) or associated estimates of uncertainty (e.g. confidence intervals) |
| ☐ | ☒ | For null hypothesis testing, the test statistic (e.g. $F$, $t$, $r$) with confidence intervals, effect sizes, degrees of freedom and $P$ value noted<br>*Give P values as exact values whenever suitable.* |
| ☒ | ☐ | For Bayesian analysis, information on the choice of priors and Markov chain Monte Carlo settings |
| ☒ | ☐ | For hierarchical and complex designs, identification of the appropriate level for tests and full reporting of outcomes |
| ☒ | ☐ | Estimates of effect sizes (e.g. Cohen's $d$, Pearson's $r$), indicating how they were calculated |

*Our web collection on statistics for biologists contains articles on many of the points above.*

## Software and code

Policy information about availability of computer code

| | |
|---|---|
| Data collection | All raw sequencing data have been submitted to the NCBI Sequence Read Archive (SRA, https://www.ncbi.nlm.nih.gov/sra) or European Nucleotide Archive (ENA) with accession numbers available in Table S1. All other data are available in the main text or supplementary materials. |
| Data analysis | The code used in analysis and data analyzed are available at GitHub through the following links: https://github.com/emilyli0325/CQ.AAT1.git (XL), https://github.com/MPB-mrcg?tab=repositories (AAN), https://github.com/kbuttons/CQ.AAT1.progeny.git (KBS). |

For manuscripts utilizing custom algorithms or software that are central to the research but not yet described in published literature, software must be made available to editors and reviewers. We strongly encourage code deposition in a community repository (e.g. GitHub). See the Nature Portfolio guidelines for submitting code & software for further information.

## Data

Policy information about availability of data

All manuscripts must include a data availability statement. This statement should provide the following information, where applicable:
- Accession codes, unique identifiers, or web links for publicly available datasets
- A description of any restrictions on data availability
- For clinical datasets or third party data, please ensure that the statement adheres to our policy

> A data availability statement is included in the manuscript
> All raw sequencing data have been submitted to the NCBI Sequence Read Archive (SRA, https://www.ncbi.nlm.nih.gov/sra) or European Nucleotide Archive (ENA) with accession numbers available in Supplementary Table 1.

## Human research participants

Policy information about studies involving human research participants and Sex and Gender in Research.

| | |
|---|---|
| Reporting on sex and gender | *Use the terms sex (biological attribute) and gender (shaped by social and cultural circumstances) carefully in order to avoid confusing both terms. Indicate if findings apply to only one sex or gender; describe whether sex and gender were considered in study design whether sex and/or gender was determined based on self-reporting or assigned and methods used. Provide in the source data disaggregated sex and gender data where this information has been collected, and consent has been obtained for sharing of individual-level data; provide overall numbers in this Reporting Summary. Please state if this information has not been collected. Report sex- and gender-based analyses where performed, justify reasons for lack of sex- and gender-based analysis.* |
| Population characteristics | *Describe the covariate-relevant population characteristics of the human research participants (e.g. age, genotypic information, past and current diagnosis and treatment categories). If you filled out the behavioural & social sciences study design questions and have nothing to add here, write "See above."* |
| Recruitment | *Describe how participants were recruited. Outline any potential self-selection bias or other biases that may be present and how these are likely to impact results.* |
| Ethics oversight | *Identify the organization(s) that approved the study protocol.* |

Note that full information on the approval of the study protocol must also be provided in the manuscript.

# Field-specific reporting

Please select the one below that is the best fit for your research. If you are not sure, read the appropriate sections before making your selection.

☒ Life sciences ☐ Behavioural & social sciences ☐ Ecological, evolutionary & environmental sciences

For a reference copy of the document with all sections, see nature.com/documents/nr-reporting-summary-flat.pdf

# Life sciences study design

All studies must disclose on these points even when the disclosure is negative.

| | |
|---|---|
| Sample size | Population analysis - we used malaria samples available from longitudinal sampling of Plasmodium falciparum in the Gambia (n=315). For the global analysis we also analyzed available archived sequence data from the public databases and published manuscripts (N=4051). Power calculations were not conducted, but the numbers far exceed those needed to identify temporal or spatial trends |
| Data exclusions | No data were excluded from these analyses. |
| Replication | Replication is detailed in the manuscript. (i) We conducted two independent biological replications of genetic crosses for the linkage analysis. (ii) We used three independent CRISPR/Cas9 gene edits for each pfAAT1 haplotype generated, to avoid spurious conclusions from off target effects. (iii) We conducted multiple (10-28) independent measures of IC50 to improve robustnesss of drug response measures. (iv) We conducted 3 independent replicates of each each pairwise competition experiment when quantifying parasite fitness. Replicates provided consistent results: this is documented in the manuscript |
| Randomization | The bulk segregant analysis used parasite progeny that were randomly assigned to treatment or control groups - this was done by aliquoting parasite cultures for the two treatment. No randomization was conducted for the population genomic and CRISPR analyses. |
| Blinding | The outcome of competition experiments and Bulk segregant analyses was conducted blind, and the results were determined by genome sequencing or genotyping experiments. |

# Reporting for specific materials, systems and methods

We require information from authors about some types of materials, experimental systems and methods used in many studies. Here, indicate whether each material, system or method listed is relevant to your study. If you are not sure if a list item applies to your research, read the appropriate section before selecting a response.

## Materials & experimental systems

| n/a | Involved in the study |
|---|---|
| ☒ | ☐ Antibodies |
| ☐ | ☒ Eukaryotic cell lines |
| ☒ | ☐ Palaeontology and archaeology |
| ☐ | ☒ Animals and other organisms |
| ☒ | ☐ Clinical data |
| ☒ | ☐ Dual use research of concern |

## Methods

| n/a | Involved in the study |
|---|---|
| ☒ | ☐ ChIP-seq |
| ☒ | ☐ Flow cytometry |
| ☒ | ☐ MRI-based neuroimaging |

# Eukaryotic cell lines

Policy information about cell lines and Sex and Gender in Research

| | |
|---|---|
| Cell line source(s) | No cell lines were used that were derived from human or vertebrate models |
| Authentication | We verified the identity of Plasmodium cultures using illumina short read sequencing |
| Mycoplasma contamination | The cell lines were not tested for mycoplasma contamination |
| Commonly misidentified lines (See ICLAC register) | None |

# Animals and other research organisms

Policy information about studies involving animals; ARRIVE guidelines recommended for reporting animal research, and Sex and Gender in Research

| | |
|---|---|
| Laboratory animals | FRG NOD Huhep mice were purchased from Yecuris Corporation. |
| Wild animals | *Provide details on animals observed in or captured in the field; report species and age where possible. Describe how animals were caught and transported and what happened to captive animals after the study (if killed, explain why and describe method; if released, say where and when) OR state that the study did not involve wild animals.* |
| Reporting on sex | Male and female FRG NOD huHep mice18 with human chimeric livers were used in the experiments. Both male and female mice support liver stage P. falciparum development and can be used for these studies |
| Field-collected samples | *For laboratory work with field-collected samples, describe all relevant parameters such as housing, maintenance, temperature, photoperiod and end-of-experiment protocol OR state that the study did not involve samples collected from the field.* |
| Ethics oversight | The study was performed in accordance with the Guide for the Care and Use of Laboratory Animals of the National Institutes of Health (NIH), USA. The Seattle Children's Research Institute (SCRI) has an Assurance from the Public Health Service (PHS) through the Office of Laboratory Animal Welfare (OLAW) for work approved by its Institutional Animal Care and Use Committee (IACUC). All of the work carried out in this study was specifically reviewed and approved by the SCRI IACUC. |

Note that full information on the approval of the study protocol must also be provided in the manuscript.

