## [Peer Review File · Nature Microbiology]

Peer Review Information

Journal: Nature Microbiology

Manuscript Title: Chloroquine resistance evolution in Plasmodium falciparum is mediated by the putative amino acid transporter AAT1

Corresponding author name(s): Dr Timothy Anderson

This manuscript has been previously reviewed at another journal. This document only contains reviewer comments, rebuttal and decision letters for versions considered at Nature Microbiology. Mentions of prior referee reports have been redacted

Reviewer Comments & Decisions:

Decision Letter, initial version:

Message: 15th September 2022

Dear Dr Anderson,

Thank you for your patience while your manuscript "The amino acid transporter (pfaat1) plays a pivotal role in chloroquine resistance evolution in malaria parasites" was under peer-review at Nature Microbiology. It has now been seen by 4 referees, whose expertise and comments you will find at the end of this email. Although they find your work of some potential interest, they have raised a number of concerns that will need to be addressed before we can consider publication of the work in Nature Microbiology.

In particular, the referees ask for additional experiments to better support the conclusions of the study, and for better data presentation and additional discussion of the findings. Specifically, referee #1 feels that the data in Figure 4A should be better presented, and that the structure modeling part of the study is not very informative. Referee #2 feels that there are some inconsistencies in the gene editing vs the genetic cross progeny data and is not convinced that the current data shows that AAT1 is a contributor to chloroquine resistance (rather than a secondary factor that is offsetting fitness costs associated with mutant CRT). The referee also feels the manuscript currently doesn't reach the level of robustness required for Nature Microbiology. Furthermore, referee #2 identifies a number of issues that will need to be addressed. The referee also says that "it would be important to test the impact of the AAT1 Southeast Asian mutations S258L+F313S on CQ IC50 and fitness in 3D7 as well as other African parasites from The Gambia and Mali. The manuscript would be strengthened by adding data from AAT1 CRISPR/Cas9 editing on African parasite lines." Referee #2 also states that "The yeast data are not particularly convincing and I personally feel they are overinterpreted." Referee #3 has a number of comments on the presentation of results, evidence behind some claims, and missing discussion points. This referee also states that "The claim that the 258L allele has an independent origin in Asia and Africa should be better supported.". The referee also mentions that there needs to be better description for

2the longitudinal samples. Referee #4 states that "The different effects of *aat1* mutations (S258L and F313S) on CQR and fitness were done with the SE Asian parasite NHP4026. The results would deserve evaluations in an African parasite line to prove that the observed effects were not due to the Asian parasite's genetic background." The referee also says that "it would be very useful if the mechanistic experiments performed in yeast for the malaria parasite could be replicated in malaria parasites using their CRISPR mutants."

Editorially, we will require all referee concerns to be addressed in full and we expect a substantial revision. Should further experimental data allow you to address these criticisms, we would be happy to look at a revised manuscript.

Please include a data availability statement as a separate section after Methods but before references, under the heading "Data Availability". This section should inform readers about the availability of the data used to support the conclusions of your study. This information includes accession codes to public repositories (data banks for protein, DNA or RNA sequences, microarray, proteomics data etc...), references to source data published alongside the paper, unique identifiers such as URLs to data repository entries, or data set DOIs, and any other statement about data availability. At a minimum, you should include the following statement: "The data that support the findings of this study are available from the corresponding author upon request", mentioning any restrictions on availability. If DOIs are provided, we also strongly encourage including these in the Reference list (authors, title, publisher (repository name), identifier, year). For more guidance on how to write this section please see: <http://www.nature.com/authors/policies/data/data-availability-statements-data-citations.pdf>

* Include a “Response to referees” document detailing, point-by-point, how you addressed each referee comment. If no action was taken to address a point, you must provide a compelling argument. This response will be sent back to the referees along with the revised manuscript.

* If you have not done so already we suggest that you begin to revise your manuscript so that it conforms to our Article format instructions at <http://www.nature.com/nmicrobiol/info/final-submission>. Refer also to any guidelines provided in this letter.

[redacted]

Note: This url links to your confidential homepage and associated information about manuscripts you may have submitted or be reviewing for us. If you wish to forward this e-mail to co-authors, please delete this link to your homepage first.

3Nature Microbiology is committed to improving transparency in authorship. As part of our efforts in this direction, we are now requesting that all authors identified as 'corresponding author' on published papers create and link their Open Researcher and Contributor Identifier (ORCID) with their account on the Manuscript Tracking System (MTS), prior to acceptance. This applies to primary research papers only. ORCID helps the scientific community achieve unambiguous attribution of all scholarly contributions. You can create and link your ORCID from the home page of the MTS by clicking on 'Modify my Springer Nature account'. For more information please visit www.springernature.com/orcid.

If you wish to submit a suitably revised manuscript we would hope to receive it within 6 months. If you cannot send it within this time, please let us know. We will be happy to consider your revision, even if a similar study has been accepted for publication at Nature Microbiology or published elsewhere (up to a maximum of 6 months).

Yours sincerely,
[redacted]

Reviewer Expertise:

Referee #1: Plasmodium biology, drug resistance
Referee #2: Plasmodium biology, drug resistance
Referee #3: Parasite biology, malaria, genomics
Referee #4: Plasmodium biology, antimalarial drugs

Reviewer Comments:

Reviewer #1 (Remarks to the Author):

Summary and background

Despite the way Genetics is routinely presented in high school and undergraduate classes, most genetic/genomic changes that have a recognizable phenotype involve more than a single gene. This manuscript describes a coordinated series of experiments performed by researchers in several laboratory groups in the US, the UK, the Gambia and Thailand. This paper has capitalized on the availability of samples from the Gambia collected over a crucial period during which the dominant antimalarial at the time, chloroquine (CQ), was intensively used in this small country and then the regimen was changed to a combination of CQ + sulfadoxine-pyrimethamine and then an ACT, lumefantrine + artemether. The main question posed is- are changes in a single well known determinant of CQ resistance (pfcr1) accompanied by any other genomic changes that parallel those in prevalence of pfcr1 in the Gambian parasites? If so, does this interaction affect the parasite response to CQ treatment?

The answer is definitely yes! The complex sets of experimental approaches used have returned a detailed picture of the functional interactions between pfcr1 and pfaat1 in malaria parasites. The group has identified the epistatic interactions of these two genes and probed the functional consequences for the malaria parasites that carry them. The answer reminds researchers that a single allele of a gene is rarely the only determinant of a complex phenotype even in a unicellular organism.

The data are assembled by combining some “older” approaches- whole genome analysis of the Gambian parasite series over time by the MalariaGEN project. These longitudinal data were then compared by several newer approaches. First, a relatively new approach- a series of Pf genetic crosses that differed in pfcr1 was completed in mouse strains altered to express human liver and red cells. These made it possible to identify genetic changes anywhere in the Pf genomes that paralleled the changes in pfcr1 in the Gambian set over time.

Progressively smaller sections of the Pf genome that did show this parallel response to antimalarial exposure were identified, and finally narrowed to a section on chromosome 6 that contained another gene, pfaat1. This gene was known to encode a protein with a similar cellular location in the parasite and also a similar function transporting amino acids out of the digestive vacuole. Most important, changes in prevalence of particular mutations in pfaat1 increased in prevalence in the Gambian data set over time. The particular changes in these two genes appeared to be responding to similar selection pressures - pfcr1 with the key mutation K76T and distinct alleles of pfaat1 that carried S258L.

The longitudinal study was then extended to parasites that evolved over a similar time frame and several changes of antimalarial treatment in Shoklo, Thailand on the Thai-Myanmar border. These parasites all carried the same pfcr1 allele that encoded a 5 amino acid bearing pfcr1 allele CVIET, amino

acid 72-76. They observed similar evolution of the pfcr1 and pfaa1 haplotype pair in these SE Asian parasites, as well.

Then, lab based studies were initiated to explore whether this parallel temporal pattern had functional consequences for the phenotype in parasite cultures with pfcr1 that expressed the CVIET allele and various alleles of pfaat1. Here, the capacity of CRISPR/cas9 -based editing of cloned parasites was used to create a series of clones with various combinations (haplotypes) of pfcr1 and pfaat1 alleles. Particular responses to CQ treatment with respect to susceptibility to CQ in vitro, and competitions between pairs of clones with various combinations of the pfaat1 and pfcr1 measuring sustained growth in vitro as a surrogate for fitness. Here, they hit the jackpot: parasites that carried the combinations of pfcr1 K76T and aat1 S258L had differences in susceptibility of Pf parasites to CQ and the success of growth in the presence of CQ, in vitro.

As an extra bonus, they studied *Saccharomyces cerevisiae* (baker's yeast) that expressed the pfaat1 gene bearing the S258L protein and measured susceptibility to CQ and other quinolone drugs, and amino acid transport in the presence of CQ. They inferred that even in yeast, the AAT1 protein was involved in amino acid and CQ transport. In addition, the authors used now widely available models to try to explore changes in the protein structure of various single and double mutant alleles of the AAT1 protein predicted by the models.

Based on these predictions, they considered how the acquisition of an F313S mutation by pfaat1 S258L might affect the protein structure and be manifest in changes in transport functions that they observed.

Assessment

This paper is remarkable in several ways. The creative use of already available genomic information to ask a new question is a particular starting point. The process of winnowing the chunk of DNA that contained the region that increased in prevalence in the Gambian parasites required a complex series of genetic crosses in humanized mice. This technology was applied by the consortium that had developed the approach, and convincingly identified the aat1 gene as the responding gene. As important, the consortium grew to include researchers in SE Asia and those with expertise in gene editing and with testing the in vitro responses to CQ growth. Despite the large number of groups involved, the manuscript is clearly written and the logic for the various approaches used to identify the pfaat1 gene is explained. However, the addition of a flow chart showing the logic of each step, and the outcome

followed by the next step- Bulk Segregant Analysis followed by detailed crosses of specific clonal haplotypes for example might be a helpful guide for readers.

The figures are clearly designed and informative with the exception of Figure 4A. The lists at the very top apply to the vertical bars at the far left? Is the first vertical bar the 4 “contests” that were run, the second bar shows comparisons between controls and CQ selected. The red and black are heat maps to show alleles from 3D7 or NHP across the genomes. I’m not sure that the composite heat map conveys this information at all clearly. Some more discussion on how to present this information is needed, or consider omitting 4A or presenting the proportions of each haplotype in a table.

Specifics

The methods appear to be reasonably detailed, and provide specific references for additional information. However, the genetic crosses in particular are not easily performed by groups without this experience, or access to the specialized humanized mice so collaborating with these authors is the most sensible pathway for researchers who might want to use similar strategies.

I understand the motivation to use structure modeling to try to understand how the particular interactions of the amino acid changes in the AAT1 protein carrying different combinations of changes at the 258 and 314 codons. However, the conclusions are not particularly informative and add little to this very comprehensive paper. This line of thought would best explored in detail with collaborators with structural expertise. In the context of the current paper, this section is better omitted.

It has been known for a long time that selection of CQ resistant parasites was first observed in Asia and later that exact allele CVIET (codons 72-76) was brought to South Africa then was spread by migration of infected people all over the African continent. However, in South America a different extended sequence SVMNT (aa 72-76) evolved and predominates. In South American parasites, CQ resistance is conferred by this genotype and it does not seem to cause any decrease in fitness in parasites that carry it. Would it be useful to mention this in the discussion? Perhaps this large group is already considering how that South American version of CQ resistance might behave differently? It’s at least worth a mention.

Reviewer #2 (Remarks to the Author):

This submission by Amambua-Ngwa et al presents some interesting evidence supporting mutations in AAT1 as a contributor to the evolution of chloroquine resistance (CQR) in *Plasmodium falciparum* (Pf). The report combines analysis of field samples collected over time from The Gambia, and the distribution of AAT1 haplotypes globally. With the results of a genetic cross between a Thai CQ-resistant (NHP4026) and an African CQ-sensitive (3D7) line. These data collectively provide evidence supporting AAT1 as a secondary determinant, acting epistatically with mutant forms of the primary CQ efflux transporter CRT.

Overall, this is an interesting and impressive body of work that I think will be of broad interest. Some aspects, however, require further review. There are several gaps of knowledge in this work that need to be addressed to better substantiate the broader conclusions of this study. The study does not elucidate the temporal history of whether AAT1 or CRT mutations first emerged in the field and which AAT1 mutations among the four are critical for helping compensate for mutant CRT parasites' poor fitness. A large part of this study is trying to link and extrapolate findings between datasets that are not entirely conclusive.

Major comments:

1. The authors present an identity-by-descent (IBD) analysis to demonstrate the strong co-selection of mutations in AAT1 and the CRT CQR marker mutation K76T in a set of 321 isolates from The Gambia, sampled from 1984 to 2014. It is not clear why this analysis has been limited to only samples from The Gambia when the MalariaGen Pf6 dataset contains more than 3,000 samples from various regions in Africa (note: researchers leading that effort are already co-authors). It is unclear why the authors conducted their longitudinal analysis solely on The Gambia – perhaps because they had access to samples from before CQR spread there? There are countries such as Mali where CQ use was discontinued and CRT reverted to wild-type (WT, i.e. K76) and it would be interesting to know the trajectory of AAT mutations in those areas. If AAT1 and CRT were co-evolving, as is proposed, can the authors explain why the prevalence of mutant CRT K76T does not correlate with the proportions of AAT1 S258L between the various continents and African regions (Figure 2A)? Is that association holding only in West Africa?
2. The relationship between AAT1 and CQR and the cross remains somewhat murky. The bulk segregant analysis (BSA) in Figure 3 clearly showed selection for the chromosome (Chr.) 6 peak containing the AAT1 double mutant (S258L and F313S). One wonders if this is driven primarily by mutant AAT1 compensating for mutant CRT, as opposed to mutant AAT1 augmenting CQR. Two lines of evidence support this: 1) Figure 4A shows that almost all mutant CRT progeny that were cloned with CQ pressure are mutant AAT1, yet WT CRT progeny are nearly equally distributed between mutant and WT AAT1 (a simple Chi-squared test would provide evidence whether co-inheritance is statistically non-random); 2)

8gene editing studies shown in Figure 5B showed that the S258L/F313S parasite (NHP4026) had a CQ IC50 value that was not higher than the WT 258S/313F haplotype (NF54). Based on that result there should be not be selection for mutant AAT1 under drug pressure. Indeed, the 258S/313F line had a CQ IC50 that was slightly higher than S258L/F313S. So the argument that the double mutant AAT1 haplotype augments CQR is not supported by the gene editing results.

3. Are there additional genetic factors that can explain why reverting NHP4026 to wild-type AAT1 did not decrease the CQ IC50 (in Figure 5B) as expected from the progeny data? It is also interesting to note that the authors hardly obtained any progeny carrying the WT AAT1/MUT CRT (only 2 out of total 109 unique recombinants; see Figure 4A) during the cloning of the recombinant pools, despite the result that the removal of mutations in AAT1 in the NHP4026 parent enhanced the fitness of the parasite significantly. More explanation is required.

4. In Figure 3B for the BSA, there is a significant QTL peak on chr 14 that appears to be selected by CQ in both pools. The peak has a similar G prime score as the Chr. 6 AAT1 loci, which may have additional importance in CQR or fitness. Does this QTL contribute to the CQ response, and what is its role in relation to AAT1 and CRT?

5. The gene editing data shows that the highest CQ IC50 was observed with the 258L/313F mutant. Yet in SEA it seems that almost all parasites (96%) are F313S and the combination of 258L/313S (i.e. NHP4026) is substantially less resistant. So why would CQ have been the selective pressure driving the 313S mutation?

6. The study shows that the AAT1 single mutant S258L is more resistant to CQ but less fit than the double mutant, S258L/F313S in the absence of CQ pressure/ However, from their analysis of clinical isolates, African parasites either are WT or single mutant S258L, whereas Asian parasites are generally double mutants. Because transmission rates differ greatly between Africa vs Southeast Asia, one would expect the least fit single mutants to be easily outcompeted by other parasites in a high-transmission environment with high multiplicity of infection in the absence of CQ use over the past few decades. However, this is not the case here. In contrast, the transmission rates in SEA are much lower and intra-host competition is less of a barrier to allowing the proliferation of less fit and more resistant parasites. How do the authors reconcile their findings in the context of transmission rates?

7. What is the NHP4026 CRT haplotype? How does this compare to CRT haplotypes in The Gambia or other African regions? Among CRT K76T mutants, several haplotypes of CRT exist and their prevalence markedly differs between geographical regions (see PMID 33824913). It is rather surprising that the authors have not mentioned this when the AAT1 haplotypes are well documented in this manuscript. Also, it would be important to test the impact of the AAT1 Southeast Asian mutations S258L+F313S on CQ IC50 and fitness in 3D7 as well as other African parasites from The Gambia and Mali. The manuscript would be strengthened by adding data from AAT1 CRISPR/Cas9 editing on African parasite lines. This

would help test the authors' main hypothesis that AAT1 plays a central role in CQR evolution, especially since the co-selection was observed in parasites from The Gambia.

8. There are several overarching statements made with regards to epistasis (lines 170, 189, 203) but the article needs to mention the type of statistical analysis used to confirm this relationship between AAT1 and CRT in CQR or between the AAT1 mutations and CQR or fitness. QTL mapping for CQ IC50 on the individual cloned progeny would ascertain the association and epistatic interactions between CRT and AAT1 and perhaps the QTL on Chr. 14. The authors should look into this. Nonetheless, the AAT1 gene editing performed herein investigates the epistasis of AAT1 with CRT only in the NHP4026 SEA line but not in an African genetic background. Hence the major conclusion in this study is limited to a particular parasite background from a single origin.

9. The authors should also further clarify their gene editing data. I'm unclear whether the 258L/313S mutant described in Figure 5 is the NHP4026 parent edited to express the same mutations (with silent mutations to prevent cleavage involving the PAM site) or whether that was NHP4026 and the only gene-edited mutants were the other three combinations. Ideally the best is to have gene-edited all four combinations, and better yet would be to have a control with the silent mutations introduced to prevent re-cleavage of the edited locus.

10. The yeast data are not particularly convincing and I personally feel they are overinterpreted. Figure S10 shows that the S258L variant grows faster than WT in the presence of 5 mM CQ. The authors claim that this provides evidence that S258L confers CQR in yeast. To be robust the authors should have expressed all isoforms in yeast, not just WT and S258L. Also they need to show concentration-dependency in their growth data. The Figure does not stipulate how many independent assays were conducted. Also, there is no evidence that this transporter is present on the yeast membrane. These data are therefore preliminary and in my opinion insufficient to claim that S258L confers CQR. Looking at the parasite data in Figure 5B one would argue that the 313F mutation in the background of S258L is the mutation that augments CQR.

11. The authors rely on their yeast data and references to prior literature to state that AAT1 transports CQ as well as amino acids. I see no evidence of CQ transport in their data. Others have done this with CRT in *Xenopus* oocytes, as shown by Rowena Martin's group that carefully optimized expression and obtained surface expression and then showed differences in accumulation of tritiated drug. Other groups have obtained similar evidence with proteoliposomes or cultured isogenic parasites. For example earlier work by Michael Lanzer or Pat Bray showed that Pf lines expressing mutant CRT had less CQ accumulation compared to isogenic lines expressing WT CRT. Without this type of data the authors should not present a model claiming that AAT1 can transport CQ.

12. Similarly, there is no evidence that AAT1 is actually an amino acid transporter. For example the group of Michael Blackman published in 2021 (PMID 33762339) that another putative amino acid transporter (PF3D7_1231400) is actually a regulator of calcium mobilization. Not to say this AAT1 is

doing something similar, but to point out that annotations without experimental evidence can be misleading. The authors could test their model by examining levels of amino acids or peptides in isogenic lines expressing different AAT1 alleles, using methodology developed by several labs including the recent report from the Lanzer lab that provided evidence for CRT transporting oligopeptides derived from hemoglobin (PMID 35867395). If the authors have evidence that AAT1 is actually involved in amino acid or oligopeptide transport then they should show it.

Minor comments:

1. Line 108: The authors state that the AAT M522I mutation is synonymous. Should it not be non-synonymous?
2. What is the status of the non-S208L mutations in Africa? Including S313F, Q454E and K541N. The text on lines 236-7 is not clear – are those associations with F313S in Africa?
3. Figure 2 – Africa appears to be ~60-65% S258L and the rest is listed as “Other”. What are the other mutations? Also, can the data be broken down by region?
4. Figure 3 – it would help to designate the location of AAT1 and CRT using arrows in panels A and B. The legend refers to Fig. S1 and S2 (Lines 441-442). Are these referencing the correct figures?
5. Figure 4 – can the authors also indicate the location of AAT1 and CRT in panel A? Note in Figure 4 that the MUT AAT / MUT CRT have a slightly higher CQ IC50 than WT AAT1 / MUT CRT. That does not agree with Figure 5B – which should be mentioned. In Figure 4D, the CQ IC50 levels for each recombinant progeny (unique) should be indicated by individual points so that reader can tell how many unique clones were tested. With the large error bars for the WT/MUT and mut/mut groups and limited number of progeny tested of N=2 for the WT/MUT group, more replicates and independent unique recombinants are required. There is a lack of robustness in the data. This is essential to justify the authors’ statement that AAT1 is epistatic to CRT.
6. Figure S1 – can the authors explain whether the selection coefficient reflects a change in relative growth per parasite asexual blood stage cycle? It’s hard to know what s means.
7. Figure S2 – should this be 2001 and not 2000? The coloured boxes in the Key should be reordered chronologically.
8. Figure S3 – the 2001 data (not 2000?) shows linkage disequilibrium (LD) with a Chr. 7 segment downstream of CRT – has that been observed elsewhere? To validate the LD between AAT1 and CRT, the reciprocal analysis showing the R² between CRT K76T SNP and SNPs within Chr. 6 surrounding AAT1 should be done across the various time intervals.
9. Figure S5 – should F313S and S258L/F313S be coloured differently from each other in the PfaAT1 allele Key?
10. Figure S9 – the legend needs to be corrected when it states that the 258L mutation is more resistant to lumefantrine (LUM). That is true for 258L/313S but not for 258L/313F, when compared with 258S/313S.

11. Figure S13 – it is not clear how many independently generated clones for each CRISPR/Cas9 modified lines were used in the pairwise competition fitness experiments (lines 676-677). The methods mentioned that two different clones from each editing (two biological replicates) were run in technical triplicates but the graph in Figure S13 depicts three replicates. Please clarify what these replicates refer to.

12. There is some detail missing in the Methods and Results sections on the duration of the in vitro IC50 drug response assays for CQ and other compounds. Are these 72 hr exposures?

Reviewer #3 (Remarks to the Author):

This paper delves into the fascinating, intertwined evolutionary trajectory of amino acid variants in two genes involved in resistance to CQ, *pfcr1* and *pfaat1*. The combination of population genetics, QTL mapping, and genome editing is impressive and comes together to show convincing evidence for the specific role of *aat1*, and for epistasis, especially between alleles in *pfaat1* in SEA. These results are of significant interest both to the malaria community as well as for the study of pathogen evolution. I do have a number of comments on the presentation of results, evidence behind some claims, and missing discussion points:

1. Although the evidence accumulates on the role of AAT1 in CQ resistance, the presentation of the narrative is odd. The set up is that the authors aim to “understand the contribution of additional parasite loci to CQ-resistance evolution” (lines 83-84), but the results begin immediately with *pfaat1* and the allele frequency trajectory of S258L in particular, without explaining how they identified this gene or allele. Only after the genome-wide results that follow does zeroing in on *aat1* make sense. Or was *pfaat1* S258L the polymorphism that had the largest change in frequency between 1984 and 2014 in the dataset? Are there any other alleles showing a similar pattern or do these two stand out? In the haplotype differentiation scan, *pfdhfr* also comes up – does the allele frequency trajectory look the same or is that different from the alleles in *pfcr1* and *pfaat1*? Either clarifying why *pfaat1* was picked up in this first analysis in particular, or that it was only looked at following other findings that zeroed in on *pfaat1*, would make the ordering more logical.
2. The strong IBD signal at *pfaat1* and *pcr1* in 1984 before the CQ-resistant alleles were present is puzzling. The authors say there was only a synonymous variant (but this is given as M552I, isn't that nonsynonymous?), suggests there may have been previous selective sweeps and they may have targeted a regulatory variant. I did not follow this logic – would this be expected if there was a

completed sweep? Or would that be the case if the selected variant was subsequently lost, and is that what they suggest happened since presumably the parasites in 1984 were susceptible to CQ?

3. Strong selection is inferred on both alleles from the allele frequency change and IBD analyses.

Although it might seem obvious to the authors, an explicit description of what signatures of selection are being detected and whether these are necessarily selective sweeps would be advised.

4. Referencing which signatures and observations have been made previously should be included. Some of this is provided in the Discussion, but which findings are new to this paper vs. being re-iterated here should be indicated in the results. For instance, the IBD signal was reported in the isoRelate paper already (Henden et al. 2018) and several studies have found strong signatures of positive selection on *pfaat1*. Additionally, context in the discussion for what exactly this paper resolves and what remains unexplained would be helpful to reach a wider audience.

5. The claim that the 258L allele has an independent origin in Asia and Africa should be better supported. What is truly the evidence for this? The haplotype networks do not seem convincing as presented. Fig. S5 seems to be the main evidence for this, where 258L is found in several parts of the tree. However, this is a large region (50kb) that is being considered and it is not obvious that 2 origins fits better than 1. Additionally, it is not clear which parts of Fig. 2B the labels of “Africa” and “Asia” refer to. To make the claim for convergent evolution a stronger test is needed. I also could not tell the difference between the colors used for F313S and S258L.F313S.

6. Why does Fig. 3B look very different from Fig. 3C and from Fig. S6 and S7? In 3B the chr6 QTL does not appear to reach much higher than the significance threshold, not up to 50 like it does in the other plots. Also, it is not obvious that AAT is at the peak of the QTL in Fig. 3C – could there also be a zoom in to the QTL evidence together with the genes plot?

7. Is the epistasis shown in Fig. 4D sufficient to explain both the strong LD between *pfcr1* and *pfaat1* in natural populations (Fig. S3) and/or in the cross? I was surprised that the CRISPR and growth analysis is only for polymorphisms in *pfaat1*. That’s fine, but a discussion of how these results clarify or explain the inter-chromosomal LD is warranted, or whether it remains to be determined.

8. The level of detail for sequencing and variant calling is very different between the longitudinal samples and that of BSA, with the latter much more complete. There needs to be better description for the longitudinal samples than “mapping on the *Pf* reference genome employed *bwa* and BAM files and were optimized with Picard tools”. Bam is a file format not something that is employed, and Picard is a tool so this gives no information on what was “optimized”. Additionally, the filtering is not fully described and why almost half the samples were excluded is not explained. The only two filters listed are for genotype missingness < 10% and MAF > 2%. The MAF filter is on variants, is the genotype missingness per individual? Why were so many samples filtered out? Were indels excluded?

Minor

131. In the introduction, it is mentioned that other variants in MDR1 and within CRT have been reported to modulate CQ resistance. Were those variants not identified in these analyses and does that suggest they were not major players? Given the goal
2. Some inconsistencies in the terminology referring to the variants, such as referring to amino acid polymorphisms as SNPs (“the K76T SNP”), whether referring to the polymorphism or the allele (“K76T” vs “76T”), and then later as “wt/mut” without defining. A consistent usage would be helpful.
3. Some inconsistencies in the use of the word “linkage” including “functional linkage” (what is meant by that beyond a ‘functional relationship’?) and using “linkage” when “linkage disequilibrium” appears to be meant.
4. It is stated that “all IC50 levels were below levels of clinical significance” for the drugs tested besides CQ. What are IC50 levels for clinical significance for these drugs – could that be indicated in figure S9?
5. Were any off-target sequence changes detected from the whole genome sequencing of CRISPR parasites? It just says this was done but does not describe what was found (that I found).
6. Should pfaat1 be in parentheses in the title, I was not clear whether “amino acid transporter” is the long form of the gene name, or a description of what pfaat1 is.
7. A key to the columns in supplementary tables would be useful.
8. In the legend for Fig S2, the years are not in order and it makes it more difficult to interpret
9. Why is Fig. S8 referenced in line 140 – should this reference a different figure?

Reviewer #4 (Remarks to the Author):

In this manuscript, the authors used an array of cutting-edge technologies (population genomics, genetic cross, and CRISPR-based gene editing) to elucidate the compensatory effect of the amino acid transporter Pfaat1 S258L mutation on the evolution of chloroquine (CQ) resistance. This study demonstrated the epistatic effect of pfaat1 on pfCRT. While the S258L mutation significantly boosted the CQ resistance, it also reduced the fitness of the parasites. Another mutation that is fixed in SE Asia, pfaat1-F313S, although reduced the CQ resistance level, it restored fitness of the parasite. This study provided evidence on the significance of compensatory mutations in the evolution of antimalarial drug resistance (like the ones occurring with the PfK13 mutations in SE Asia).

1. The genomics work with the longitudinal samples provided a direct view of the parallel selection of both the pfCRT and pfaat1 S258L haplotypes. Fig S3 showed interchromosomal LD between these two loci. I would like to see if pfCRT and pfaat1 surrounding regions can be zoomed in to show the size of the

selection valleys (changes in heterozygosity in these genes and flanking regions)? It is also puzzling that dhfr was also selected during this region, why?

It is odd that figure 1 included a threshold line based on a significance value at $-\log(p\text{-value})=3$, but then wrote genes out for only the "top 1% of p values". Please specify in the figure legend and justification. I would also consider moving the red lines on their graph to 5 since that is their cutoff for inclusion as a "region of interest".

2. The different effects of *aat1* mutations (S258L and F313S) on CQR and fitness were done with the SE Asian parasite NHP4026. The results would deserve evaluations in an African parasite line to prove that the observed effects were not due to the Asian parasite's genetic background. Could you also speculate on why the F313S mutation has not evolved in the African parasites, if it offers fitness compensation?

3. it would be very useful if the mechanistic experiments performed in yeast for the malaria parasite could be replicated in malaria parasites using their CRISPR mutants.

Minor comments:

I'm assuming they excluded 1990 in the iR analysis because the sample size was small, but it may be worth explicitly stating the criteria of inclusion. For example, "we performed this analysis for all years with sample size > 30" OR "we performed this analysis for all years, but 1990 yielded no significant results, likely because of sample size". Those are two pretty different statements, in my opinion, so readers shouldn't have to guess which is true.

Line 202 – 258S (not underlined)/313S (underlined)

203 – a strong epistatic interaction or strong epistatic interactions

Author Rebuttal to Initial comments

Response to reviewers

We thank the reviewer's and editor for their comprehensive and insightful comments on our manuscript. We were pleased to see that all four reviewers were excited by the results, although there was some disagreement about interpretation. We have addressed each point here (red text) included several new analyses, and made extensive changes to the manuscript in the light of these comments. We hope the revised manuscript will now be suitable for publication.

Reviewer Comments:

Reviewer #1 (Remarks to the Author):

Summary and background

Despite the way Genetics is routinely presented in high school and undergraduate classes, most genetic/genomic changes that have a recognizable phenotype involve more than a single gene. This manuscript describes a coordinated series of experiments performed by researchers in several laboratory groups in the US, the UK, the Gambia and Thailand. This paper has capitalized on the availability of samples from the Gambia collected over a crucial period during which the dominant antimalarial at the time, chloroquine (CQ), was intensively used in this small country and then the regimen was changed to a combination of CQ + sulfadoxine-pyrimethamine and then an ACT, lumefantrine + artemether. The main question posed is- are changes in a single well known determinant of CQ resistance (pfcr) accompanied by any other genomic changes that parallel those in prevalence of pfcr in the Gambian parasites? If so, does this interaction affect the parasite response to CQ treatment?

The answer is definitely yes! The complex sets of experimental approaches used have returned a detailed picture of the functional interactions between pfcr and pfcy in malaria parasites. The group has identified the epistatic interactions of these two genes and probed the functional consequences for the malaria parasites that carry them. The answer reminds researchers that a single allele of a gene is rarely the only determinant of a complex phenotype even in a unicellular organism.

The data are assembled by combining some "older" approaches- whole genome analysis of the Gambian parasite series over time by the MalariaGEN project. These longitudinal data were then compared by several newer approaches. First, a relatively new approach- a series of Pf genetic crosses that differed in pfcr was completed in mouse strains altered to express human liver and red cells. These made it possible to identify genetic changes anywhere in the Pf genomes that paralleled the changes in pfcr in the Gambian set over time.

16Progressively smaller sections of the Pf genome that did show this parallel response to antimalarial exposure were identified, and finally narrowed to a section on chromosome 6 that contained another gene, *pfaat1*. This gene was known to encode a protein with a similar cellular location in the parasite and also a similar function transporting amino acids out of the digestive vacuole. Most important, changes in prevalence of particular mutations in *pfaat1* increased in prevalence in the Gambian data set over time. The particular changes in these two genes appeared to be responding to similar selection pressures - *pfcr*t with the key mutation K76T and distinct alleles of *pfaat1* that carried S258L.

The longitudinal study was then extended to parasites that evolved over a similar time frame and several changes of antimalarial treatment in Shoklo, Thailand on the Thai-Myanmar border. These parasites all carried the same *pfcr*t allele that encoded a 5 amino acid bearing *pfcr*t allele CVIET, amino acid 72-76. They observed similar evolution of the *pfcr*t and *pfaat1* haplotype pair in these SE Asian parasites, as well.

Then, lab based studies were initiated to explore whether this parallel temporal pattern had functional consequences for the phenotype in parasite cultures with *pfcr*t that expressed the CVIET allele and various alleles of *pfaat1*. Here, the capacity of CRISPR/cas9 -based editing of cloned parasites was used to create a series of clones with various combinations (haplotypes) of *pfcr*t and *pfaat1* alleles. Particular responses to CQ treatment with respect to susceptibility to CQ in vitro, and competitions between pairs of clones with various combinations of the *pfaat1* and *pfcr*t measuring sustained growth in vitro as a surrogate for fitness. Here, they hit the jackpot: parasites that carried the combinations of *pfcr*t K76T and *aat1* S258L had differences in susceptibility of Pf parasites to CQ and the success of growth in the presence of CQ, in vitro.

As an extra bonus, they studied *Saccharomyces cerevisiae* (baker's yeast) that expressed the *pfaat1* gene bearing the S258L protein and measured susceptibility to CQ and other quinolone drugs, and amino acid transport in the presence of CQ. They inferred that even in yeast, the AAT1 protein was involved in amino acid and CQ transport. In addition, the authors used now widely available models to try to explore changes in the protein structure of various single and double mutant alleles of the AAT1 protein predicted by the models.

Based on these predictions, they considered how the acquisition of an F313S mutation by *pfaat1* S258L might affect the protein structure and be manifest in changes in transport functions that they observed.

Assessment

This paper is remarkable in several ways. The creative use of already available genomic information to ask a new question is a particular starting point. The process of winnowing the chunk of DNA that contained the region that increased in prevalence in the Gambian parasites required a complex series of genetic crosses in humanized mice. This technology was applied by the consortium that had developed the approach, and convincingly identified the *aat1* gene as the responding gene. As important, the consortium grew to include researchers in SE Asia and those with expertise in gene editing and with testing the in vitro responses to CQ growth. Despite the large number of groups involved, the manuscript is clearly written and the logic for the various approaches used to identify the *pfaat1* gene is explained. However, the addition of a flow chart showing the logic of each step, and the outcome followed by the next step- Bulk Segregant Analysis followed by detailed crosses of specific clonal haplotypes for example might be a helpful guide for readers.

We thank the reviewer for their enthusiastic comments, and recognizing the array of complementary approaches we have used. We have added a flow chart to illustrate the project design – Supplementary figure S15.

The figures are clearly designed and informative with the exception of Figure 4A. The lists at the very top apply to the vertical bars at the far left? Is the first vertical bar the 4 “contests” that were run, the second bar shows comparisons between controls and CQ selected. The red and black are heat maps to show alleles from 3D7 or NHP across the genomes. I’m not sure that the composite heat map conveys this information at all clearly. Some more discussion on how to present this information is needed, or consider omitting 4A or presenting the proportions of each haplotype in a table.

We have retained the heatmap, but simplified the figure design by removing the color codes to improve clarity. We have also added arrows to show the genomic position of *pfprt* and *pfaat1* (as suggested by reviewer 2). We think the revised figure now conveys the key information much more clearly.

Specifics

The methods appear to be reasonably detailed, and provide specific references for additional information. However, the genetic crosses in particular are not easily performed by groups without this experience, or access to the specialized humanized mice so collaborating with these authors is the most sensible pathway for researchers who might want to use similar strategies.

The genetic crosses certainly require specialized facilities (mosquito colonies) and expertise. The papers cited (e.g. Vaughan et al, Nature Methods 2015) provide the methods for conducting crosses with humanized mice, while Brenneman et al (2022) describes methods for bulk segregant analysis of drug response. We now cite additional papers (Vendrey et al 2020; Li et al, 2022; Kumar et al 2022) to provide further information on these approaches (line 593-595). We are open to collaborations using the humanized mouse crosses to investigate interesting *Plasmodium* traits, including several now underway.

I understand the motivation to use structure modeling to try to understand how the particular interactions of the amino acid changes in the AAT1 protein carrying different combinations of changes at the 258 and 314 codons. However, the conclusions are not particularly informative and add little to this very comprehensive paper. This line of thought would best be explored in detail with collaborators with structural expertise. In the context of the current paper, this section is better omitted.

We have reduced the section describing the structural analysis, to remove speculation on the role of these amino acids from the text. However, we would like to retain a supplementary figure containing this information (Fig S14). The structural analysis demonstrates that these high frequency non-synonymous mutations are found in transmembrane domains and shows the similarities with PfCRT. While we cannot make strong inferences about function or cause of epistasis, this initial analysis will provide a basis for further future work on this topic.

It has been known for a long time that selection of CQ resistant parasites was first observed in Asia and later that exact allele CVIET (codons 72-76) was brought to South Africa then was spread by migration of infected people all over the African continent. However, in South America a different extended sequence SVMNT (aa 72-76) evolved and predominates. In South American parasites, CQ resistance is conferred by this genotype and it does not seem to cause any decrease in fitness in parasites that carry it. Would it be useful to mention this in the discussion? Perhaps this large group is already considering how that South American version of CQ resistance might behave differently? It's at least worth a mention.

This is an excellent point. We know that SNPs within PfCRT (and the haplotype at codons 72-76) contribute to drug resistance and/or fitness of parasites. We have now expanded our population genomic analysis to investigate the haplotype distribution of PfCRT haplotypes (rather than only PfCRT K76T) in relation to PfAAT1 mutations (see revised Fig. 2, Table S2, and Fig. S6 showing distribution of PfCRT and PfAAT1 haplotypes in Africa).

For both Southeast Asia and Africa, the major PfCRT allele present is CVIET, with only regional occurrence of CVIDT. The major PfCRT alleles in South America are CVMNT (27.0%), SVMNT (29.7%) and CVMET (43.2%), while the PfAAT1 alleles include S258L (64.9%), V231D (16.2%), I248T (10.8%) and V231D/P446A (8.1%). We are not specifically working with South American samples, but we now refer to work by another group (Carrasquilla et al., PLoS Pathogens, in press) that shows strong selection around pfaat1 in South America. See line 126-130 and Fig. S6 in revised manuscript.Reviewer #2 (Remarks to the Author):

This submission by Amambua-Ngwa et al presents some interesting evidence supporting mutations in AAT1 as a contributor to the evolution of chloroquine resistance (CQR) in *Plasmodium falciparum* (Pf). The report combines analysis of field samples collected over time from The Gambia, and the distribution of AAT1 haplotypes globally. With the results of a genetic cross between a Thai CQ-resistant (NHP4026) and an African CQ-sensitive (3D7) line. These data collectively provide evidence supporting AAT1 as a secondary determinant, acting epistatically with mutant forms of the primary CQ efflux transporter CRT.

Overall, this is an interesting and impressive body of work that I think will be of broad interest. Some aspects, however, require further review. There are several gaps of knowledge in this work that need to be addressed to better substantiate the broader conclusions of this study. The study does not elucidate the temporal history of whether AAT1 or CRT mutations first emerged in the field and which AAT1 mutations among the four are critical for helping compensate for mutant CRT parasites' poor fitness. A large part of this study is trying to link and extrapolate findings between datasets that are not entirely conclusive.

Thanks for recognizing the scope and importance of this study. We appreciate the very thorough review that has led to important improvements to our manuscript. Regarding the temporal history of *pfcr*t and *pfaat*1 alleles – we agree it is an intriguing question that underscores the significance of our finding that another gene, besides the very well-studied *pfcr*t, may have played a pivotal role in the emergence and spread of chloroquine resistance. While our data cannot determine the order in which mutations in PfCRT and PfAAT1 occurred, we show clearly that mutations in both genes were absent in 1984 samples, but present in 1990. Therefore, a key conclusion of this work is that both mutations arose very close together in time and spread in parallel, providing strong evidence that these mutations are driven by the same selection pressure. This is strongly suggestive of a functional interaction (further supported by detailed work in this manuscript). We now emphasize this result in the text (line 95-105, line 113-118).

Our data demonstrate that the PfAAT1 S258L mutation increases CQ IC₅₀ but carries a large fitness burden. In Asia, the PfAAT1 S258L + F313S allele compensates for the lack of fitness associated with mutant PfCRT K76T. The results from the genetic cross are consistent with the PfAAT1 S258L + F313S mutant compensating for reduced fitness in PfCRT K76T (see below major comments #2 for details).

21Major comments:

1. The authors present an identity-by-descent (IBD) analysis to demonstrate the strong co-selection of mutations in AAT1 and the CRT CQR marker mutation K76T in a set of 321 isolates from The Gambia, sampled from 1984 to 2014. It is not clear why this analysis has been limited to only samples from The Gambia when the MalariaGen Pf6 dataset contains more than 3,000 samples from various regions in Africa (note: researchers leading that effort are already co-authors). It is unclear why the authors conducted their longitudinal analysis solely on The Gambia – perhaps because they had access to samples from before CQR spread there? There are countries such as Mali where CQ use was discontinued and CRT reverted to wild-type (WT, i.e. K76) and it would be interesting to know the trajectory of AAT mutations in those areas. If AAT1 and CRT were co-evolving, as is proposed, can the authors explain why the prevalence of mutant CRT K76T does not correlate with the proportions of AAT1 S258L between the various continents and African regions (Figure 2A)? Is that association holding only in West Africa?

The reviewer is correct – we focused on longitudinal samples from the Gambia because we (Alfred Amambua Ngwa, David Conway, Umberto D’Alessandro) had access to longitudinal samples collected from between 1984 and 2014 from the Gambia. Such longitudinal samples are not available in the PF6 database from other regions of Africa. We certainly agree that parallel analyses in Malawi would be extremely valuable, but we do not have access to these samples or sequence data.

With regard to correlations between PfAAT1 and PfCRT mutations. We thank the reviewer for the suggestion to examine PfCRT haplotypes and PfAAT1 evolution in more depth across Africa. In the light of the reviewer’s comments we have expanded our analysis to examine distribution of PfAAT1 and PfCRT alleles across Africa. This new fine-grained analysis across Africa clearly shows that PfCRT K76T correlates strongly with the proportions of PfAAT1 S258L in West Africa ($R^2 = 0.65$, $p=0.0017$) and across all African populations ($R^2 = 0.44$, $p=0.0021$). This analysis further strengthens the argument for co-evolution and epistasis between these two genes. We have added this to the main text (lines 126-130); the analysis is provided as Supplementary Fig S6.

It is not clear why this analysis has been limited to only samples from The Gambia when the MalariaGen Pf6 dataset contains more than 3,000 samples from various regions in Africa (note: researchers leading that effort are already co-authors).

We did not in conduct IBD analyses for other countries across Africa in the initial submission because the lead Author (Amambua-Ngua) has previously published a paper in Science detailing IBD evidence for strong selection on chr 6 (region containing PfAAT1) as well as PfCRT in African populations (Amambua-Ngua et al. Science 2019). However, we have now included such analyses using public databases, so this information is accessible in one place (Fig S3). These analyses add further evidence that *pfaat1* has been under strong selection in global locations and complement published results (Amambua-Ngua et al. Science 2019; Henden et al. PLoS Genetics 2018; Carrasquilla et al. Biorxiv 2022).

We thank the reviewer for these insights and think the expanded population genetic analysis greatly strengthens the argument that these two genes have co-evolved.

2. The relationship between AAT1 and CQR and the cross remains somewhat murky. The bulk segregant analysis (BSA) in Figure 3 clearly showed selection for the chromosome (Chr.) 6 peak containing the AAT1 double mutant (S258L and F313S). One wonders if this is driven primarily by mutant AAT1 compensating for mutant CRT, as opposed to mutant AAT1 augmenting CQR. Two lines of evidence support this: 1) Figure 4A shows that almost all mutant CRT progeny that were cloned with CQ pressure are mutant AAT1, yet WT CRT progeny are nearly equally distributed between mutant and WT AAT1 (a simple Chi-squared test would provide evidence whether co-inheritance is statistically non-random); 2) gene editing studies shown in Figure 5B showed that the S258L/F313S parasite (NHP4026) had a CQ IC₅₀ value that was not higher than the WT 258S/313F haplotype (NF54). Based on that result there should be not be selection for mutant AAT1 under drug pressure. Indeed, the 258S/313F line had a CQ IC₅₀ that was slightly higher than S258L/F313S. So the argument that the double mutant AAT1 haplotype augments CQR is not supported by the gene editing results.

Thank you for laying out this argument so clearly. We agree that a compensatory role for the PfAAT1 S258L/F313S double mutant allele provides a very straightforward explanation of the results obtained from our genetic cross, including our failure to recover cloned progeny carrying the WT AAT1/MUT CRT (see point below also). The Chi-squared test ($X^2 = 12.295$, $p\text{-value} = 0.000454$) provides evidence that inheritance is non-random and is now included in the revised manuscript. See Table S6 and line 175-179 for details.

The progeny IC₅₀ data provided evidence that the PfAAT1 S258L/F313S allele confers a marginal increase in IC₅₀. However, as the reviewer notes, the statistical strength of the comparison of IC₅₀ in the progeny clones is limited because we recovered only two clones carry WT PfAAT1/MUT PfCRT. Furthermore, the initial submission contained errors in the IC₅₀ analysis, as parental parasites were inadvertently included. The corrected analysis does not show

a significant impact of PfAAT1 genotype on IC_{50} . We have noted this limitation in the text, moved Fig 4D from the main text to a supplementary figure (Fig S11), removed the argument for an epistatic interaction between PfCRT and PfAAT1 as a contributor to IC_{50} s, and focused the functional analysis on the CRISPR experiments using isogenic parasites.

3. Are there additional genetic factors that can explain why reverting NHP4026 to wild-type AAT1 did not decrease the CQ IC_{50} (in Figure 5B) as expected from the progeny data? It is also interesting to note that the authors hardly obtained any progeny carrying the WT AAT1/MUT CRT (only 2 out of total 109 unique recombinants; see Figure 4A) during the cloning of the recombinant pools, despite the result that the removal of mutations in AAT1 in the NHP4026 parent enhanced the fitness of the parasite significantly. More explanation is required.

The gene editing experiments reveal that clones carrying the ancestral PfAAT1 allele in combination with PfCRT K76T have the highest fitness cost. In contrast, the close association of the PfAAT1 S258L/F313S alleles with PfCRT K76T in the progeny from the cross reveals the opposite relationship. We speculate that these opposing results may reflect differing selection in the blood-stage parasites in the case of CRISPR experiments, or in the mosquito and liver stage in the case of genetic crosses.

4. In Figure 3B for the BSA, there is a significant QTL peak on chr 14 that appears to be selected by CQ in both pools. The peak has a similar G prime score as the Chr. 6 AAT1 loci, which may have additional importance in CQR or fitness. Does this QTL contribute to the CQ response, and what is its role in relation to AAT1 and CRT?

Careful inspection of the chr 14 peak suggests that it is not driven by CQ selection (Fig S8, now Fig S10). Frequencies of SNPs in this region do not change under CQ-pressure, suggesting that this peak is not driven by CQ. This is explained in Fig S10 legend.

We have previously observed that this QTL is driven by specific culture media (Kumar et al. *Frontiers in Cellular and Infection Microbiology*, 2022). The 3D7 allele frequency of this loci increases in culture containing AlbuMAX, but decrease in culture containing serum. In this study, we used culture contain AlbuMAX, which explains the increase of 3D7 allele frequency in control cultures.

5. The gene editing data shows that the highest CQ IC_{50} was observed with the 258L/313F mutant. Yet in SEA it seems that almost all parasites (96%) are F313S and the combination of 258L/313S (i.e. NHP4026) is substantially less resistant. So why would CQ have been the selective pressure driving the 313S mutation?

To clarify, we did not propose that elevated CQ IC_{50} drives PfAAT1 F313S in Asia. Instead, we argued that addition of F313S reduces IC_{50} , while partially restoring fitness in SE Asia. CQ was withdrawn as first line treatment against *P. falciparum* 20-30 years ago in Asia, but until recently has been widely used for the treatment of *P. vivax*. There is therefore still CQ pressure on co-infecting *P. falciparum* infections. We believe that the PfAAT1 F313S allele acts as a compensatory mutation in restoring fitness in parasites carrying PfCRT K76T resistance-associated mutations. We have modified the text to clarify our argument (line 216-229).

6. The study shows that the AAT1 single mutant S258L is more resistant to CQ but less fit than the double mutant, S258L/F313S in the absence of CQ pressure/ However, from their analysis of clinical isolates, African parasites either are WT or single mutant S258L, whereas Asian parasites are generally double mutants. Because transmission rates differ greatly between Africa vs Southeast Asia, one would expect the least fit single mutants to be easily outcompeted by other parasites in a high-transmission environment with high multiplicity of infection in the absence of CQ use over the past few decades. However, this is not the case here. In contrast, the transmission rates in SEA are much lower and intra-host competition is less of a barrier to allowing the proliferation of less fit and more resistant parasites. How do the authors reconcile their findings in the context of transmission rates?

The different trajectory of PfAAT1 evolution in Africa and Asia is one of the exciting aspects of this work, so understanding the drivers is of great interest. African and SE Asian parasite populations differ in transmission intensity, as the reviewer notes, but also in levels of inbreeding and effective recombination rate, and in levels of drug pressure (see also comment above). We agree that parasites bearing PfCRT K76T alleles may have been retained across Asia because within-host competition is less intense than in Africa, as pointed out by the reviewer. Compensatory evolution involving PfAAT1 F313S in addition to S258L mutations also can contribute to improve fitness of SEA parasites carrying PfCRT K76T alleles.

Our results are consistent with PfAAT1 S258L spreading in the Gambia because it confers higher levels of CQ-resistance. We agree that PfAAT1 S258L alleles will be expected to decline with removal of CQ pressure, as has occurred in the case of PfCRT K76T alleles. Further analysis of longitudinal datasets will reveal whether this is occurring. While PfCRT K76T has declined in African countries, it has remained close to fixation in SE Asia, perhaps because of residual CQ pressure from treatment of *P. vivax*. Based on our CRISPR results we suggest that the common Asian PfAAT1 S258L/F313S alleles help restore fitness to parasites carrying PfCRT K76T alleles.

7. What is the NHP4026 CRT haplotype? How does this compare to CRT haplotypes in The Gambia or other African regions? Among CRT K76T mutants, several haplotypes of CRT exist and their prevalence markedly differs between geographical regions (see PMID 33824913). It is rather surprising that the authors have not mentioned this when the AAT1 haplotypes are well documented in this manuscript. Also, it would be important to test the impact of the AAT1 Southeast Asian mutations S258L+F313S on CQ IC50 and fitness in 3D7 as well as other African parasites from The Gambia and Mali. The manuscript would be strengthened by adding data from AAT1 CRISPR/Cas9 editing on African parasite lines. This would help test the authors' main hypothesis that AAT1 plays a central role in CQR evolution, especially since the co-selection was observed in parasites from The Gambia.

We have now expanded the population genomic analysis to examine the distribution of *pfcr* haplotypes in relation to PfAAT1 mutations (see revised Fig 2 and Fig S6).

The NHP4026 parasite used for editing carries the CVIET *pfcr* haplotype common in Asia and Africa (see also our reply to reviewer 1), as expected. This makes it a suitable parasite for gene editing studies. We agree with the reviewer that gene editing of *pfaat1* in African parasite genetic backgrounds will be valuable, a point that has led to a recently funded proposal to conduct this work. We will conduct this work over the next few years and will report these data in subsequent papers. However, this is beyond the scope of the current manuscript.

8. There are several overarching statements made with regards to epistasis (lines 170, 189, 203) but the article needs to mention the type of statistical analysis used to confirm this relationship between AAT1 and CRT in CQR or between the AAT1 mutations and CQR or fitness. QTL mapping for CQ IC50 on the individual cloned progeny would ascertain the association and epistatic interactions between CRT and AAT1 and perhaps the QTL on Chr. 14. The authors should look into this. Nonetheless, the AAT1 gene editing performed herein investigates the epistasis of AAT1 with CRT only in the NHP4026 SEA line but not in an African genetic background. Hence the major conclusion in this study is limited to a particular parasite background from a single origin.

See answer to Q. 7. We have removed the arguments for epistasis based on the analysis of cross progeny because the numbers of clones bearing each combination of PfAAT1 and PfCRT genotypes was limited. We agree that examining sufficient numbers of parasites bearing the various allele combinations, would allow investigation of epistasis. However, epistasis will be more directly addressed using gene editing of these two loci both individually and in

combination. We now have funding for this work, and will report the results in an independent paper.

Our manuscript reports 11 independent gene edits on the NHP4026 (Asian) background. This parasite carries the PfCRT CVIET haplotype which predominates in both Asia and Africa. We agree that further gene editing studies will be valuable in detailing these relationships, and now we have funding to conduct this work. However, we don't agree that gene editing of a single parasite background is a weakness. We note that the majority of the elegant gene manipulation studies to investigate PfCRT function have been conducted in a single parasite genotype (GCO3), a progeny clone from the HB3 x Dd2 genetic cross (e.g., see Sidhu et al Science 2002; Gabryszewski PLoS Pathogens 2016).

9. The authors should also further clarify their gene editing data. I'm unclear whether the 258L/313S mutant described in Figure 5 is the NHP4026 parent edited to express the same mutations (with silent mutations to prevent cleavage involving the PAM site) or whether that was NHP4026 and the only gene-edited mutants were the other three combinations. Ideally the best is to have gene-edited all four combinations, and better yet would be to have a control with the silent mutations introduced to prevent re-cleavage of the edited locus.

The PfAAT1 S258L/F313S mutant described in Figure 5 is NHP4026. This is detailed in the methods (line 692-693), and in Fig 5A, and is now further clarified in the Fig 5 legend. We did not conduct control edits with shield mutations only – this is not common practice in malaria work using CRISPR/Cas9, because malaria lacks the error-prone non-homologous end joining repair system that generates off-target changes in other eukaryotes. However, we bolstered our approach by conducting three independent edits (edits recovered from independent flasks) for each genotype constructed (11 CRISPR edits in total). Each of these show very similar results for both IC₅₀ and fitness assays, providing strong evidence that the phenotypes observed do not result from off target mutations.

10. The yeast data are not particularly convincing and I personally feel they are overinterpreted. Figure S10 shows that the S258L variant grows faster than WT in the presence of 5 mM CQ. The authors claim that this provides evidence that S258L confers CQR in yeast. To be robust the authors should have expressed all isoforms in yeast, not just WT and S258L. Also they need to show concentration-dependency in their growth data. The Figure does not stipulate how many independent assays were conducted. Also, there is no evidence that this transporter is present on the yeast membrane. These data are therefore preliminary and in my opinion insufficient to claim that S258L confers CQR. Looking at the parasite data in Figure 5B one would argue that the 313F mutation in the background of S258L is the mutation that augments CQR.

27We respectfully disagree that the yeast results are over-interpreted. The new yeast data reported in this paper add to a body of published work (Tindall et al 2018) examining the impact of the PfAAT1 T162E mutation that underlies CQ resistance in rodent malaria parasites. That manuscript provides experimental data to show that PfAAT1 is expressed in the plasma membrane of yeast, just as PfCRT is expressed in the plasma membrane of yeast. We have now included further details in the figure S13 legend to detail the number of replicates (3 independent experiments). Tindall et al (2018) also provides clear evidence on the localization of PfAAT1 in the yeast membrane –now noted in the Fig S13 legend. We agree with the reviewer that observations of *Plasmodium* genes in experimental systems should be interpreted with caution, but we reject the characterization of these results as preliminary, and think they make a valuable contribution our understanding of the role of PfAAT1 in *Plasmodium*.

One clarification regarding Fig 5B – Pf AAT1 313F is the ancestral state at this codon, not a mutation. We have clarified this in the text to avoid confusion. Hence our argument that the PfAAT1 S258L mutation confers high CQ IC₅₀ and low fitness. Addition of the PfAAT1 F313S mutation is seen in Asia where PfAAT1 alleles with the 258L/313S double mutation are found.

11. The authors rely on their yeast data and references to prior literature to state that AAT1 transports CQ as well as amino acids. I see no evidence of CQ transport in their data. Others have done this with CRT in *Xenopus* oocytes, as shown by Rowena Martin's group that carefully optimized expression and obtained surface expression and then showed differences in accumulation of tritiated drug. Other groups have obtained similar evidence with proteoliposomes or cultured isogenic parasites. For example earlier work by Michael Lanzer or Pat Bray showed that Pf lines expressing mutant CRT had less CQ accumulation compared to isogenic lines expressing WT CRT. Without this type of data the authors should not present a model claiming that AAT1 can transport CQ.

The prior published yeast data from our co-authors provide strong evidence that PfAAT1 can transport both CQ and amino acids (Tindall et al., 2015). Review papers by Rowena Martin also provide evidence that PfAAT1 is an amino acid transporter (Martin Biol Rev 2020). We have presented a model that summarizes these multiple lines of published evidence, providing a clear framework for direct functional hypothesis tests in in future work. While we agree that further work using the *Xenopus*, yeast or *Plasmodium* system to investigate mechanisms of action will be extremely valuable, this is clearly beyond the scope of the current paper. We are confident that our novel observations will inspire much future work on pfAAT1 ranging from evolutionary genetics to molecular and cellular function. In the revised text, we have further cited available

evidence to strengthen the argument that PfAAT1 can transport CQ as well as amino acids (line 236-240).

12. Similarly, there is no evidence that AAT1 is actually an amino acid transporter. For example the group of Michael Blackman published in 2021 (PMID 33762339) that another putative amino acid transporter (PF3D7_1231400) is actually a regulator of calcium mobilization. Not to say this AAT1 is doing something similar, but to point out that annotations without experimental evidence can be misleading. The authors could test their model by examining levels of amino acids or peptides in isogenic lines expressing different AAT1 alleles, using methodology developed by several labs including the recent report from the Lanzer lab that provided evidence for CRT transporting oligopeptides derived from hemoglobin (PMID 35867395). If the authors have evidence that AAT1 is actually involved in amino acid or oligopeptide transport then they should show it.

There is abundant evidence that PfAAT1 may act as an amino acid transporter (Martin 2020 (Biological reviews of the Cambridge Philosophical Society 95, 305-332, (2020))). However, we agree with the reviewer that more work is warranted to understand the function of PfAAT1, and we are confident the model we have presented will help to guide this work. We now cite the evidence that PfAAT1 is an amino acid transporter in the revised paper to clarify this important point (lines 236-240).

Minor comments:

1. Line 108: The authors state that the AAT M522I mutation is synonymous. Should it not be non-synonymous?

Thank you for pointing this out – now corrected

2. What is the status of the non-S208L mutations in Africa? Including S313F, Q454E and K541N. The text on lines 236-7 is not clear – are those associations with F313S in Africa?

I think there is some confusion here. *Pfaat1* mutations (F313S, Q454E and K541N) are at high frequency in Asian parasites but are rare or absent in Africa. We have now updated Fig 2 to clearly show frequencies of both AAT1 and CRT haplotypes.

3. Figure 2 – Africa appears to be ~60-65% S258L and the rest is listed as “Other”. What are the other mutations? Also, can the data be broken down by region?

In Africa, the majority of PfAAT1 mutant alleles are S258L, while most of the remainder are wildtype (white segments of pie charts). Other alleles (grey segments of pie charts comprise

<1.7%). See response above and updated Fig 2, and Table S2, for a complete list of PfAAT1 and PfCRT haplotypes by location.

4. Figure 3 – it would help to designate the location of AAT1 and CRT using arrows in panels A and B. The legend refers to Fig. S1 and S2 (Lines 441-442). Are these referencing the correct figures?

We have now added arrows to mark position of both PfAAT1 and PfCRT – thanks for this suggestion. We have also added a vertical line in panel C to show the location of *pfaat1*. S1 and S2 should be S6 and S7 – this is now modified in the revised manuscript. Thanks for pointing this out.

5. Figure 4 – can the authors also indicate the location of AAT1 and CRT in panel A? Note in Figure 4 that the MUT AAT / MUT CRT have a slightly higher CQ IC₅₀ than WT AAT1 / MUT CRT. That does not agree with Figure 5B – which should be mentioned. In Figure 4D, the CQ IC₅₀ levels for each recombinant progeny (unique) should be indicated by individual points so that reader can tell how many unique clones were tested. With the large error bars for the WT/MUT and mut/mut groups and limited number of progeny tested of N=2 for the WT/MUT group, more replicates and independent unique recombinants are required. There is a lack of robustness in the data. This is essential to justify the authors' statement that AAT1 is epistatic to CRT.

We have made significant changes to clarify this figure (see also answers to Reviewer 1). We have: (i) marked the location of PfAAT1 and PfCRT on panel A; (ii) Removed the color key to avoid confusion; (iii) moved Fig 4D to a Fig S11 - the revised graph shows data from all progeny in a bar chart, and points are added to indicate the number of replicates. We examined 18 progeny, with 2-5 in each genotypic class, with 4-6 technical replicates for each genotype.

We agree that these data (now moved to Fig S11) provide only a weak argument that PfAAT1 is acting epistatically and have now removed this argument from the manuscript (see answers to reviewer 2, points 2 and 3). This can best be investigated using CRISPR editing of both *pfCRT* and *pfaat1* in future work.

The reviewer makes the important point that MUT PfAAT1/MUT PfCRT shows higher IC₅₀ than WT PfAAT1/MUT PfCRT in the progeny clone data (Fig 4D, now Fig S11) but slightly lower IC₅₀ in the CRISPR analyses (Fig 5B). We have re-analyzed the data, there is no significant difference between MUT PfAAT1/MUT PfCRT and WT PfAAT1/MUT PfCRT progeny (in the initial analysis the parental parasites were included by mistake). We agree with the reviewer that

30there is a limitation with this analysis because we only had two progeny for the WT PfAAT1/WT PfCRT. The CRISPR data is likely to be the most robust, because all parasites examined have the same genetic background. We have now noted this limitation in the text, and moved figure 4D to the supplement (Fig S11).

6. Figure S1 – can the authors explain whether the selection coefficient reflects a change in relative growth per parasite asexual blood stage cycle? It's hard to know what s means.

s is measured per parasite generation, spanning the duration of infection in the mosquito and human host. As this may vary in *P. falciparum* infections, we use estimates of 2, 4 and 6 months respectively. This is now clarified in the Figure S1 legend. We have recalculated s after changing sampling date from 2000 to 2001 – this results in minor changes to s .

7. Figure S2 – should this be 2001 and not 2000? The coloured boxes in the Key should be reordered chronologically. Thank you for pointing this out.

This should be 2001. We have now replaced 2000 with 2001 elsewhere in the manuscript and figures. The key to Fig 2 is also now reordered chronologically. Fig S1 has also been replotted and results in minor changes to the selection coefficient estimates.

8. Figure S3 – the 2001 data (not 2000?) shows linkage disequilibrium (LD) with a Chr. 7 segment downstream of CRT – has that been observed elsewhere? To validate the LD between AAT1 and CRT, the reciprocal analysis showing the R^2 between CRT K76T SNP and SNPs within Chr. 6 surrounding AAT1 should be done across the various time intervals.

A previous paper (Band et al, see main text – line 120-122) showed that the strongest interchromosomal LD is between SNPs in PfAAT1 and PfCRT. The data from Gambia are consistent with this and further show that LD declines following removal of chloroquine as first line treatment. We now plot the reciprocal patterns as suggested (now Fig. S4). 1984 is not included because neither PfAAT1 S258L nor Pf CRT K76T were segregating at this time.

9. Figure S5 – should F313S and S258L/F313S be coloured differently from each other in the PfAAT1 allele Key?

Thanks for pointing this out – now changed as suggested.

10. Figure S9 – the legend needs to be corrected when it states that the 258L mutation is more

resistant to lumefantrine (LUM). That is true for 258L/313S but not for 258L/313F, when compared with 258S/313S.

The legend for Figure S9 (now Fig S12) is now reworded to correct this.

11. Figure S13 – it is not clear how many independently generated clones for each CRISPR/Cas9 modified lines were used in the pairwise competition fitness experiments (lines 676-677). The methods mentioned that two different clones from each editing (two biological replicates) were run in technical triplicates but the graph in Figure S13 depicts three replicates. Please clarify what these replicates refer to.

This is now corrected in the methods, and mentioned in the Figure S13 (now Figure S17) legend. We used three independent edits (biological replicates) for each genotype, and ran each competition experiment with two technical replicates.

12. There is some detail missing in the Methods and Results sections on the duration of the in vitro IC50 drug response assays for CQ and other compounds. Are these 72 hr exposures?

Yes, these are 72-hour exposures – this is now noted in the methods (line 681).

Reviewer #3 (Remarks to the Author):

This paper delves into the fascinating, intertwined evolutionary trajectory of amino acid variants in two genes involved in resistance to CQ, *pfcr1* and *pfaat1*. The combination of population genetics, QTL mapping, and genome editing is impressive and comes together to show convincing evidence for the specific role of *aat1*, and for epistasis, especially between alleles in *pfaat1* in SEA. These results are of significant interest both to the malaria community as well as for the study of pathogen evolution. I do have a number of comments on the presentation of results, evidence behind some claims, and missing discussion points:

1. Although the evidence accumulates on the role of AAT1 in CQ resistance, the presentation of the narrative is odd. The set up is that the authors aim to “understand the contribution of additional parasite loci to CQ-resistance evolution” (lines 83-84), but the results begin immediately with *pfaat1* and the allele frequency trajectory of S258L in particular, without explaining how they identified this gene or allele. Only after the genome-wide results that follow does zeroing in on *aat1* make sense. Or was *pfaat1* S258L the polymorphism that had the largest change in frequency between 1984 and 2014 in the dataset? Are there any other alleles showing a similar pattern or do these two stand out? In the haplotype differentiation scan, *pfdhfr* also comes up – does the allele frequency trajectory look the same or is that different from the alleles in *pfcr1* and *pfaat1*? Either clarifying why *pfaat1* was picked up in this first analysis in particular, or that it was only looked at following other findings that zeroed in on *pfaat1*, would make the ordering more logical.

Thank you for pointing this out. As the reviewer suggested, we homed in on PfAAT1 S258L, because this SNP showed the strongest change in frequency over time. We have now added a sentence to clarify this defining point (lines 92-95).

2. The strong IBD signal at *pfaat1* and *pcr1* in 1984 before the CQ-resistant alleles were present is puzzling. The authors say there was only a synonymous variant (but this is given as M552I, isn't that nonsynonymous?), suggests there may have been previous selective sweeps and they may have targeted a regulatory variant. I did not follow this logic – would this be expected if there was a completed sweep? Or would that be the case if the selected variant was subsequently lost, and is that what they suggest happened since presumably the parasites in 1984 were susceptible to CQ?

33This is a synonymous variant (I552I) but was erroneously written as M552I in the original manuscript – now corrected in the revision. Thank you for catching this error.

We also found the strong IBD in 1984 extremely interesting, although we cannot explain this observation with current data. Chloroquine was used in this region since the 1950s, so there has certainly been time for prior selective events driven by CQ. While high-level CQ resistance driven by amino acid mutations in PfCRT spread in Africa in ~1978, it is plausible that low level CQ resistance driven by non-coding regulatory changes could have occurred between 1950 and 1984. These could have been complete or partial sweeps, that were then replaced during spread of high level CQ-R PfCRT alleles imported from SE Asia, and the spread of the PfaAT1 S258L allele as describe here. This would be analogous to the situation with antifolate resistance evolution in Africa, where low level antifolate resistance alleles (single and double mutant *dhfr*) evolved locally in several locations, but these alleles were replaced by selective sweeps driven by high level resistance alleles (triple mutant *dhfr*) imported from Asia. These are exciting and important questions that we hope will be addressed in subsequent studies. We have added some extra text to the revised manuscript (line 113-115) to further explain this.

3. Strong selection is inferred on both alleles from the allele frequency change and IBD analyses. Although it might seem obvious to the authors, an explicit description of what signatures of selection are being detected and whether these are necessarily selective sweeps would be advised.

Thank you for this suggestion. We now include a sentence stating “*That these rapid changes in allele frequency occur at pfcr1, pfaat1 and pfdhfr, but not elsewhere in the genome, provides unambiguous evidence for strong directional selection.*” (lines 103-105) to better explain our conclusions.

4. Referencing which signatures and observations have been made previously should be included. Some of this is provided in the Discussion, but which findings are new to this paper vs. being re-iterated here should be indicated in the results. For instance, the IBD signal was reported in the isoRelate paper already (Henden et al. 2018) and several studies have found strong signatures of positive selection on pfaat1. Additionally, context in the discussion for what exactly this paper resolves and what remains unexplained would be helpful to reach a wider audience.

At the suggestion of Reviewer 2 (question 1) we conducted a more thorough investigation of IBD in different populations from public databases. This expands on previous published analyses, and serves to bring all the population genomic data together in one place. We have

acknowledged the previous work on this topic, including work by one of us (Amambua-Ngwa et al, Science 2019), and the Henden et al. PLoS Genetics 2019 and Carrasquilla et al. Biorxiv 2022 papers in the manuscript text (line 115-118) and the new Suppl Fig S3 legend.

5. The claim that the 258L allele has an independent origin in Asia and Africa should be better supported. What is truly the evidence for this? The haplotype networks do not seem convincing as presented. Fig. S5 seems to be the main evidence for this, where 258L is found in several parts of the tree. However, this is a large region (50kb) that is being considered and it is not obvious that 2 origins fits better than 1. Additionally, it is not clear which parts of Fig. 2B the labels of “Africa” and “Asia” refer to. To make the claim for convergent evolution a stronger test is needed. I also could not tell the difference between the colors used for F313S and S258L.F313S.

Thank you for the suggestion. We have made modifications to clarify both Fig 2 and Fig S5 (now Fig S7):

- For Fig 2B (now Fig 2C), we removed the labels of “Africa” and “Asia”. Instead, we added a Fig 2D, which is colored by geographical location. In Fig 2D, majority haplotypes from the same region (Asia or Africa) were clustered together, indicating independent origin of *pfaat1* alleles. Henden et al. PLoS Genetics 2019 reached the same conclusion for the chr 6 region using an identity-by-descent analysis of parasites from global locations. We have also modified the manuscript to make this claim clearer (line 135-137). However, we note that this observation is not a central feature of this manuscript.
- For Fig S5 (now Fig S4), we modified the color code to make the figure clearer.

As for the 50kb region, it was selected based on the extended haplotype homozygosity (EHH) distance around the *pfaat1* gene (Fig S5).

6. Why does Fig. 3B look very different from Fig. 3C and from Fig. S6 and S7? In 3B the chr6 QTL does not appear to reach much higher than the significance threshold, not up to 50 like it does in the other plots. Also, it is not obvious that AAT is at the peak of the QTL in Fig. 3C – could there also be a zoom in to the QTL evidence together with the genes plot?

Thank you for pointing this out.

- Fig 3B (as well as Fig 3A) only shows G' values from BSA experiments with 48h CQ treatment, drug concentration ranges from 50-250 nM, and collected at day 4.

- Fig 3C is fine mapping of the chr. 6 QTL, which includes BSA results from both 48h (day 4 collection) and 96h (day 5 collection) CQ treatment, but with only the highest CQ treatment (250nM).
- Fig S6 and S7 (now Fig S8 and S9) are BSA results for all the BSA experiments.

We have now modified the figure legends to clarify this information.

For Fig 3C, we now added a red dashed line to indicate the location of *pfaat1* gene, which is at the peak of chr. 6 QTL.

7. Is the epistasis shown in Fig. 4D sufficient to explain both the strong LD between *pfert* and *pfaat1* in natural populations (Fig. S3) and/or in the cross? I was surprised that the CRISPR and growth analysis is only for polymorphisms in *pfaat1*. That's fine, but a discussion of how these results clarify or explain the inter-chromosomal LD is warranted, or whether it remains to be determined.

The strongest evidence for epistasis comes from the strong association between PfAAT1 and PfCRT SNPs in our cross progeny and in natural populations. The evidence for epistasis in the IC₅₀ data from the progeny is quite weak because we only recovered only two progeny clones carrying AAT1-WT/CRT-MUT alleles (see also answers to Reviewer 2). In the revised text, we describe the limitations of the analysis in Fig 4D (now Fig S11), and de-emphasize the conclusion that epistasis dictates the level of resistance.

We now have funding to conduct CRISPR work to edit both *pfert* and *pfaat1* in parasites from both Africa and Asia. This will allow us to directly examine epistasis in parasites from different genetic backgrounds, but represents a substantial body of additional work. We will be conducting this work in the coming years and intend to publish this work in subsequent papers.

8. The level of detail for sequencing and variant calling is very different between the longitudinal samples and that of BSA, with the latter much more complete. There needs to be better description for the longitudinal samples than “mapping on the Pf reference genome employed bwa and BAM files and were optimized with Picard tools”. Bam is a file format not something that is employed, and Picard is a tool so this gives no information on what was “optimized”. Additionally, the filtering is not fully described and why almost half the samples were excluded is not explained. The only two filters listed are for genotype missingness < 10% and MAF > 2%. The MAF filter is on variants, is the genotype missingness per individual? Why were so many samples filtered out? Were indels excluded?

We now have modified the sequencing and genotyping section (line 525-549) to show the analysis pipelines in detail.

Minor

1. In the introduction, it is mentioned that other variants in MDR1 and within CRT have been reported to modulate CQ resistance. Were those variants not identified in these analyses and does that suggest they were not major players? Given the goal

The MDR variants segregating in this cross (Ala750Thr and Phe1226Tyr) are listed in Table S3. These are not known to impact CQ resistance.

2. Some inconsistencies in the terminology referring to the variants, such as referring to amino acid polymorphisms as SNPs (“the K76T SNP”), whether referring to the polymorphism or the allele (“K76T” vs “76T”), and then later as “wt/mut” without defining. A consistent usage would be helpful.

We have now made this consistent in the text. Where WT and MUT are used in Fig. 4, these abbreviations are explained in the legend.

3. Some inconsistencies in the use of the word “linkage” including “functional linkage” (what is meant by that beyond a ‘functional relationship’?) and using “linkage” when “linkage disequilibrium” appears to be meant.

Now corrected in the revised text

4. It is stated that “all IC50 levels were below levels of clinical significance” for the drugs tested besides CQ. What are IC50 levels for clinical significance for these drugs – could that be indicated in figure S9?

This is now defined (with citations) in the figure legend (now Fig S12).

5. Were any off-target sequence changes detected from the whole genome sequencing of CRISPR parasites? It just says this was done but does not describe what was found (that I found).

We were not able to find any SNP or indel changes between the original NHP4026 and any CRISPR edited parasites other than the target and shield mutations. We have now included this result in the revised manuscript (line 702-704).

6. Should pfaat1 be in parentheses in the title, I was not clear whether “amino acid transporter” is the long form of the gene name, or a description of what pfaat1 is.

Thanks for pointing this out. We now modified the title to “An amino acid transporter AAT1 plays a pivotal role in chloroquine resistance evolution in malaria parasites”.

7. A key to the columns in supplementary tables would be useful.

We have now included a key to columns for each supplementary table.

8. In the legend for Fig S2, the years are not in order and it makes it more difficult to interpret

We now modified the legend for Fig S2 to make sure the years are in order.

9. Why is Fig. S8 referenced in line 140 – should this reference a different figure?

Fig S8 (now Fig S10) shows the change in allele frequency during the BSA experiments, so was cited correctly. In the revised manuscript we cite Fig S8-10) because all three supplementary figures are relevant to this point.

Reviewer #4 (Remarks to the Author):

In this manuscript, the authors used an array of cutting-edge technologies (population genomics, genetic cross, and CRISPR-based gene editing) to elucidate the compensatory effect of the amino acid transporter Pfaat1 S258L mutation on the evolution of chloroquine (CQ) resistance. This study demonstrated the epistatic effect of pfaat1 on pfcr1. While the S258L mutation significantly boosted the CQ resistance, it also reduced the fitness of the parasites. Another mutation that is fixed in SE Asia, pfaat1-F313S, although reduced the CQ resistance level, it restored fitness of the parasite. This study provided evidence on the significance of compensatory mutations in the evolution of antimalarial drug resistance (like the ones occurring with the PfK13 mutations in SE Asia).

1. The genomics work with the longitudinal samples provided a direct view of the parallel selection of both the pfcr1 and pfaat1 S258L haplotypes. Fig S3 showed interchromosomal LD between these two loci. I would like to see if pfcr1 and pfaat1 surrounding regions can be zoomed in to show the size of the selection valleys (changes in heterozygosity in these genes and flanking regions)? It is also puzzling that dhfr was also selected during this region, why?

We have now expanded Fig. S3 (now Fig S4, see also response to reviewer 1). We now show plots of EHH surrounding PfAAT1 S258L and PfCRT K76T for years 1984, 2001, 2008 and 2014 (Fig S5). Evidence for reduced diversity in the region around *pfcr1* are well established. However, these plots show additional evidence for selection on *pfaat1* in these samples.

Regarding *dhfr* – selection on this locus is expected. Antifolate drugs were widely used in Africa, as a replacement for chloroquine. Mutations in *dhfr* confer resistance to pyrimethamine, and selective sweeps at this locus have been extensively documented. This locus provides a positive control for our methods.

It is odd that figure 1 included a threshold line based on a significance value at $-\log(p\text{-value})=3$, but then wrote genes out for only the "top 1% of p values". Please specify in the figure legend and justification. I would also consider moving the red lines on their graph to 5 since that is their cutoff for inclusion as a "region of interest".

We have now done this as suggested.

392. The different effects of *aat1* mutations (S258L and F313S) on CQR and fitness were done with the SE Asian parasite NHP4026. The results would deserve evaluations in an African parasite line to prove that the observed effects were not due to the Asian parasite's genetic background. Could you also speculate on why the F313S mutation has not evolved in the African parasites, if it offers fitness compensation?

The NHP4026 parasite used for editing carries the CVIET *pfcr1* haplotype common in Asia and Africa (See also our reply to reviewer 1) as expected. This makes it a suitable parasite for gene editing studies. We agree with the reviewer that gene editing of *pfat1* in African parasite genetic backgrounds will be valuable and we now have funding to do this work. We plan to conduct this work over the next few years and will report these data in subsequent papers. However, this is beyond the scope of the current manuscript.

We have now provided possible explanations for why the PfAAT1 F313S mutation has not evolved in African parasites in the revised text (line 277-291). In brief, SE Asian *P. falciparum* still faces some exposure to CQ, due to treatment of *P. vivax* infections with CQ. Therefore, in Asia, PfAAT1 alleles were selected for compensation to retain the CQ resistance phenotype. In contrast in Africa, where there is no CQ pressure, wild-type parasites evolve to replace the resistant PfCRT K76T allele and PfAAT1 S258L allele (e.g., Nwakanma et al. JID 2014).

3. it would be very useful if the mechanistic experiments performed in yeast for the malaria parasite could be replicated in malaria parasites using their CRISPR mutants.

We now have funding to conduct this work over the next few years and will report these data in subsequent papers. However, this extensive and important work is beyond the scope of the current manuscript.

Minor comments:

I'm assuming they excluded 1990 in the iR analysis because the sample size was small, but it may be worth explicitly stating the criteria of inclusion. For example, "we performed this analysis for all years with sample size > 30" OR "we performed this analysis for all years, but 1990 yielded no significant results, likely because of sample size". Those are two pretty different statements, in my opinion, so readers shouldn't have to guess which is true.

We excluded the 1990 sample from the iR analysis because the sample size was quite low (n=13). This information is now provided in the methods, and the legend to Fig. 1

40Line 202 – 258S (not underlined)/313S (underlined)

We underlined derived alleles, but not ancestral alleles – we have made this consistent throughout the manuscript.

203 – a strong epistatic interaction or strong epistatic interactions

Thanks for pointing this out. Now modified.

Decision Letter, first revision:

Message: 26th January 2023

Dear Professor Anderson,

Thank you for your patience while your manuscript "An amino acid transporter AAT1 plays a pivotal role in chloroquine resistance evolution in malaria parasites" was under peer-review at Nature Microbiology. It has now been seen by 2 of the original referees, whose expertise and comments you will find at the of this email. You will see from their comments below that while they find your work of interest, some important points are raised. We are very interested in the possibility of publishing your study in Nature Microbiology, but would like to consider your response to these concerns in the form of a revised manuscript before we make a final decision on publication.

In particular, you will see that while referee #4 is satisfied with the revised version, referee #2 still raises some important points to be addressed through text and figure changes. Importantly, some of the statements and conclusions need to be toned down.

If you have not done so already please begin to revise your manuscript so that it conforms to our Article format instructions at <http://www.nature.com/nmicrobiol/info/final-submission/>

41The usual length limit for a Nature Microbiology Article is six display items (figures or tables) and 4,000 words. We have some flexibility, and can allow a revised manuscript at 4,500 words, but please consider this a firm upper limit. There is a trade-off of ~250 words per display item, so if you need more space, you could move a Figure or Table to Supplementary Information.

Some reduction could be achieved by focusing any introductory material and moving it to the start of your opening 'bold' paragraph, whose function is to outline the background to your work, describe in a sentence your new observations, and explain your main conclusions. The discussion should also be limited. Methods should be described in a separate section following the discussion, we do not place a word limit on Methods.

Nature Microbiology titles should give a sense of the main new findings of a manuscript, and should not contain punctuation. Please keep in mind that we strongly discourage active verbs in titles, and that they should ideally fit within 90 characters each (including spaces).

Please include a data availability statement as a separate section after Methods but before references, under the heading "Data Availability". This section should inform readers about the availability of the data used to support the conclusions of your study. This information includes accession codes to public repositories (data banks for protein, DNA or RNA sequences, microarray, proteomics data etc...), references to source data published alongside the paper, unique identifiers such as URLs to data repository entries, or data set DOIs, and any other statement about data availability. At a minimum, you should include the following statement: "The data that support the findings of this study are available from the corresponding author upon request", mentioning any restrictions on availability. If DOIs are provided, we also strongly encourage including these in the Reference list (authors, title, publisher (repository name), identifier, year). For more guidance on how to write this section please see: <http://www.nature.com/authors/policies/data/data-availability-statements-data-citations.pdf>

To improve the accessibility of your paper to readers from other research areas, please pay particular attention to the wording of the paper's opening bold paragraph, which serves both as an introduction and as a brief, non-technical summary in about 150 words. If, however, you require one or two extra sentences to explain your work clearly, please include them even if the paragraph is over-length as a result. The opening paragraph should not contain references. Because scientists from other sub-disciplines will be interested in your results and their implications, it is important to explain essential but

specialised terms concisely. We suggest you show your summary paragraph to colleagues in other fields to uncover any problematic concepts.

If your paper is accepted for publication, we will edit your display items electronically so they conform to our house style and will reproduce clearly in print. If necessary, we will re-size figures to fit single or double column width. If your figures contain several parts, the parts should form a neat rectangle when assembled. Choosing the right electronic format at this stage will speed up the processing of your paper and give the best possible results in print. We would like the figures to be supplied as vector files - EPS, PDF, AI or postscript (PS) file formats (not raster or bitmap files), preferably generated with vector-graphics software (Adobe Illustrator for example). Please try to ensure that all figures are non-flattened and fully editable. All images should be at least 300 dpi resolution (when figures are scaled to approximately the size that they are to be printed at) and in RGB colour format. Please do not submit Jpeg or flattened TIFF files. Please see also 'Guidelines for Electronic Submission of Figures' at the end of this letter for further detail.

Figure legends must provide a brief description of the figure and the symbols used, within 350 words, including definitions of any error bars employed in the figures.

When submitting the revised version of your manuscript, please pay close attention to our [href="https://www.nature.com/nature-research/editorial-policies/image-integrity">Digital Image Integrity Guidelines](https://www.nature.com/nature-research/editorial-policies/image-integrity) and to the following points below:

Please include a statement before the acknowledgements naming the author to whom correspondence and requests for materials should be addressed.

Finally, we require authors to include a statement of their individual contributions to the paper -- such as experimental work, project planning, data analysis, etc. -- immediately after the acknowledgements. The statement should be short, and refer to authors by their initials. For details please see the Authorship section of our joint Editorial policies at http://www.nature.com/authors/editorial_policies/authorship.html

- * include a point-by-point response to any editorial suggestions and to our referees. Please include your response to the editorial suggestions in your cover letter, and please upload

43your response to the referees as a separate document.

* please also submit a version of the manuscript with tracked-changes

* ensure it complies with our format requirements for Letters as set out in our guide to authors at www.nature.com/nmicrobiol/info/gta/

* state in a cover note the length of the text, methods and legends; the number of references; number and estimated final size of figures and tables

* resubmit electronically if possible using the link below to access your home page:

[redacted]

*This url links to your confidential homepage and associated information about manuscripts you may have submitted or be reviewing for us. If you wish to forward this e-mail to co-authors, please delete this link to your homepage first.

Please ensure that all correspondence is marked with your Nature Microbiology reference number in the subject line.

Nature Microbiology is committed to improving transparency in authorship. As part of our efforts in this direction, we are now requesting that all authors identified as 'corresponding author' on published papers create and link their Open Researcher and Contributor Identifier (ORCID) with their account on the Manuscript Tracking System (MTS), prior to acceptance. This applies to primary research papers only. ORCID helps the scientific community achieve unambiguous attribution of all scholarly contributions. You can create and link your ORCID from the home page of the MTS by clicking on 'Modify my Springer Nature account'. For more information please visit www.springernature.com/orcid.

We hope to receive your revised paper within three weeks. If you cannot send it within this time, please let us know.

Yours sincerely,
[redacted]

Reviewer Expertise:

Referee #2: Plasmodium biology, drug resistance

Referee #4: Plasmodium biology, antimalarial drugs

Reviewers Comments:

Reviewer #2 (Remarks to the Author):

In this resubmission by Nwa et al., the authors have made substantial positive improvements to the manuscript including additional haplotypic and IBD analysis across Africa and other regions, incorporating new Figures S3-S6, and rewriting and clarifying points in the main results and methods. The convergence of population genomic and genetic cross data is an exciting data set that clearly shows some selective pressure on the AAT1 region and provides evidence that mutations in this gene can contribute to modulating chloroquine (CQ) resistance and parasite fitness. There is some compelling longitudinal evidence that the S258L mutation in AAT1 in west Africa has evolved in parallel with mutations in PfCRT. The latter protein is known to be the primary driver of CQ resistance via a gain of drug efflux. The study raises a number of interesting questions that will spur much more research in the field.

I definitely support publication of this study, which I think is an important milestone for the field. There are, however, several changes that I think are vital to make before the manuscript is ready for publication. These are detailed in my lengthy comments listed below. I apologize for some overlap between them as it took multiple rereads here to work through this large and complex dataset. Below is a list of my primary concerns:

1. The genetic cross data in Figure 4 does not fully match the gene editing data in Figure 5. The CQ selection of the bulk progeny populations in Figure 4a shows enrichment for the mutant (MUT) pfCRT allele from the resistant parent NHP4026 (that expresses the "Dd2" PfCRT haplotype with 8 mutations separating it from the wild-type allele present in 3D7), with most of those progeny also carrying MUT AAT1 (carrying the S258L and F313S mutations). Those data are consistent with prior knowledge that MUT PfCRT has a substantial fitness cost. Only two MUT PfCRT progeny carry wild-type (WT AAT1 (i.e. S258 and F313), despite the WT allele being present in ~40% of the progeny recovered without drug pressure. That would suggest that MUT AAT1 is contributing to CQ resistance, or is epistatically involved with MUT PfCRT presumably by compensating for a physiologic growth defect. The NHP4026 gene editing data, however, shows that MUT AAT1 has a CQ IC50 value slightly lower than WT AAT1, which would suggest that MUT AAT1 does not augment CQ resistance (CQR). Similarly, MUT AAT1 has a substantially lower fitness cost than WT AAT1 in that background, arguing against the idea that MUT AAT1 is being selected by CQ pressure in the progeny because it (epistatically) enhances the growth of MUT PfCRT progeny. This should be clearly spelled out. Figure S11 is similar in showing that MUT PfCRT progeny have no significant differences in mean CQ IC50 values between the two progeny with WT AAT1 and the six phenotyped MUT AAT1 progeny, again suggesting that MUT AAT1 is not driving the enrichment for CQ-selected progeny that inherited that region of chromosome 6. One alternative interpretation of the data is that the chromosome 6 segment that is enriched from the NHP4026 parent in the CQ-selected progeny is close to but not actually AAT1. That would be consistent with a genetic and population genomic signal that tracks to AAT1 as a nearby molecular marker. The authors need to point out this disconnect between the gene editing and CQ bulk selection studies and also evoke the alternative hypothesis that AAT1 may be a marker of a region that co-evolved with CQR.

45

2. The authors repeatedly state that AAT1 is an amino acid transporter based largely on an earlier yeast expression study (ref. 26 by Tindall et al., 2018, Scientific Reports; the senior author S. Avery is a coauthor on this present submission). In that study the amino acid tryptophan was found to partially inhibit quinine accumulation in yeast cells expressing AAT1. Inferences based on distant sequence homology and initial data in a heterologous expression system should be presented with caution. As an example, a recent publication clearly showed that a protein formerly annotated as a separate amino acid transporter in *P. falciparum* is actually essential for calcium mobilization and serves as a crucial link between calcium signaling and protein kinase G function that is required for parasite egress from infected red blood cells (PMID 33762339). As the authors know, expression and functional characterization of malarial proteins in yeast can often be problematic because of their highly divergent AT contents and codon usages. The tryptophan inhibition data do not in my opinion provide sufficient experimental evidence of amino acid transport. I would strongly recommend that the authors modify their text to consistently refer to AAT1 as a putative amino acid transporter, including in the title. Of note, Tindall et al. also consistently refer to this as a putative amino acid transporter. Also, there is also only preliminary evidence that this protein can transport CQ, again based on yeast data.

3. As best I know, there are no amino acid or peptide metabolomics data or drug accumulation studies focusing on AAT1 with isogenic parasite lines that would support their model that AAT1 can transport amino acids or drugs in parasites. As such, their Figure 6 model is, in my opinion, too speculative and should be moved to the supplement. Additional points about Figure 6 are raised below.

4. The authors have yet to experimentally address an important caveat of the current study, which is that they have not edited AAT1 in an African line with a regional, non-Dd2 PfCRT haplotype to prove the co-evolution and functional relationship between AAT1 and PfCRT, especially to validate their proposal that the AAT1 S258L mutation was required for the evolution of PfCRT K76T in African backgrounds. Other reviewers also pointed this out and requested that the same gene editing experiment be performed. The authors indicate that they recently obtained funding to conduct this work and that it will be the subject of a future study. The authors nonetheless need to explicitly address this limitation. It is established that African and Asian PfCRT haplotypes differ in important ways, namely at residues 326 and 356, and so stating that both regions share the common CVIET haplotype is only part of the more complex story here in terms of how PfCRT haplotypes differ regionally. I acknowledge the vast amount of high-quality work already performed for this study and thus am not insisting that the paper be put on hold until those studies are complete. At a minimum, however, the authors must acknowledge that further gene editing studies in African parasites with local haplotypes are necessary to validate that the AAT1 S258L mutation was required for the evolution of mutant K76T-containing PfCRT in African backgrounds. They should also cite prior studies that document the different PfCRT haplotypes present in Asian versus African parasites (PMID 26208441, 31040246). Of note, the PfCRT Dd2 haplotype studied herein (in NHP4026 from Southeast Asia) was not detected at all in an earlier analysis of nearly 800 African *P. falciparum* genomes (PMID 31040246).

5. In light of the comments raised above, I feel the authors have yet to sufficiently prove that "the amino acid transporter AAT1 plays a pivotal role in chloroquine resistance

46

evolution", as is currently stated in their title. I would recommend a more circumspect title, e.g. by saying "Genomic and genetic evidence implicates (or "implicating") the putative amino acid transporter AAT1 in the evolution of Plasmodium falciparum chloroquine resistance".

There also remain author responses to three of my earlier major comments in which I have additional comments that are highlighted below (the authors' replies are listed in italics). Below these I list additional comments.

Earlier major comment 1: The authors present an identity-by-descent (IBD) analysis to demonstrate the strong co-selection of mutations in AAT1 and the CRT CQR marker mutation K76T in a set of 321 isolates from The Gambia, sampled from 1984 to 2014. It is not clear why this analysis has been limited to only samples from The Gambia when the MalariaGen Pf6 dataset contains more than 3,000 samples from various regions in Africa (note: researchers leading that effort are already co-authors). It is unclear why the authors conducted their longitudinal analysis solely on The Gambia – perhaps because they had access to samples from before CQR spread there? There are countries such as Mali where CQ use was discontinued and CRT reverted to wild-type (WT, i.e. K76) and it would be interesting to know the trajectory of AAT mutations in those areas. If AAT1 and CRT were co-evolving, as is proposed, can the authors explain why the prevalence of mutant CRT K76T does not correlate with the proportions of AAT1 S258L between the various continents and African regions (Figure 2A)? Is that association holding only in West Africa?

Authors' reply: The reviewer is correct – we focused on longitudinal samples from the Gambia because we (Alfred Amambua Ngwa, David Conway, Umberto D'Allessandro) had access to longitudinal samples collected from between 1984 and 2014 from the Gambia. Such longitudinal samples are not available in the PF6 database from other regions of Africa. We certainly agree that parallel analyses in Malawi would be extremely valuable, but we do not have access to these samples or sequence data.

With regard to correlations between PfaAT1 and PfcRT mutations. We thank the reviewer for the suggestion to examine PfcRT haplotypes and PfaAT1 evolution in more depth across Africa. In the light of the reviewer's comments, we have expanded our analysis to examine distribution of PfaAT1 and PfcRT alleles across Africa. This new fine-grained analysis across Africa clearly shows that PfcRT K76T correlates strongly with the proportions of PfaAT1 S258L in West Africa ($R^2 = 0.65$, $p=0.0017$) and across all African populations ($R^2 = 0.44$, $p=0.0021$). This analysis further strengthens the argument for co-evolution and epistasis between these two genes. We have added this to the main text (lines 126-130); the analysis is provided as Supplementary Fig S6.

New Comment 1.1: It is excellent to see this detailed in-depth analysis of AAT and PfcRT mutations across Africa (Figure S6). The correlation in west Africa is striking and provides important data to link the evolution of both sets of polymorphisms. It is interesting that the East African isolates (Kenya and Tanzania) did not show a clear association of AAT S258L with PfcRT CVIET. Instead, parasites in these countries were mostly PfcRT WT but yet were almost 100% S258L for AAT1, suggesting a fixation of this mutant AAT1 allele. This is rather surprising given that the S258L mutation alone made the *aat1*-edited mutant *pfcr*t NHP4026 parasites less fit (Figure 5c). The authors should explicitly address this. Also,

47

given that the S258/F313 WT parasites were shown to be fitter than the S258L/F313S or S258L/F313 parasites (Figure 5), can the authors provide an explanation as to why we do not observe higher frequencies for the S258/F313 WT AAT1 in clinical isolates across Asia and Africa, especially after the removal of CQ usage?

Earlier major comment 7. What is the NHP4026 CRT haplotype? How does this compare to CRT haplotypes in The Gambia or other African regions? Among CRT K76T mutants, several haplotypes of CRT exist and their prevalence markedly differs between geographical regions (see PMID 33824913). It is rather surprising that the authors have not mentioned this when the AAT1 haplotypes are well documented in this manuscript. Also, it would be important to test the impact of the AAT1 Southeast Asian mutations S258L+F313S on CQ IC50 and fitness in 3D7 as well as other African parasites from The Gambia and Mali. The manuscript would be strengthened by adding data from AAT1 CRISPR/Cas9 editing on African parasite lines. This would help test the authors' main hypothesis that AAT1 plays a central role in CQR evolution, especially since the co-selection was observed in parasites from The Gambia.

Authors' reply: We have now expanded the population genomic analysis to examine the distribution of pfCRT haplotypes in relation to PfaAT1 mutations (see revised Fig 2 and Fig S6).

The NHP4026 parasite used for editing carries the CVIET pfCRT haplotype common in Asia and Africa (see also our reply to reviewer 1), as expected. This makes it a suitable parasite for gene editing studies. We agree with the reviewer that gene editing of pfaat1 in African parasite genetic backgrounds will be valuable, a point that has led to a recently funded proposal to conduct this work. We will conduct this work over the next few years and will report these data in subsequent papers. However, this is beyond the scope of the current manuscript.

New comment 7.1: The revised Figure 2 does not indicate the full PfCRT haplotypes (eg. Dd2, Cam734, 3D7, GB4, Cam783) present in different regions. Earlier analysis of the MalariaGEN Pf3K data, assembled from downloaded genome data, found that African and Asian PfCRT haplotypes show important differences. The Dd2 haplotype present in the current Asian parent NHP4026 was absent from African parasites (out of 783 genomes sequenced). Instead, African mutant PfCRT haplotypes were predominantly GB4 or Cam783, which differ at positions 326 and/or 356 (PMID 31040246). While these parasites share the CVIET haplotype at positions 72-76, it is not correct to state that Dd2 was introduced into Africa and swept across the region, in contrast to the statement from Reviewer 1. Either Dd2 migrated and later "devolved" to simpler haplotypes that lost the mutations at positions 326 and/or 356, or the alternative GB4 or Cam783 allele(s) spread from Asia to Africa. The authors should clarify their text accordingly. Ideally, the authors could present a revised PfCRT haplotype analysis for African strains based on the latest reported Pf7K genome database, if that analysis could be conducted quickly, as this would provide a more thorough analysis of associations between AAT1 and PfCRT. If not possible, at least the authors should acknowledge the different haplotypes and state that their gene editing did not examine African haplotypes.

Earlier major comment 9. The authors should also further clarify their gene editing data.

48I'm unclear whether the 258L/313S mutant described in Figure 5 is the NHP4026 parent edited to express the same mutations (with silent mutations to prevent cleavage involving the PAM site) or whether that was NHP4026 and the only gene-edited mutants were the other three combinations. Ideally the best is to have gene-edited all four combinations, and better yet would be to have a control with the silent mutations introduced to prevent re-cleavage of the edited locus.

Authors' reply: The PfaAT1 S258L/F313S mutant described in Figure 5 is NHP4026. This is detailed in the methods (line 692-693), and in Fig 5A, and is now further clarified in the Fig 5 legend. We did not conduct control edits with shield mutations only – this is not common practice in malaria work using CRISPR/Cas9, because malaria lacks the error-prone non-homologous end joining repair system that generates off-target changes in other eukaryotes. However, we bolstered our approach by conducting three independent edits (edits recovered from independent flasks) for each genotype constructed (11 CRISPR edits in total). Each of these show very similar results for both IC50 and fitness assays, providing strong evidence that the phenotypes observed do not result from off target mutations.

New comment 9.1: The 11 independent gene edits performed are biological repeats for 3 separate editing events (two single mutants and the double wild-type reversion) in a single Asian parasite. As stated elsewhere, the authors should acknowledge that additional gene editing studies will be required to confirm these data in other genetic backgrounds, including ones with African PfcRT haplotypes. Also, while I agree that binding site mutants are not essential because of the lack of non-homologous end joining, these controls remain quite standard practice. The authors should add a note in their legend that a binding-site control mutant was not generated, on the basis that *P. falciparum* lacks error-prone non-homologous end joining (PMID 25184562).

Please see below a series of additional comments. Some of these overlap with comments raised above but here relate to specific lines.

6. Line 41-2: "mutations in [pfcrt]...confer CQ resistance in Plasmodium falciparum, but typically affect parasite fitness". This statement suggests that fitness is the main phenotype, not CQ resistance, which is not accurate. I would suggest rewording to ""mutations in [pfcrt]... confer CQ resistance in Plasmodium falciparum, and can also affect parasite fitness".

7. Line 49: To be completely transparent, this should be revised to say "Parasite genetic crosses then identified a chromosome 6 quantitative trait locus containing pfaat1 (harboring the S258L/F313S mutations) that is selected by CQ treatment." The next sentence should read "Gene editing demonstrated that pfaat1 S258L with F313 potentiates CQ-resistance..."

8. Fig S3 (cited on line 118): The authors observe an additional chromosome 10 peak in West Africa in the IBD analysis that is not observed in the West African Gambian analysis in Figure 1. As this peak is not observed in Asia, could this also be a potential region that explains differences in PfcRT haplotype prevalence in Asia vs Africa? The authors should refer to this peak in the Figure S3 legend and if possible, provide any more information about this region.

49

9. Line 150: 3D7 is stated to be a West African parasite but this is up for debate. Earlier suggestions were based on early partial microsatellite data. The most comprehensive analysis I know of was from Aline Uwimana and colleagues (PMID 3274827) who analyzed >300 genomes collected worldwide and found 3D7 to cluster next to parasites from Rwanda (east-central Africa). Maybe just state "African parasite" and add that citation.

10. Line 193: "...is consistent with the pfAAT1 QTL being driven by parasite fitness in our genetic crosses." This is a strong statement that needs to be moderated. As stated above, there was a discrepancy in the fitness data from the genetic cross results. In Figure 4a, it was shown that NHP4026 MUT AAT1 (S258L/F313S) was selected for over the WT S258/F313, along with mutant PfCRT (CVIET) in the presence of CQ, suggesting that the MUT S258L/F313S AAT1 may be involved in CQR and/or compensate for mutant pfCRT-mediated fitness defects. However, in Figure 5, they show that when comparing their PfCRT mutant NHP4026 lines that expressed MUT S258L/F313S or WT S258/F313 AAT1, the WT (that was selected against under CQ pressure) was actually more fit and also slightly more CQR than S258L/F313S. These results are contradictory to those observed in Figure 3. In addition to their proposed explanation on lines 232-235, the authors could state that other genes in the chromosome 6 QTL segment may play a role and also that additional genetic backgrounds need to be tested to further examine these associations.

11. Lines 195-208: The text reads "This revealed a highly significant impact of the S258L mutation, which increased CQ IC50 1.5-fold, and a more moderate but significant impact of F313S and the double mutation (S258L/F313S) (Fig. 5b, Supplementary Table 8). The observation that 258L shows an elevated IC50 only in combination with the ancestral 313F allele reveals an epistatic interaction between these amino acid variants (Fig 5B)." This doesn't fully correspond to the data shown in Figure 5. Those data show that the greatest increase in CQ IC50 is with S258L + F313. By comparison, S258L + F313S is 1.5 fold less resistant and has an even slightly lower IC50 than the WT S258 + F313. Those data would argue that only the S258L + F313 combination is able to augment CQR mediated by mutant PfCRT and that F313 is key. Indeed, S258L + F313S has a lower CQ IC50 than S258 + F313S, arguing that in the presence of the F313S mutation the S258L mutation makes parasites less CQ-resistant. The explanation as presently stated can easily be misinterpreted and needs revision. The title of that paragraph should also be changed, for example to say "Gene editing evidence that polymorphisms in AAT1 can modulate CQ resistance in mutant pfCRT parasites".

12. Coming back to Figures 5 and 6: If AAT1 were actually a mediator of fitness in the mutant PfCRT background, and WT AAT1 + MUT PfCRT was shown herein to be the most fit (Figure 5c), then this haplotype would have been expected to take over, not S258L AAT1 as in the Gambia or haplotypes containing F313S as in Southeast Asia (SEA). Further work is clearly required to further disentangle these relationships. For now, I would advocate to place Figure 6 in the supplement given the preliminary basis of this model. Also, the model needs to include a fourth panel with WT AAT1 + MUT CRT, which is presently lacking.

13. Lines 227-8: "However, S258L carries a high fitness cost that in SEA parasites was likely mitigated by addition of the compensatory substitution, F313S." However, it appears that a very small minority of the SEA parasites have S258L only, while the majority of the

50population has F313S + another mutation, including S258L, Q454E, or K541N. Could these data suggest that F313S is the predominant AAT1 haplotype in SEA, and that other mutations, including S258L, arose on this background?

14. Line 239: "CQ uptake can be competitively inhibited..." This sentence needs to be modified to show that the data were produced by another group. This can be addressed by changing this sentence to "CQ uptake was previously shown to be competitively inhibited..."

15. Line 244: The authors use citation 26 to state that "expression of another amino acid variant (T162E)...restores cell growth in the presence of CQ.. Together, these new and published results suggest that yeast expression of pfaat1 mutations impact resistance and fitness by altering the rates of amino acid and CQ transport." However, this is not an accurate interpretation of the data in that article, which provides evidence that expression of the T162E AAT1 mutant resulted in increased yeast cell growth in the presence of an exceptionally high concentration of CQ (1 mM). The sentence as it stands needs to be modified to remove any reference to fitness and amino acid or CQ transport, as the results presented therein related only to growth. A revised sentence could read as: "expression of another amino acid variant (T162E)...restores cell growth in the presence of 1 mM CQ". The next sentence needs to be deleted as neither the Tindall et al. paper nor Figure S13 present data on fitness or amino acid or CQ transport. Also, it would have been preferable to include the T162E mutant as a control in their present study to see how those data matched the prior publication, especially as the current study used 5 mM CQ and the prior study used 1 mM CQ.

16. Lines 287-8: "restoration of fitness by F313S may help to explain retention of CQ-resistant pfcrT K76T alleles in SEA." This is very speculative, as there is no temporal evidence that F313S "restored" fitness costs mediated by S258L. In fact, the evidence points more to the possibility of F313S being the more dominant, ancestral haplotype in SEA, upon which other mutations arose. The text could be modified to state: "We speculate that restoration of fitness by F313S may help to explain retention of CQ-resistant PfCRT K76T alleles in SEA".

17. Line 261-2: The text "we find clear evidence that a second locus, pfaat1, has played a central role in CQ resistance" should be omitted as this was not proven, and the authors themselves state that aat1 plays more of a role in fitness. The authors could state that they provide compelling evidence that AAT1 polymorphisms have co-evolved with mutant PfCRT and might contribute to CQ resistance and/or restoration of reduced parasite fitness incurred as a result of PfCRT mutations.

18. Line 281: Referring to the text "low fitness of parasites carrying pfcrT K76T and pfaat1 S258L in the absence of drug pressure". This was only shown in the NHP4026 SEA genetic background. The authors need to clarify that this was demonstrated in a single Asian parasite background before extending their interpretation to discuss parasite infections in Africa

19. Lines 294-5: It is unclear whether PfaAT1 is strictly located in the DV. In fact in reference 43 that the authors cited, it is written: "PfaAT1 has been localised to the digestive vacuole membrane (Cowell et al., 2018; PMID 29326268), but the use of a non-

endogenous promoter and locus for the expression of the tagged transporter raises the possibility that it might also be present at the parasite plasma membrane (see Section IV3.a).” This caveat should be noted in the description of the AAT1 subcellular location. The additional reference to Cowell et al. (ref 35) should be added.

20. Lines 304-307: For reasons described above, this sentence is highly speculative and needs to be moderated, with the Figure moved to the supplement.

21. Figure S2: Please define in the legend how the haplotype structures are derived.

22. Figures S8 and S9: Please also place an arrow to show the location of pfcr1. Also, in Figure S8 There appears to be a minor signal with the chromosome 14 locus.

23. Figure S10: I would recommend changing the legend from “At the chr.14 QTL region, allele frequencies are unchanged following drug treatment, suggesting this QTL is unrelated to drug treatment” to “At the chr.14 QTL region, allele frequencies show no to little change following drug treatment, suggesting this QTL is unrelated to drug treatment.”

24. Figure S14 legend: The legend has some very speculative text about amino acid transport, which has not been shown in this study. I would recommend changing the text to state: “transmembrane region that includes helices 3, 5, and 8 potentially allowing for partial restoration of the predicted amino acid transport activity”.

Reviewer #4 (Remarks to the Author):

All comments were satisfactorily addressed. Additional experiments to further explore the *aat1* mutations in African parasite backgrounds are planned.

Author Rebuttal, first revision:

2/24/2023

[redacted]

Reviewer Expertise:

Referee #2: Plasmodium biology, drug resistance

Referee #4: Plasmodium biology, antimalarial drugs

52Reviewers Comments:

Reviewer #2 (Remarks to the Author):

In this resubmission by Nwa et al., the authors have made substantial positive improvements to the manuscript including additional haplotypic and IBD analysis across Africa and other regions, incorporating new Figures S3-S6, and rewriting and clarifying points in the main results and methods. The convergence of population genomic and genetic cross data is an exciting data set that clearly shows some selective pressure on the AAT1 region and provides evidence that mutations in this gene can contribute to modulating chloroquine (CQ) resistance and parasite fitness. There is some compelling longitudinal evidence that the S258L mutation in AAT1 in west Africa has evolved in parallel with mutations in PfCRT. The latter protein is known to be the primary driver of CQ resistance via a gain of drug efflux. The study raises a number of interesting questions that will spur much more research in the field.

Thank you for these positive comments

I definitely support publication of this study, which I think is an important milestone for the field. There are, however, several changes that I think are vital to make before the manuscript is ready for publication. These are detailed in my lengthy comments listed below. I apologize for some overlap between them as it took multiple rereads here to work through this large and complex dataset. Below is a list of my primary concerns:

1. The genetic cross data in Figure 4 does not fully match the gene editing data in Figure 5. The CQ selection of the bulk progeny populations in Figure 4a shows enrichment for the mutant (MUT) pfcr allele from the resistant parent NHP4026 (that expresses the “Dd2” PfCRT haplotype with 8 mutations separating it from the wild-type allele present in 3D7), with most of those progeny also carrying MUT AAT1 (carrying the S258L and F313S mutations). Those data are consistent with prior knowledge that MUT PfCRT has a substantial fitness cost. Only two MUT PfCRT progeny carry wild-type (WT AAT1 (i.e. S258 and F313), despite the WT allele being present in ~40% of the progeny recovered without drug pressure. That would suggest that MUT AAT1 is contributing to CQ resistance, or is epistatically involved with MUT PfCRT presumably by compensating for a physiologic growth defect. The NHP4026 gene editing data, however, shows that MUT AAT1 has a CQ IC50 value slightly lower than WT AAT1, which

would suggest that MUT AAT1 does not augment CQ resistance (CQR). Similarly, MUT AAT1 has a substantially lower fitness cost than WT AAT1 in that background, arguing against the idea that MUT AAT1 is being selected by CQ pressure in the progeny because it (epistatically) enhances the growth of MUT PfCRT progeny. This should be clearly spelled out. Figure S11 is similar in showing that MUT PfCRT progeny have no significant differences in mean CQ IC50 values between the two progeny with WT AAT1 and the six phenotyped MUT AAT1 progeny, again suggesting that MUT AAT1 is not driving the enrichment for CQ-selected progeny that inherited that region of chromosome 6. One alternative interpretation of the data is that the chromosome 6 segment that is enriched from the NHP4026 parent in the CQ-selected progeny is close to but not actually AAT1. That would be consistent with a genetic and population genomic signal that tracks to AAT1 as a nearby molecular marker. The authors need to point out this disconnect between the gene editing and CQ bulk selection studies and also evoke the alternative hypothesis that AAT1 may be a marker of a region that co-evolved with CQR.

We have pointed out the discrepancy between the results from gene editing and the genetic cross and we have suggested a possible explanation in the text (lines 232-237), where we wrote: *“We speculate that these opposing results may reflect differing selection pressures in blood stage parasites in the case of CRISPR experiments, or in the mosquito and liver stages of the life cycle in the case of genetic crosses”*. We thank the reviewer for suggesting a possible alternative explanation. However, we think that the strong impact of CRISPR/Cas9 gene editing of pfAAT1 mutations on both IC50 and fitness provide strong evidence that these mutations are responsible. Linked loci are much less likely to explain the data, so we have not included the reviewers suggested explanation.

2. The authors repeatedly state that AAT1 is an amino acid transporter based largely on an earlier yeast expression study (ref. 26 by Tindall et al., 2018, Scientific Reports; the senior author S. Avery is a coauthor on this present submission). In that study the amino acid tryptophan was found to partially inhibit quinine accumulation in yeast cells expressing AAT1. Inferences based on distant sequence homology and initial data in a heterologous expression system should be presented with caution. As an example, a recent publication clearly showed that a protein formerly annotated as a separate amino acid transporter in *P. falciparum* is actually essential for calcium mobilization and serves as a crucial link between calcium signaling and protein kinase G function that is required for parasite egress from infected red blood cells (PMID 33762339). As the authors know, expression and functional characterization of malarial proteins in yeast can often be problematic because of their highly divergent AT contents and codon usages. The tryptophan inhibition data do not in my opinion provide sufficient experimental evidence of amino acid transport. I would strongly recommend that the authors modify their text to consistently refer to AAT1 as a putative amino acid transporter, including in the title. Of

note, Tindall et al. also consistently refer to this as a putative amino acid transporter. Also, there is also only preliminary evidence that this protein can transport CQ, again based on yeast data.

We certainly agree with the reviewer that caution is needed, and now refer to pfAAT1 as a putative amino acid transporter throughout the manuscript, and in the revised title.

3. As best I know, there are no amino acid or peptide metabolomics data or drug accumulation studies focusing on AAT1 with isogenic parasite lines that would support their model that AAT1 can transport amino acids or drugs in parasites. As such, their Figure 6 model is, in my opinion, too speculative and should be moved to the supplement. Additional points about Figure 6 are raised below.

The model we present is based on our current understanding of this system, and our hope is that it will stimulate future work on this topic. We have added a sentence stating this (lines 317-318). We have now moved this to the supplement (Fig S15) as suggested.

4. The authors have yet to experimentally address an important caveat of the current study, which is that they have not edited AAT1 in an African line with a regional, non-Dd2 PfCRT haplotype to prove the co-evolution and functional relationship between AAT1 and PfCRT, especially to validate their proposal that the AAT1 S258L mutation was required for the evolution of PfCRT K76T in African backgrounds. Other reviewers also pointed this out and requested that the same gene editing experiment be performed. The authors indicate that they recently obtained funding to conduct this work and that it will be the subject of a future study. The authors nonetheless need to explicitly address this limitation. It is established that African and Asian PfCRT haplotypes differ in important ways, namely at residues 326 and 356, and so stating that both regions share the common CVIET haplotype is only part of the more complex story here in terms of how PfCRT haplotypes differ regionally. I acknowledge the vast amount of high-quality work already performed for this study and thus am not insisting that the paper be put on hold until those studies are complete. At a minimum, however, the authors must acknowledge that further gene editing studies in African parasites with local haplotypes are necessary to validate that the AAT1 S258L mutation was required for the evolution of mutant K76T-containing PfCRT in African backgrounds. They should also cite prior studies that document the different PfCRT haplotypes present in Asian versus African parasites (PMID 26208441, 31040246). Of note, the PfCRT Dd2 haplotype studied herein (in NHP4026 from Southeast Asia) was not detected at all in an earlier analysis of nearly 800 African *P. falciparum* genomes (PMID 31040246).

We now clearly address this limitation of our study in the discussion (lines 237-243), discuss the different pfCRT haplotypes present in Africa and Asia and cite the papers mentioned: *“While African pfCRT CQR alleles originated in SEA and share a common ancestor and identity at amino acids 72-76, most SEA parasites (including NHP4026) carry one or two additional mutations in pfCRT (N326S and I356T) associated with higher CQ IC₅₀ and reduced fitness (PMID 31040246). The predominant pfCRT haplotype in the Gambia differs from NHP4026 at one amino acid, carrying the ancestral 326S, while NHP4026 carries the 326N mutation. It will be important to examine the impact of pfAAT1 mutations on African genetic backgrounds in future work.”*

5. In light of the comments raised above, I feel the authors have yet to sufficiently prove that “the amino acid transporter AAT1 plays a pivotal role in chloroquine resistance evolution”, as is currently stated in their title. I would recommend a more circumspect title, e.g. by saying “Genomic and genetic evidence implicates (or “implicating”) the putative amino acid transporter AAT1 in the evolution of Plasmodium falciparum chloroquine resistance”.

The title has been modified to be more circumspect and now reads:

“Evidence implicating the putative amino acid transporter AAT1 in chloroquine resistance evolution”

This is 97 characters, so just above the 90 characters guideline

There also remain author responses to three of my earlier major comments in which I have additional comments that are highlighted below (the authors’ replies are listed in italics). Below these I list additional comments.

Earlier major comment 1: The authors present an identity-by-descent (IBD) analysis to demonstrate the strong co-selection of mutations in AAT1 and the CRT CQR marker mutation K76T in a set of 321 isolates from The Gambia, sampled from 1984 to 2014. It is not clear why this analysis has been limited to only samples from The Gambia when the MalariaGen Pf6 dataset contains more than 3,000 samples from various regions in Africa (note: researchers leading that effort are already are co-authors). It is unclear why the authors conducted their longitudinal analysis solely on The Gambia – perhaps because they had access to samples from before CQR spread there? There are countries such as Mali where CQ use was

discontinued and CRT reverted to wild-type (WT, i.e. K76) and it would be interesting to know the trajectory of AAT mutations in those areas. If AAT1 and CRT were co-evolving, as is proposed, can the authors explain why the prevalence of mutant CRT K76T does not correlate with the proportions of AAT1 S258L between the various continents and African regions (Figure 2A)? Is that association holding only in West Africa?

Authors' reply: The reviewer is correct – we focused on longitudinal samples from the Gambia because we (Alfred Amambua Ngwa, David Conway, Umberto D'Allessandro) had access to longitudinal samples collected from between 1984 and 2014 from the Gambia. Such longitudinal samples are not available in the PF6 database from other regions of Africa. We certainly agree that parallel analyses in Malawi would be extremely valuable, but we do not have access to these samples or sequence data.

With regard to correlations between PfaAT1 and PfcCRT mutations. We thank the reviewer for the suggestion to examine PfcCRT haplotypes and PfaAT1 evolution in more depth across Africa. In the light of the reviewer's comments, we have expanded our analysis to examine distribution of PfaAT1 and PfcCRT alleles across Africa. This new fine-grained analysis across Africa clearly shows that PfcCRT K76T correlates strongly with the proportions of PfaAT1 S258L in West Africa ($R^2 = 0.65$, $p=0.0017$) and across all African populations ($R^2 = 0.44$, $p=0.0021$). This analysis further strengthens the argument for co-evolution and epistasis between these two genes. We have added this to the main text (lines 126-130); the analysis is provided as Supplementary Fig S6.

New Comment 1.1: It is excellent to see this detailed in-depth analysis of AAT and PfcCRT mutations across Africa (Figure S6). The correlation in west Africa is striking and provides important data to link the evolution of both sets of polymorphisms. It is interesting that the East African isolates (Kenya and Tanzania) did not show a clear association of AAT S258L with PfcCRT CVIET. Instead, parasites in these countries were mostly PfcCRT WT but yet were almost 100% S258L for AAT1, suggesting a fixation of this mutant AAT1 allele. This is rather surprising given that the S258L mutation alone made the *aat1*-edited mutant *pfcr*t NHP4026 parasites less fit (Figure 5c). The authors should explicitly address this. Also, given that the S258/F313 WT parasites were shown to be fitter than the S258L/F313S or S258L/F313 parasites (Figure 5), can the authors provide an explanation as to why we do not observe higher frequencies for the S258/F313 WT AAT1 in clinical isolates across Asia and Africa, especially after the removal of CQ usage?

We thank the reviewer for this observation. Our CRISPR/Cas9 experiments were conducted on a parasite carrying *pfcr*t-CVIET CQR alleles. In this background we observed reduced fitness of the *pfat1*-S258L

57allele. However, parallel experiments are required to understand the impact of pfAAT1 mutations on fitness in parasites carrying the WT pfCRT CQS background (as observed in East Africa). We now have funding to conduct these experiments, and will be able to answer the reviewer's question when these experiments have been completed. However, this is beyond the scope of the current paper.

Earlier major comment 7. What is the NHP4026 CRT haplotype? How does this compare to CRT haplotypes in The Gambia or other African regions? Among CRT K76T mutants, several haplotypes of CRT exist and their prevalence markedly differs between geographical regions (see PMID 33824913). It is rather surprising that the authors have not mentioned this when the AAT1 haplotypes are well documented in this manuscript. Also, it would be important to test the impact of the AAT1 Southeast Asian mutations S258L+F313S on CQ IC50 and fitness in 3D7 as well as other African parasites from The Gambia and Mali. The manuscript would be strengthened by adding data from AAT1 CRISPR/Cas9 editing on African parasite lines. This would help test the authors' main hypothesis that AAT1 plays a central role in CQR evolution, especially since the co-selection was observed in parasites from The Gambia.

Authors' reply: We have now expanded the population genomic analysis to examine the distribution of pfCRT haplotypes in relation to PfAAT1 mutations (see revised Fig 2 and Fig S6).

The NHP4026 parasite used for editing carries the CVIET pfCRT haplotype common in Asia and Africa (see also our reply to reviewer 1), as expected. This makes it a suitable parasite for gene editing studies. We agree with the reviewer that gene editing of pfAAT1 in African parasite genetic backgrounds will be valuable, a point that has led to a recently funded proposal to conduct this work. We will conduct this work over the next few years and will report these data in subsequent papers. However, this is beyond the scope of the current manuscript.

New comment 7.1: The revised Figure 2 does not indicate the full PfCRT haplotypes (eg. Dd2, Cam734, 3D7, GB4, Cam783) present in different regions. Earlier analysis of the MalariaGEN Pf3K data, assembled from downloaded genome data, found that African and Asian PfCRT haplotypes show important differences. The Dd2 haplotype present in the current Asian parent NHP4026 was absent from African parasites (out of 783 genomes sequenced). Instead, African mutant PfCRT haplotypes were predominantly GB4 or Cam783, which differ at positions 326 and/or 356 (PMID 31040246). While these parasites share the CVIET haplotype at positions 72-76, it is not correct to state that Dd2 was introduced into Africa and swept across the region, in contrast to the statement from Reviewer 1. Either Dd2 migrated and later "devolved" to simpler haplotypes that lost the mutations at positions 326 and/or

356, or the alternative GB4 or Cam783 allele(s) spread from Asia to Africa. The authors should clarify their text accordingly. Ideally, the authors could present a revised PfCRT haplotype analysis for African strains based on the latest reported Pf7K genome database, if that analysis could be conducted quickly, as this would provide a more thorough analysis of associations between AAT1 and PfCRT. If not possible, at least the authors should acknowledge the different haplotypes and state that their gene editing did not examine African haplotypes.

Thank you for pointing this out. We now mention these differences between Asian and African pfCRT sequences examined in the revised manuscript. We have also indicated that further gene editing studies of *pfAAT1* in an African background should be conducted. We have included the following sentences in the revised paper: *“While African pfCRT CQR alleles originated in SEA and share a common ancestor and identity at amino acids 72-76, most SEA parasites (including NHP4026) carry one or two additional mutations in pfCRT (N326S and I356T) associated with higher CQ IC₅₀ and reduced fitness (PMID 31040246). The predominant pfCRT haplotype in the Gambia differs from NHP4026 at one amino acid, carrying the ancestral 326S, while NHP4026 carries the 326N mutation. It will be important to examine the impact of pfAAT1 mutations on and African genetic background in future work.”* (lines 237-243).

Earlier major comment 9. The authors should also further clarify their gene editing data. I'm unclear whether the 258L/313S mutant described in Figure 5 is the NHP4026 parent edited to express the same mutations (with silent mutations to prevent cleavage involving the PAM site) or whether that was NHP4026 and the only gene-edited mutants were the other three combinations. Ideally the best is to have gene-edited all four combinations, and better yet would be to have a control with the silent mutations introduced to prevent re-cleavage of the edited locus.

Authors' reply: The PfAAT1 S258L/F313S mutant described in Figure 5 is NHP4026. This is detailed in the methods (line 692-693), and in Fig 5A, and is now further clarified in the Fig 5 legend. We did not conduct control edits with shield mutations only – this is not common practice in malaria work using CRISPR/Cas9, because malaria lacks the error-prone non-homologous end joining repair system that generates off-target changes in other eukaryotes. However, we bolstered our approach by conducting three independent edits (edits recovered from independent flasks) for each genotype constructed (11 CRISPR edits in total). Each of these show very similar results for both IC₅₀ and fitness assays, providing strong evidence that the phenotypes observed do not result from off target mutations.

New comment 9.1: The 11 independent gene edits performed are biological repeats for 3 separate editing events (two single mutants and the double wild-type reversion) in a single Asian parasite. As

59stated elsewhere, the authors should acknowledge that additional gene editing studies will be required to confirm these data in other genetic backgrounds, including ones with African PfCRT haplotypes. Also, while I agree that binding site mutants are not essential because of the lack of non-homologous end joining, these controls remain quite standard practice. The authors should add a note in their legend that a binding-site control mutant was not generated, on the basis that *P. falciparum* lacks error-prone non-homologous end joining (PMID 25184562).

We now mention the need for gene editing studies in parasites with an African background in the text (lines 237-243). We also note in the methods section (lines 699-700) that binding site control mutations were not conducted and include the sentence “*Binding-site control mutants were not generated, as P. falciparum* lacks error-prone non-homologous end joining (PMID 25184562).”

Please see below a series of additional comments. Some of these overlap with comments raised above but here relate to specific lines.

6. Line 41-2: “mutations in [pfcr]...confer CQ resistance in *Plasmodium falciparum*, but typically affect parasite fitness”. This statement suggests that fitness is the main phenotype, not CQ resistance, which is not accurate. I would suggest rewording to ““mutations in [pfcr]... confer CQ resistance in *Plasmodium falciparum*, and can also affect parasite fitness”.

We have changed this sentence to read: “...and typically also affect parasite fitness” (line 42)

7. Line 49: To be completely transparent, this should be revised to say “Parasite genetic crosses then identified a chromosome 6 quantitative trait locus containing pfaat1 (harboring the S258L/F313S mutations) that is selected by CQ treatment.” The next sentence should read “Gene editing demonstrated that pfaat1 S258L with F313 potentiates CQ-resistance...”

We have not made these changes due to abstract word limitations, and because this level of detail will make the abstract unclear. However, the additional detail suggested is provided in the manuscript text.

8. Fig S3 (cited on line 118): The authors observe an additional chromosome 10 peak in West Africa in the IBD analysis that is not observed in the West African Gambian analysis in Figure 1. As this peak is not observed in Asia, could this also be a potential region that explains differences in PfCRT haplotype

prevalence in Asia vs Africa? The authors should refer to this peak in the Figure S3 legend and if possible, provide any more information about this region.

The chr. 10 peak (West Africa, A) is interesting, and contains *pfmspdbl2* associated with decreased sensitivity to halofantrine, mefloquine and lumefantrine (PMID 23587962). The chr. 8 peak (East Africa and Asia (C-H)) contains dihydropteroate synthase (sulfadoxine resistance) (PMID 9395372) and the chr 12 peak (D-F) contains GTP cyclohydrolase I, a compensatory locus for antifolate drugs (PMID 25694157). This information is now included in the Fig. S3 legend (lines 946-950).

9. Line 150: 3D7 is stated to be a West African parasite but this is up for debate. Earlier suggestions were based on early partial microsatellite data. The most comprehensive analysis I know of was from Aline Uwimana and colleagues (PMID 3274827) who analyzed >300 genomes collected worldwide and found 3D7 to cluster next to parasites from Rwanda (east-central Africa). Maybe just state “African parasite” and add that citation.

Thank you for pointing out the Uwimana *et al.* paper. We now state that 3D7 is an “African” rather than “West African” parasite in the text (line 151). This is also stated in the methods (line 600), and we cite PMID 32747827.

10. Line 193: “...is consistent with the pfAAT1 QTL being driven by parasite fitness in our genetic crosses.” This is a strong statement that needs to be moderated. As stated above, there was a discrepancy in the fitness data from the genetic cross results. In Figure 4a, it was shown that NHP4026 MUT AAT1 (S258L/F313S) was selected for over the WT S258/F313, along with mutant PfCRT (CVIET) in the presence of CQ, suggesting that the MUT S258L/F313S AAT1 may be involved in CQR and/or compensate for mutant pfCRT-mediated fitness defects. However, in Figure 5, they show that when comparing their PfCRT mutant NHP4026 lines that expressed MUT S258L/F313S or WT S258/F313 AAT1, the WT (that was selected against under CQ pressure) was actually more fit and also slightly more CQR than S258L/F313S. These results are contradictory to those observed in Figure 3. In addition to their proposed explanation on lines 232-235, the authors could state that other genes in the chromosome 6 QTL segment may play a role and also that additional genetic backgrounds need to be tested to further examine these associations.

Please see answer to point 1 and 4 and revised text (lines 237-243). We now state that experiments on different genetic backgrounds are needed. However, we do not think that suggesting that other genes in the chr 6 QTL segment may be involved is warranted given the strong phenotypic impact observed in pfAAT1 CRISPR/Cas9 editing experiments.

11. Lines 195-208: The text reads “This revealed a highly significant impact of the S258L mutation, which increased CQ IC50 1.5-fold, and a more moderate but significant impact of F313S and the double mutation (S258L/F313S) (Fig. 5b, Supplementary Table 8). The observation that 258L shows an elevated IC50 only in combination with the ancestral 313F allele reveals an epistatic interaction between these amino acid variants (Fig 5B).” This doesn’t fully correspond to the data shown in Figure 5. Those data show that the greatest increase in CQ IC50 is with S258L + F313. By comparison, S258L + F313S is 1.5 fold less resistant and has an even slightly lower IC50 than the WT S258 + F313. Those data would argue that only the S258L + F313 combination is able to augment CQR mediated by mutant PfCRT and that F313 is key. Indeed, S258L + F313S has a lower CQ IC50 than S258 + F313S, arguing that in the presence of the F313S mutation the S258L mutation makes parasites less CQ-resistant. The explanation as presently stated can easily be misinterpreted and needs revision. The title of that paragraph should also be changed, for example to say “Gene editing evidence that polymorphisms in AAT1 can modulate CQ resistance in mutant pfCRT parasites”.

We have modified the last two sentences of the paragraph to clarify this (lines 224-226) The last sentence now reads “The observation that 258L shows reduced IC₅₀ in combination with the F313S mutation reveals an epistatic interaction between these amino acid variants (Fig 5B). We think that the paragraph title suggested is overly complex and will reduce the clarity of the manuscript. We have not changed this.

12. Coming back to Figures 5 and 6: If AAT1 were actually a mediator of fitness in the mutant PfCRT background, and WT AAT1 + MUT PfCRT was shown herein to be the most fit (Figure 5c), then this haplotype would have been expected to take over, not S258L AAT1 as in the Gambia or haplotypes containing F313S as in Southeast Asia (SEA). Further work is clearly required to further disentangle these relationships. For now, I would advocate to place Figure 6 in the supplement given the preliminary basis of this model. Also, the model needs to include a fourth panel with WT AAT1 + MUT CRT, which is presently lacking.

We have now moved the model to the supplement (Fig S15). We have added the fourth panel and agree that this improves the Figure. Please see answer to point 3 also.

13. Lines 227-8: “However, S258L carries a high fitness cost that in SEA parasites was likely mitigated by addition of the compensatory substitution, F313S.” However, it appears that a very small minority of the SEA parasites have S258L only, while the majority of the population has F313S + another mutation, including S258L, Q454E, or K541N. Could these data suggest that F313S is the predominant AAT1 haplotype in SEA, and that other mutations, including S258L, arose on this background?

Determining the order of pfAAT1 mutations in SEA is problematic. This is not clear from phylogenetic or network analyses, and is potentially confounded by recombination. We have therefore reworded this section to avoid any assumptions about the order in which mutations arose (lines 224-228). However, we agree with the reviewer that the appearance of F313S in several SEA pfAAT1 alleles may suggest that this mutation arose originally. We now include the sentence “*The pairing of F313S with three different mutations, suggests that F313S arose first.*” (line 142-143) in the section entitled *Divergent selection on pfaat1 in Southeast Asia*.

14. Line 239: “CQ uptake can be competitively inhibited...” This sentence needs to be modified to show that the data were produced by another group. This can be addressed by changing this sentence to “CQ uptake was previously shown to be competitively inhibited...”

Change made (line 247)

15. Line 244: The authors use citation 26 to state that “expression of another amino acid variant (T162E)...restores cell growth in the presence of CQ.. Together, these new and published results suggest that yeast expression of pfaat1 mutations impact resistance and fitness by altering the rates of amino acid and CQ transport.” However, this is not an accurate interpretation of the data in that article, which provides evidence that expression of the T162E AAT1 mutant resulted in increased yeast cell growth in the presence of an exceptionally high concentration of CQ (1 mM). The sentence as it stands needs to be modified to remove any reference to fitness and amino acid or CQ transport, as the results presented therein related only to growth. A revised sentence could read as: “expression of another amino acid variant (T162E)...restores cell growth in the presence of 1 mM CQ”. The next sentence needs to be deleted as neither the Tindall et al. paper nor Figure S13 present data on fitness or amino acid or CQ

63transport. Also, it would have been preferable to include the T162E mutant as a control in their present study to see how those data matched the prior publication, especially as the current study used 5 mM CQ and the prior study used 1 mM CQ.

As requested, we have specified it was at 1 mM CQ that yeast cell growth was restored. We have made certain other minor wording revisions suggested to help further avoid any possible ambiguity in this text (lines 244-254). However, we don't think that the paragraph describing the yeast work contains inaccurate or misleading statements. The Tindall *et al.* paper clearly demonstrates that (i) expression of codon-optimized pFAAT1 in yeast results in increased uptake of CQ, which was directly measured with the chloroquine probe LynxTag-CQ™; (ii) Uptake of quinoline drugs was inhibited by an aromatic amino acid (tryptophan) consistent with pFAAT1 also acting as an amino acid transporter; (iii) addition of the mutation T162E reduced quinoline uptake and increased growth rates of yeast in the presence of drug. Yeast growth rates are used to measure resistance, just as IC₅₀ measurement examines growth inhibition of *Plasmodium* in the presence of drugs.

We therefore stand by our statement *“Together, these new and published results suggest that yeast expression of pfaat1 mutations impact resistance and fitness by altering the rates of amino acid and CQ transport.” We agree that inclusion of T162E mutant alongside the S258L mutant reported here would have been a useful, although non-essential control. However, this was not done.*

16. Lines 287-8: “restoration of fitness by F313S may help to explain retention of CQ-resistant pfCRT K76T alleles in SEA.” This is very speculative, as there is no temporal evidence that F313S “restored” fitness costs mediated by S258L. In fact, the evidence points more to the possibility of F313S being the more dominant, ancestral haplotype in SEA, upon which other mutations arose. The text could be modified to state: “We speculate that restoration of fitness by F313S may help to explain retention of CQ-resistant PfCRT K76T alleles in SEA”.

Change made (line 295)

17. Line 261-2: The text “we find clear evidence that a second locus, pfaat1, has played a central role in CQ resistance” should be omitted as this was not proven, and the authors themselves state that aat1 plays more of a role in fitness. The authors could state that they provide compelling evidence that AAT1

polymorphisms have co-evolved with mutant PfCRT and might contribute to CQ resistance and/or restoration of reduced parasite fitness incurred as a result of PfCRT mutations.

We have modified this sentence (now lines 269-270) to read “we find compelling evidence that a second locus, *pfaat1*, has played an important role in CQ resistance evolution. The weight of this evidence is then discussed in the discussion paragraphs below.

18. Line 281: Referring to the text “low fitness of parasites carrying *pfcr*t K76T and *pfaat1* S258L in the absence of drug pressure”. This was only shown in the NHP4026 SEA genetic background. The authors need to clarify that this was demonstrated in a single Asian parasite background before extending their interpretation to discuss parasite infections in Africa

This caveat is now clearly stated in lines 237-243

19. Lines 294-5: It is unclear whether PfAAT1 is strictly located in the DV. In fact in reference 43 that the authors cited, it is written: “PfAAT1 has been localized to the digestive vacuole membrane (Cowell et al., 2018; PMID 29326268), but the use of a non-endogenous promoter and locus for the expression of the tagged transporter raises the possibility that it might also be present at the parasite plasma membrane (see Section IV3.a).” This caveat should be noted in the description of the AAT1 subcellular location. The additional reference to Cowell et al. (ref 35) should be added.

We have now cited Cowell et al (PMID 29326268) on line 302 of the text. We have also added a sentence (lines 1039-1041) to Supp S13 legend stating “Published results demonstrate that AAT1 is expressed in the yeast cell membrane²⁶, while *pfaat1* localizes to the *Plasmodium* digestive vacuolar membrane (PMID 29326268), but may also be present in the plasma membrane.” (lines 1039-1041)

20. Lines 304-307: For reasons described above, this sentence is highly speculative and needs to be moderated, with the Figure moved to the supplement.

As stated, this is a model based on available data, and provides a working hypothesis that is intended to guide future research. We have moved this Fig 6 to the supplement as suggested (now Fig S15)

21. Figure S2: Please define in the legend how the haplotype structures are derived.

We now include the text in the figure legend (lines 934-935) stating “*Haplotype relationships were based on Identity-by-Descent of genome segments encompassing 25kb on either side of each gene (see methods).*” We also include details of how this analysis was conducted in the methods section (lines 557-562), as it was inadvertently omitted in prior manuscript versions. Thank you for catching this.

22. Figures S8 and S9: Please also place an arrow to show the location of *pfcr1*. Also, in Figure S8 There appears to be a minor signal with the chromosome 14 locus.

We have added a sentence to the figure S8 and S9 legends (lines 997 and 1004) stating: “*The peak on chr. 7 contains pfCRT.*” The minor signal on chr. 14 is not driven by drug treatment – see point 23 below.

23. Figure S10: I would recommend changing the legend from “At the chr.14 QTL region, allele frequencies are unchanged following drug treatment, suggesting this QTL is unrelated to drug treatment” to “At the chr.14 QTL region, allele frequencies show no to little change following drug treatment, suggesting this QTL is unrelated to drug treatment.”

Change made (line 1013)

24. Figure S14 legend: The legend has some very speculative text about amino acid transport, which has not been shown in this study. I would recommend changing the text to state: “transmembrane region that includes helices 3, 5, and 8 potentially allowing for partial restoration of the predicted amino acid transport activity”.

Change made (line 1058)

Reviewer #4 (Remarks to the Author):

All comments were satisfactorily addressed. Additional experiments to further explore the aat1 mutations in African parasite backgrounds are planned.

Decision Letter, second revision:

Message: Our ref: NMICROBIOL-22071823B

28th February 2023

Dear Professor Anderson,

Thank you for submitting your revised manuscript "Evidence implicating the putative amino acid transporter AAT1 in chloroquine resistance evolution" (NMICROBIOL-22071823B). Editorially, we find that the paper has improved in revision, and therefore we'll be happy in principle to publish it in Nature Microbiology, pending minor revisions to comply with our editorial and formatting guidelines.

Thank you again for your interest in Nature Microbiology Please do not hesitate to contact me if you have any questions.

Sincerely,
[redacted]

Final Decision Letter:

Mes 29th March 2023

sag

e: Dear Professor Anderson,

I am pleased to accept your Article "Chloroquine resistance evolution in Plasmodium falciparum is mediated by the putative amino acid transporter AAT1" for publication in Nature Microbiology.

67Thank you for having chosen to submit your work to us and many congratulations.

Acceptance of your manuscript is conditional on all authors' agreement with our publication policies (see <https://www.nature.com/nmicrobiol/editorial-policies>). In particular your manuscript must not be published elsewhere and there must be no announcement of the work to any media outlet until the publication date (the day on which it is uploaded onto our website).

Please note that *Nature Microbiology* is a Transformative Journal (TJ). Authors may publish their research with us through the traditional subscription access route or make their paper immediately open access through payment of an article-processing charge (APC). Authors will not be required to make a final decision about access to their article until it has been accepted. [Find out more about Transformative Journals](https://www.springernature.com/gp/open-research/transformative-journals)

Authors may need to take specific actions to achieve [compliance](https://www.springernature.com/gp/open-research/funding/policy-compliance-faqs) with funder and institutional open access mandates. If your research is supported by a funder that requires immediate open access (e.g. according to [Plan S principles](https://www.springernature.com/gp/open-research/plan-s-compliance)) then you should select the gold OA route, and we will direct you to the compliant route where possible. For authors selecting the subscription publication route, the journal's standard licensing terms will need to be accepted, including [self-archiving policies](https://www.nature.com/nature-portfolio/editorial-policies/self-archiving-and-license-to-publish). Those licensing terms will supersede any other terms that the author or any third party may assert apply to any version of the manuscript.

If you have any questions about our publishing options, costs, Open Access requirements, or

68our legal forms, please contact ASJournals@springernature.com

Congratulations once again and I look forward to seeing the article published.

With kind regards,
[redacted]

P.S. Click on the following link if you would like to recommend Nature Microbiology to your librarian <http://www.nature.com/subscriptions/recommend.html#forms>

** Visit the Springer Nature Editorial and Publishing website at http://editorial-jobs.springernature.com?utm_source=ejp_NMicro_email&utm_medium=ejp_NMicro_email&utm_campaign=ejp_NMicro for more information about our career opportunities. If you have any questions please click [here](mailto:editorial.publishing.jobs@springernature.com).**